

# The Parisi-Sourlas uplift and infinitely many solvable 4d CFTs

**Emilio Trevisani**

Laboratoire de Physique Théorique et Hautes Énergies, CNRS & Sorbonne Université,
4 Place Jussieu, 75252 Paris, France
Université Paris-Saclay, CNRS, CEA, Institut de Physique Théorique,
91191, Gif-sur-Yvette, France

## Abstract

Parisi-Sourlas (PS) supersymmetry is known to emerge in some models with random field type of disorder. When PS SUSY is present the $d$-dimensional theory allows for a $d-2$-dimensional description. In this paper we investigate the reversed question and we provide new indications that any given $\text{CFT}_{d-2}$ can be uplifted to a PS SUSY $\text{CFT}_d$. We show that any scalar four-point function of a $\text{CFT}_{d-2}$ is mapped to a set of 43 four-point functions of the uplifted $\text{CFT}_d$ which are related to each other by SUSY and satisfy all necessary bootstrap axioms. As a byproduct we find 43 non trivial relations between conformal blocks across dimensions. We test the uplift in generalized free field theory (GFF) and find that PS SUSY is a powerful tool to bootstrap an infinite class of previously unknown GFF observables. Some of this power is shown to persist in perturbation theory around GFF. We explain why all diagonal minimal models admit an uplift and we show exact results for correlators and CFT data of the $4d$ uplift of the Ising model. Despite being strongly coupled $4d$ CFTs, the uplifted minimal models contain infinitely many conserved currents and are expected to be integrable.

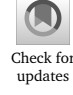

# 1  Introduction

Parisi-Sourlas supersymmetry was first introduced in [1] in order to explain a very peculiar behaviour of critical theories with quenched random field type of disorder. The story started a few years prior when Aharony, Imry, and Ma conjectured that the IR fixed point of random field models is described by pure models in two less dimensions [2]. This was motivated by the behaviour of the Feynman diagrams of the random field Ising model in $d$ dimensions, which in the IR limit match the diagrammatic computations of the pure Ising model in $d-2$ dimensions. Parisi and Sourlas provided an explanation of such behaviour by conjecturing that random field models have emergent supersymmetry in the IR, which they further showed (through a perturbative argument) to be responsible for the dimensional reduction [1]. Namely, following picture 1, the relation **A** of Aharony, Imry, and Ma, was explained by Parisi and Sourlas by a combination of **B** and **C**. The explanation by Parisi and Sourlas is very appealing since the emergence of symmetries at a fixed point is not uncommon. Moreover the original perturbative argument for **C** was later made non-perturbative and more rigorous [3–6]. A CFT argument was also recently provided in [7].

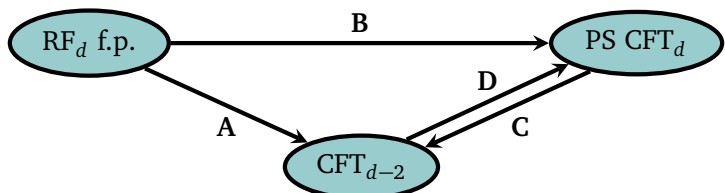

Figure 1: Relations between theories: the fixed point of a random field theories in $d$ dimensions ($RF_d$ f.p.), a Parisi-Sourlas SUSY $CFT_d$ (PS $CFT_d$) and a pure CFT in two lower dimensions ($CFT_{d-2}$).

Unfortunately the beautiful conjecture by Parisi-Sourlas was quickly found to have flaws. In particular the arrow **A** for the Ising model clearly cannot be fully correct. Indeed the RF Ising model has a critical point in $3 \leq d \leq 5$ [8], while the pure $\hat{d} \equiv d-2$-dimensional Ising model only has a critical point in $2 \leq \hat{d} \leq 3$. Thus **A** is certainly violated in $d = 3$. Furthermore in later numerical studies it was found that both **A** and **B** fail in $d = 4$ [9]. It came as a surprise that numerical simulations in $d = 5$ [10,11] gave instead strong indications for the both the emergence of supersymmetry **B** and the dimensional reduction **A**. The Parisi-Sourlas conjecture can also be studied in a different random field model, RF $\phi^3$, which has a phase transition between $d \leq 2 < 8$. In this case numerical simulations indicate that the fixed point is always compatible with **A** [12].

While many studies were produced in the past decades, no fully conclusive explanation of when/why the conjecture works was found. This motivated a revised study of the emergence of SUSY in RF models (arrow **A**) using a perturbative RG setup [13–16]. This series of works showed in epsilon expansion that the PS CFT for the Ising case is reached in the vicinity of the upper critical dimension, namely at $d = 6 - \epsilon$ for small $\epsilon$. Conversely, when $\epsilon$ is of order 1, some SUSY breaking deformations develop a large negative anomalous dimension becoming relevant for $d$ less than a critical value $d_c$. A two-loop analysis predicts that $d_c \approx 4.2 - 4.7$, which is in perfect agreement with the numerical simulation described above. The same setup was also applied to the RF $\phi^3$ model, showing that in this case all SUSY breaking deformations remain irrelevant in all dimensions, again in agreement with the numerical results. In summary it was found that the SUSY fixed point sometimes is not reached because of new relevant SUSY breaking perturbations. However, whenever it is reached, dimensional reduction is expected to always occur. This thus provides an explanation for when/why the Parisi-Sourlas conjecture works.

While this could seem the conclusion of a story, by revisiting the topic new interesting directions emerged. In particular, as we mentioned, the arrow **C** of figure 1 was also studied in an axiomatic CFT context in [7]. In this paper it was first defined what is a PS $CFT_d$. Then it was shown in which sense such a theory has a description in terms of a $CFT_{d-2}$. It was shown that a huge part of the spectrum and OPE coefficients of (also non protected) superprimaries of the PS $CFT_d$ exactly matches the CFT data of the $CFT_{d-2}$. E.g. the spectrum and OPE coefficients of *all* the scalar operators would match in the two theories (in $d \geq 3$). This result was obtained by using arguments only based on symmetry and therefore applies to any possible PS $CFT_d$. The work of [7] thus motivated a new question: given any $CFT_{d-2}$ is it possible to define a PS $CFT_d$ which dimensionally reduces to it? This is what we refer to as the Parisi-Sourlas dimensional uplift, represented as the arrow **D** of figure 1, and it is the focus of this paper. In particular the results of this paper are in support of the conjecture that, independently on the properties of the $CFT_{d-2}$, the uplift always exists. This is mostly independent from random field models, meaning that the uplifted models may or may not emerge at the fixed point of a disordered system. One may thus ask why should we study the uplift. There are various motivations.

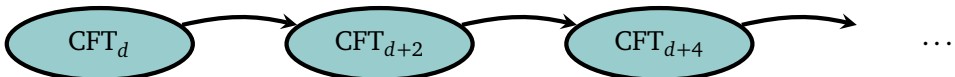

Figure 2: Iterated uplift.

One important motivation is to understand the space of possible consistent CFTs, which is the ultimate goal of the conformal bootstrap. If the uplift always exists, then it can also be iterated. Therefore given a $CFT_d$ one could in principle define an infinite tower of PS $CFT_{d+2n}$ for $n = 1, 2, \ldots$ as shown in figure 2.

There is an important catch: PS CFTs are non-unitary. Therefore figure 2 shows that the space of non-unitary CFTs is infinitely much larger than the one of unitary ones: for every unitary CFT there exists an infinite tower of non unitary CFTs in higher dimension. The tower in figure 2 has increasing amount of SUSY. It is interesting to speculate what could be the fixed point of this sequence (indeed this would live in infinite dimensions and would enjoy infinite SUSY).

Another very interesting motivation is to find the first examples of fully solvable non-trivial CFTs in $d > 2$. Indeed in $d > 2$ there is no example of interacting CFT where the full CFT data is accessible exactly. The situation is much better in $2d$ where in the minimal models it is possible to compute spectrum, OPE coefficients and even correlation functions exactly. The idea is thus to uplift all minimal models obtaining infinitely many towers of infinitely many solvable CFTs in all even dimensions. Notice also that such models might be physically relevant. For example the uplift of the Yang-Lee minimal model describes the $4d$ universality class of RF $\phi^3$ model (which also describes systems like branched polymers and lattice animals [12]). In this universality class there exists also a model defined by Brydges and Imbrie [17] (see also [18]) which has SUSY at the microscopic level. The microscopic SUSY ensures the model to dimensionally reduce all along the RG flow and not only at the IR fixed point. In the same spirit, in [19] Cardy recently proposed a supersymmetric microscopic model that flows to the uplifted Ising model. In $d = 4$ this thus flows to the uplifted Ising minimal model. It would be very interesting to see whether new supersymmetric microscopic theories can be defined such that they flow to other minimal models.

Another motivation is to use the supersymmetry of the uplifted theory as a tool for solving problems of the original theory. This might sound surprising since we argued that the two theories are somewhat equivalent. However the information is packaged very differently in the two formulations and it can happen that the SUSY formulation is more suitable to answer some questions, as we shall exemplify in the following.

Other applications will appear as we will enter the details of the paper. For example the SUSY structure of the uplifted theory provides some powerful kinematical constraint on CFTs. In particular we will see that PS SUSY implies a set of 43 relations between conformal blocks in $d-2$ and in $d$ dimensions, which are interesting in their own rights. For example one of such equation was already obtained by Dolan and Osborn to compute conformal blocks in higher dimensions from the knowledge of conformal blocks in lower dimensions. With this work we find that such equation actually has a very clear physical interpretation in the SUSY theory.

The plan of the paper is as follows. We start section 2 with a basic introduction on CFTs, which will be followed by a review on PS SUSY and dimensional reduction/uplift.

In section (3) we explicitly show how to extract different primary components from superspace correlators with two, three and four scalar insertions. We show that all of them are (as expected) compatible with conformal symmetry which is a further check that PS CFTs are well defined. In the four-point function case we show that there are 43 independent primary components in a generic four-point function which are obtained by acting with 43 differential operators (in the conformal cross ratios) on the lowest component.

In section 4 we show that these differential operators on $d-2$-dimensional conformal blocks can be written in terms of a linear combination of at most five $d$-dimensional conformal blocks. We further analyze these relation and explain how the linear combinations are tuned to cancel possible poles of the conformal blocks.

In section 5 we provide some examples of how to use the supersymmetry of the uplifted theory as a tool to compute observables of the original theory. We find that in the uplift of generalized free field theory (GFF) there exists an infinite class of non trivial correlators which have some vanishing primary components. In terms of four-point functions this means that some of the differential operators of section 3 annihilate the correlator. We then show that this constraint can be used in two ways. Firstly to obtain a recurrence relation for the coefficients in the conformal block decomposition, which we also show how to solve in the case of $\langle \phi \phi^n \phi^m \phi^m \rangle$ in closed form for any $n, m$ (which is a new result in GFF). Secondly we explain how to use the differential operators to fix the correlator itself (up to some constants). Then we also show how this type of logic can be generalized to perturbative computations around GFF, giving rise to differential equations for the perturbative correlators.

In section 6 we start by briefly introducing the minimal models. We give an RG argument of why diagonal minimal models should have a well defined uplift. We then exemplify how to uplift all four-point functions of Virasoro primaries in the Ising minimal model. We then proceed at computing the first few terms of their conformal block decomposition. We end the section by commenting on the properties of the spectrum which must contain an infinite number of conserved higher spin currents.

Section 7 is devoted to a general property of all uplifted theories: they are expected to have a larger set of operators than the reduced ones. These extra operators are projected to zero under dimensional reduction. In section 7 we exemplify a class of such operators. We then consider the uplift of a $1d$ GFF four-point function as a toy model to show how the larger kinematic space of the uplifted theory (in $3d$ there are two independent cross ratios instead of the single cross ratio of one-dimensional CFTs) is exactly reconstructed by the extra operators of the uplifted theory, which in this case are the spin $\ell \geq 1$ traceless and symmetric operators.

In section 8 we conclude the paper and mention future directions. More details are deferred to the appendices. Finally, a Mathematica code with the definitions of the differential operators (and some extra checks) is attached to the publication.

## 2 Background and review of Parisi-Sourlas CFTs

### 2.1 CFT basics

To set our conventions let us start by reviewing some basic facts on $d$-dimensional CFTs (see e.g. [20] for a more detailed review). Correlation functions up to three insertions are fixed by symmetry. One-point functions vanish besides the one of the identity which equals one. The two-point function of a scalar primary $O$ with dimension $\Delta$ takes the form

$$\langle O(x_1)O(x_2)\rangle = \frac{1}{(x_{12}^2)^\Delta}, \tag{1}$$

where $x_{ij}^\mu \equiv x_i^\mu - x_j^\mu$ and $x_i \in \mathbb{R}^d$. In a unitary theory the basis of primary operators can be diagonalized such that the two-point functions of different primaries vanish. Three-point functions of scalar primaries $O_i$ with dimensions $\Delta_i$ take the form

$$\langle O_1(x_1)O_2(x_2)O_3(x_3)\rangle = \frac{\lambda_{123}}{(x_{12}^2)^{\frac{\Delta_1+\Delta_2-\Delta_3}{2}}(x_{13}^2)^{\frac{\Delta_1+\Delta_3-\Delta_2}{2}}(x_{23}^2)^{\frac{\Delta_2+\Delta_3-\Delta_1}{2}}}, \tag{2}$$

where $\lambda_{123}$ is a theory-dependent quantity called OPE coefficient. With four insertions we can build two independent conformal invariant cross ratios,

$$u \equiv \frac{x_{12}^2 x_{34}^2}{x_{13}^2 x_{24}^2}, \qquad v \equiv \frac{x_{14}^2 x_{23}^2}{x_{13}^2 x_{24}^2}, \tag{3}$$

therefore a four-point functions of scalars can be fixed as

$$\langle O_1(x_1)O_2(x_2)O_3(x_3)O_4(x_4)\rangle = K_{\Delta_i}(x_i)f(u,v), \tag{4}$$

where $f(u,v)$ is a theory-dependent function and $K$ is a kinematic factor defined as

$$K_{\Delta_i}(x_i) \equiv \frac{1}{(x_{12}^2)^{\frac{1}{2}(\Delta_1+\Delta_2)}(x_{34}^2)^{\frac{1}{2}(\Delta_3+\Delta_4)}} \left(\frac{x_{14}^2}{x_{24}^2}\right)^{-\frac{\Delta_{12}}{2}} \left(\frac{x_{14}^2}{x_{13}^2}\right)^{\frac{\Delta_{34}}{2}}, \tag{5}$$

where $\Delta_{ij} \equiv \Delta_i - \Delta_j$. Sometimes we shall use different cross ratios as $z, \bar{z}$ and $\rho, \bar{\rho}$ [21]. They can be related to $u, v$ using

$$u = z\bar{z}, \qquad v = (1-z)(1-\bar{z}), \qquad z = \frac{4\rho}{(1+\rho)^2}, \qquad \bar{z} = \frac{4\bar{\rho}}{(1+\bar{\rho})^2}. \tag{6}$$

A crucial property of CFTs is the operator product expansion (OPE) which allows to replace two operator insertions by a sum of single operators insertions,

$$O_1(x)O_2(0) = \sum_{\Delta\ell} \lambda_{12O}\left[\frac{x_{\mu_1}\cdots x_{\mu_\ell}}{|x|^{\Delta_1+\Delta_2-\Delta+\ell}}O^{\mu_1\cdots\mu_\ell}(x) + \ldots\right], \tag{7}$$

where the sum runs over all operators $O$ with dimension $\Delta$ and spin $\ell$ and the dots stand for the contribution of the descendants operators. Using the OPE of operators $O_1$ and $O_2$ inside the four-point function (4), we obtain

$$f(u,v) = \sum_{\Delta\ell} a_{\Delta\ell}\, g_{\Delta\ell}(u,v), \tag{8}$$

where $a_{\Delta\ell} \equiv \lambda_{12O}\lambda_{O34}$. This formula represents the decomposition of a four-point function in conformal blocks $g_{\Delta\ell}$ which are functions fixed by symmetry that encode the exchange of a primary $O$ with dimension $\Delta$ and spin $\ell$ along with all its descendants. The blocks $g_{\Delta\ell}$ are function of the variables $\Delta_{12}$, $\Delta_{34}$ and $d$, but we will often suppress these dependencies in order to streamline the notation.

## 2.2 Conformal blocks

Conformal blocks are fixed by symmetry. In even dimensions $d$ they can be expressed in closed form [22]. For example in $d = 2, 4$ they read [23]

$$g_{\Delta\ell}^{(d=2)}(z,\bar{z}) = \frac{k_{\Delta-\ell}(z)k_{\Delta+\ell}(\bar{z}) + k_{\Delta+\ell}(z)k_{\Delta-\ell}(\bar{z})}{2^{\ell+\delta_{\ell,0}}}, \tag{9}$$

$$g_{\Delta\ell}^{(d=4)}(z,\bar{z}) = \frac{z\bar{z}}{\bar{z}-z}\frac{[k_{\Delta-\ell-2}(z)k_{\Delta+\ell}(\bar{z}) - k_{\Delta+\ell}(z)k_{\Delta-\ell-2}(\bar{z})]}{2^\ell}, \tag{10}$$

where $k_\eta$ is defined as follows

$$k_\eta(z) \equiv z^{\eta/2}\,_2F_1\left(\frac{\eta-\Delta_{12}}{2}, \frac{\eta+\Delta_{34}}{2}, \eta, z\right). \tag{11}$$

Table 1: Position of the poles and labels of the blocks at the residue in (12), where $\Delta_A \equiv \Delta_A^\star + n_A$.

| Type, $n$ | $\Delta_A^\star$ | $n_A$ | $\ell_A$ |
|---|---|---|---|
| I,  $n \in [1, \infty]$ | $1 - \ell - n$ | $n$ | $\ell + n$ |
| II,  $n \in [1, \ell]$ | $\ell + d - 1 - n$ | $n$ | $\ell - n$ |
| III,  $n \in [1, \infty]$ | $\frac{d}{2} - n$ | $2n$ | $\ell$ |

For generic dimensions $d$, the blocks are typically not known in a closed form but nevertheless they can be determined in several ways (see [20] for a review of various techniques). For this paper it will be useful to keep in mind a recursion relation [24–27] that defines the blocks from their analytic structure in $\Delta$. Indeed conformal blocks have poles in $\Delta$ with residue proportional to other blocks,[1]

$$g_{\Delta \ell}^{(d)} \sim \frac{R_A}{\Delta - \Delta_A^*} g_{\Delta_A \ell_A}^{(d)}. \tag{12}$$

The label $A \equiv \text{type}, n$ is specified by a type between $\mathrm{I}, \mathrm{II}, \mathrm{III}$ and an integer $n$ in some range, while all other labels appearing in (12) are defined in table 1. The coefficients $R_A$ at the residue are known in a closed form as follows

$$
\begin{aligned}
R_{\mathrm{I},n} &= \frac{-n(-2)^n}{(n!)^2} \prod_{\delta = \Delta_{12}, \Delta_{34}} \left( \frac{\delta + 1 - n}{2} \right)_n, \\
R_{\mathrm{II},n} &= \frac{-n\,\ell!}{(-2)^n (n!)^2 (\ell - n)!} \frac{(d + \ell - n - 2)_n}{\left( \frac{d}{2} + \ell - n \right)_n \left( \frac{d}{2} + \ell - n - 1 \right)_n} \prod_{\delta = \Delta_{12}, \Delta_{34}} \left( \frac{\delta + 1 - n}{2} \right)_n, \\
R_{\mathrm{III},n} &= \frac{-n(-1)^n \left( \frac{d}{2} - n - 1 \right)_{2n}}{(n!)^2 \left( \frac{d}{2} + \ell - n - 1 \right)_{2n} \left( \frac{d}{2} + \ell - n \right)_{2n}} \prod_{\delta = \Delta_{12}, \Delta_{34}} \prod_{\sigma = \pm 1} \left( \frac{\delta - \sigma \frac{d}{2} - \sigma \ell - n + 1 + \sigma}{2} \right)_n.
\end{aligned}
\tag{13}
$$

One can thus reconstruct the full conformal blocks by summing all the poles and an entire function in $\Delta$. This gives rise to the following recurrence relation of the conformal blocks as functions of the radial coordinates $r \equiv |\rho|$ and $\eta \equiv \cos \arg \rho$,

$$h_{\Delta \ell}(r, \eta) = h_{\infty \ell}(r, \eta) + \sum_A \frac{R_A}{\Delta - \Delta_A^\star} (4r)^{n_A} h_{\Delta_A \ell_A}(r, \eta), \tag{14}$$

where $g_{\Delta \ell}^{(d)}(r, \eta) \equiv (4r)^\Delta h_{\Delta \ell}(r, \eta)$ and the sum over $A$ is a sum over the three types and over $n$. Finally the regular part in $\Delta$ of $h_{\Delta \ell}$ is defined by

$$h_{\infty \ell}(r, \eta) \equiv \frac{\ell! \left( 1 - r^2 \right)^{1 - \frac{d}{2}}}{(-2)^\ell \left( \frac{d}{2} - 1 \right)_\ell} \frac{\left( r^2 - 2\eta r + 1 \right)^{\frac{\Delta_{12} - \Delta_{34} - 1}{2}}}{\left( r^2 + 2\eta r + 1 \right)^{\frac{\Delta_{12} - \Delta_{34} + 1}{2}}} C_\ell^{\frac{d}{2} - 1}(\eta), \tag{15}$$

where $C_\ell^\gamma(x)$ is a Gegenbauer polynomial. The normalization of the blocks is chosen to match the one of the closed form solutions in $d = 2, 4$ written above.

## 2.3 Parisi-Sourlas supersymmetry

In this section we present a short introduction to PS SUSY based on the more complete results of [7].

---

[1] In even dimensions higher order poles are allowed to appear, but here we work in arbitrary $d$ as an analytic continuation around odd $d$, where only single poles appear.

In order to introduce PS supersymmetry it is convenient to work in a superspace $\mathbb{R}^{d|2}$, parametrized by coordinates $y^a = \{x^\alpha, \theta, \bar{\theta}\}$, where the index $a$ takes $d+2$ possible values $a = 1, \ldots d, \theta, \bar{\theta}$. A QFT with PS SUSY is invariant under superrotations and supertranslations which are the transformations that preserve the superspace distance

$$y^2 \equiv y^a y^b (g_{d|2})_{ab}, \quad \text{with} \quad g_{d|2} = \begin{pmatrix} g_d & 0 \\ 0 & g_{sp(2)} \end{pmatrix}, \tag{16}$$

where $g_d$ is just the rank-$d$ identity matrix which defines the metric of a Euclidean $d$-dimensional space, while $g_{sp(2)} = \begin{pmatrix} 0 & -1 \\ 1 & 0 \end{pmatrix}$ is the symplectic metric. We name the generator of supertranslations and superrotations respectively as $P^a$ and $M^{ab}$. They can be represented as

$$P^a = \partial^a, \qquad M^{ab} = y^a \partial^b - (-1)^{[a][b]} y^b \partial^a, \tag{17}$$

where the derivative in superspace is defined as $\partial_a \equiv \frac{\partial}{\partial y^a} \equiv (\partial_\alpha, \partial_\theta, \partial_{\bar{\theta}})$ and the bracket in $[a]$ measures if the index $a$ is bosonic or fermionic, namely $[a] = 0$ for $a = 1, \ldots, d$, while $[\theta] = 1 = [\bar{\theta}]$. A PS CFT is further invariant under superdilation $D$ and special superconformal transformation $K^a$, which are represented as

$$D = -y^a \partial_a, \qquad K^a = 2y^a y^b \partial_b - y^b y_b \partial_a. \tag{18}$$

The full set of generators of a PS CFT satisfies the $osp(d+1, 1|2)$ algebra.

It is convenient to consider local (super) operators $\mathcal{O}(y)$ inserted at points $y^a$ in superspace. We will consider operators that have a well defined superconformal dimension $\Delta$ under the action of the superdilation generator $D$ and that transform in irreducible representations of $OSp(d|2)$, such that tensor indices will be of the form $a_i = 1, \ldots, d, \theta, \bar{\theta}$. Standard $OSp(d|2)$ representations are labelled by Young tableaux with $[d/2]$ rows of length $\ell_1 \geq \ell_2 \geq \ldots \geq \ell_{[d/2]} \geq 0$ and an arbitrary long column, as follows

$$\tag{19}$$

where rows are graded symmetric, columns graded antisymmetric and all traces are removed. For example the spin $\ell$ graded symmetric and traceless representation is labelled by a single row with $\ell$ boxes. We can write a spin $\ell$ operator in components as follows[2]

$$\mathcal{O}^{a_1 \cdots a_\ell}(y) = \mathcal{O}_0^{a_1 \cdots a_\ell}(x) + \theta \mathcal{O}_{\bar{\theta}}^{a_1 \cdots a_\ell}(x) + \bar{\theta} \mathcal{O}_{\theta}^{a_1 \cdots a_\ell}(x) + \theta \bar{\theta} \mathcal{O}_{\theta\bar{\theta}}^{a_1 \cdots a_\ell}(x). \tag{20}$$

When $\ell = 0$ the components $\mathcal{O}_0$ and $\mathcal{O}_{\theta\bar{\theta}}$ are bosonic while $\mathcal{O}_\theta$ and $\mathcal{O}_{\bar{\theta}}$ are fermionic. As a convention, throughout the paper we will use upper indices of operators to denote graded tensor indices, while the subscript of the operator will be reserved to denote the components. In terms of the usual dilation generator, the component operators in (20) have dimensions $\Delta$, $\Delta + 1$, $\Delta + 2$ where each theta in the expansion is responsible for a one unit shift in the conformal dimensions. Since $\theta$ and $\bar{\theta}$ do not carry indices, all components of $\mathcal{O}$ transform in

---

[2]The following notation slightly differs from the one of [7] in order to make R-symmetry properties of the operators more readable.

the same $OSp(d|2)$ representation. However every $OSp(d|2)$ representation can be decomposed in $SO(d)$ ones, so each component of (20) can be further decomposed in four different operators with $SO(d)$-spin equal to $\ell, \ell-1, \ell-2$, e.g.

$$\mathcal{O}_0^{\alpha_1...\alpha_\ell}(x), \qquad \mathcal{O}_0^{\alpha_1...\alpha_{\ell-1}\theta}(x), \qquad \mathcal{O}_0^{\alpha_1...\alpha_{\ell-1}\bar{\theta}}(x), \qquad \mathcal{O}_0^{\alpha_1...\alpha_{\ell-2}\theta\bar{\theta}}(x). \qquad (21)$$

Every time an index takes the value $a = \theta, \bar{\theta}$, then the operator changes grading. E.g. $\mathcal{O}_0^{\alpha_1...\alpha_\ell}$ and $\mathcal{O}_0^{\alpha_1...\alpha_{\ell-2}\theta\bar{\theta}}$ are bosonic while $\mathcal{O}_0^{\alpha_1...\alpha_{\ell-1}\theta}$ and $\mathcal{O}_0^{\alpha_1...\alpha_{\ell-1}\bar{\theta}}$ are fermionic. It is also worth commenting on the R-symmetry of the model, which is $Sp(2)$ and it is generated by $M^{\theta\theta}$, $M^{\bar{\theta}\bar{\theta}}$ and $M^{\theta\bar{\theta}} = M^{\bar{\theta}\theta}$. It is easy to see that, given a scalar $\mathcal{O}$, the component $\mathcal{O}_{\bar{\theta}}$ has charge $-1$ under $M^{\theta\bar{\theta}}$, while $\mathcal{O}_\theta$ has charge $+1$. The bosonic components are charged zero. Similarly by taking indices in the direction $\theta, \bar{\theta}$ we select operators with different R-symmetry charge. E.g. given a vector $\mathcal{O}^a$, $\mathcal{O}_0^\theta$ has charge $+1$, $\mathcal{O}_\theta^\theta$ has charge $+2$, while $\mathcal{O}_{\bar{\theta}}^\theta$ has charge 0. As usual, R-symmetry is a global symmetry, thus all non-vanishing correlation functions must be invariant under $Sp(2)$.

Since the theory has superconformal symmetry, it is convenient to classify the operators as superconformal primaries and descendants. The super primaries are operators $\mathcal{O}(0)$ defined such that they are annihilated by the special superconformal generators, $[K^a, \mathcal{O}(0)] = 0$. Superdescendants are instead obtained by acting with derivatives $\partial_a$ on the superprimaries.

Correlation functions of local operators are restricted by superconformal symmetry. One-point functions take the form $\langle \mathcal{O}(y) \rangle = \delta_{\mathcal{O}\mathbb{I}}$, while two- and three-point functions are fixed as

$$\langle \mathcal{O}(y_1)\mathcal{O}(y_2) \rangle = \frac{1}{(y_{12}^2)^\Delta}, \qquad (22)$$

$$\langle \mathcal{O}_1(y_1)\mathcal{O}_2(y_2)\mathcal{O}_3(y_3) \rangle = \frac{\lambda_{123}}{(y_{12}^2)^{\frac{\Delta_1+\Delta_2-\Delta_3}{2}}(y_{13}^2)^{\frac{\Delta_1+\Delta_3-\Delta_2}{2}}(y_{23}^2)^{\frac{\Delta_2+\Delta_3-\Delta_1}{2}}}, \qquad (23)$$

where $y_{ij}^a \equiv y_i^a - y_j^a$. Four-point functions can be written as

$$\langle \mathcal{O}_1(y_1)\mathcal{O}_2(y_2)\mathcal{O}_3(y_3)\mathcal{O}_4(y_4) \rangle = K_{\Delta_i}(y_i)f(U,V), \qquad (24)$$

where the superspace cross ratios are defined as

$$U \equiv \frac{y_{12}^2 y_{34}^2}{y_{13}^2 y_{24}^2}, \qquad V \equiv \frac{y_{14}^2 y_{23}^2}{y_{13}^2 y_{24}^2}, \qquad (25)$$

and $K_{\Delta_i}$ is the same as in (5), where we replace $x$ by $y$, namely

$$K_{\Delta_i}(y_i) \equiv \frac{1}{(y_{12}^2)^{\frac{1}{2}(\Delta_1+\Delta_2)}(y_{34}^2)^{\frac{1}{2}(\Delta_3+\Delta_4)}} \left( \frac{y_{14}^2}{y_{24}^2} \right)^{-\frac{\Delta_{12}}{2}} \left( \frac{y_{14}^2}{y_{13}^2} \right)^{\frac{\Delta_{34}}{2}}. \qquad (26)$$

One can also introduce other cross ratios like the ones of (6), e.g. $U = Z\bar{Z}, V = (1-Z)(1-\bar{Z})$ and similarly for $\rho$ and $\bar{\rho}$. The OPE in superspace takes the form

$$\mathcal{O}_1(y)\mathcal{O}_2(0) = \sum_{\Delta\ell} \lambda_{12\mathcal{O}} \left[ \frac{y_{a_1}\cdots y_{a_\ell}}{|y|^{\Delta_1+\Delta_2-\Delta+\ell}} \mathcal{O}^{a_1...a_\ell}(x) + \dots \right], \qquad (27)$$

where because of the graded commuting properties of $y^a$, the spin $\ell$ operators $\mathcal{O}^{a_1...a_\ell}$ transform in the graded symmetric and traceless representation of $OSp(d|2)$. The dots contain

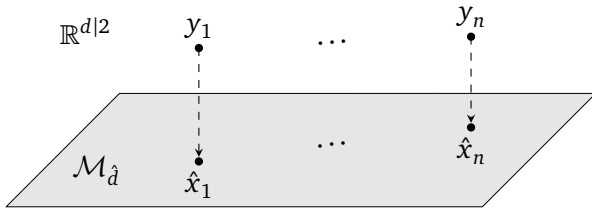

Figure 3: Dimensional reduction: All points $y_i \in \mathbb{R}^{d|2}$ are projected to $\mathcal{M}_{\hat{d}}$.

contributions from the superdescendants. We can apply the OPE in the four-point function and get

$$f(U,V) = \sum_{\Delta \ell} a_{\Delta \ell} \, G^{(d|2)}_{\Delta \ell}(U,V),$$ (28)

where $a_{\Delta \ell} \equiv \lambda_{12O} \lambda_{O34}$ and $G^{(d|2)}_{\Delta \ell}$ is the superspace superconformal block, which is equal to a usual conformal blocks $g^{(d-2)}_{\Delta \ell}$ in two less dimensions [7]

$$G^{(d|2)}_{\Delta \ell} = g^{(d-2)}_{\Delta \ell}.$$ (29)

The equality between superspace superblocks and lower-dimensional blocks was proven in [7] to hold also for correlators of generic tensor operators.

## 2.4 Dimensional reduction

Any given PS CFT can be dimensionally reduced to a CFT in two less dimension $\hat{d} \equiv d - 2$. Let us explain how this works. The main idea is that, given a correlator in superspace, if one restricts all the insertions to $\mathcal{M}_{\hat{d}} \equiv \{y \in \mathbb{R}^{d|2} : y = (\hat{x}, 0, 0, 0, 0), \hat{x} \in \mathbb{R}^{\hat{d}}\}$ as in figure 3, then one defines the dimensionally reduced theory. Namely all reduced correlators are obtained as

$$\langle \mathcal{O}_1(y_1) \dots \mathcal{O}_n(y_n) \rangle|_{\mathcal{M}_{\hat{d}}} = \langle \hat{O}_1(\hat{x}_1) \dots \hat{O}_n(\hat{x}_n) \rangle,$$ (30)

where $\hat{O}_i$ are operators of a $\text{CFT}_{\hat{d}}$. Here we considered $\mathcal{O}_i$ and $\hat{O}_i$ to be scalars, but the generalization to operators with spin is straightforward (see comments below and in [7]). It is well known that any set of correlators which arises from any type of reduction automatically satisfies the axioms of crossing and conformal block decomposition as a usual $\text{CFT}_{\hat{d}}$. What is special to this kind of reduction is that the resulting $\text{CFT}_{\hat{d}}$ can be shown to be local[3] and with a very small operator content, which is smaller (or equal) than the superprimary content of the original PS $\text{CFT}_d$ [7]. Both properties are very atypical since: 1) the stress tensor is not expected to be conserved on $\mathcal{M}_{\hat{d}}$ since in principle the energy is allowed to leak orthogonally to $\mathcal{M}_{\hat{d}}$ and 2) usually for every operator in a higher dimensional theory we expect infinitely many operators in the dimensionally reduced theory, and therefore a much larger spectrum. These remarkable properties are due to a decoupling of infinitely many operators because of SUSY. Let us explain how this decoupling arises.

Restricting operator insertions to a hyperplane in the CFT literature is often called "trivial defect". In our case the trivial defect breaks the $OSp(d + 1, 1|2)$ symmetry to $SO(d - 1, 1) \times OSp(2|2)$ where $SO(d - 1, 1)$ is understood as the conformal group in $\hat{d}$ dimensions and $OSp(2|2)$ is a global symmetry. The prescription to get dimensional reduction

---

[3]The precise statement is that, whenever the PS CFT contains a conserved super stress tensor, then the reduced theory contains a conserved stress tensor (as long as $\hat{d} > 1$). When $\hat{d} = 1$, as expected, the reduced stress tensor does not exist as we shall discuss in section 7.

(30) focuses on the singlet sector of the $OSp(2|2)$ symmetry. For any given primary operator $\mathcal{O}$ there is an infinite set of $OSp(2|2)$ singlets in the trivial defect. Indeed, in addition to the $OSp(2|2)$-singlet primary $\mathcal{O}(\hat{x})$ considered in (30), one may also consider $OSp(2|2)$-singlet primary operators obtained as $(\partial_{\perp}^2)^n \mathcal{O}(\hat{x})$, where $\partial_{\perp}$ is the derivative in the direction perpendicular to the hyperplane $\mathcal{M}_{\hat{d}}$, namely $\partial_{\perp}^a = g_{2|2}^{ab} \partial_b$. The statement is that all these operators decouple from the theory if they are inserted in correlators of only $OSp(2|2)$-singlets. This can be easily understood, since the $g_{2|2}$ metric orthogonal to the plane has two free indices that cannot be contracted with anything to build an $OSp(2|2)$-singlet. Indeed

$$g_{2|2}^{ab} \hat{x}_{ib} = 0, \qquad g_{2|2}^{ab} g_{2|2\,ba} = 0, \tag{31}$$

the former relation due to the orthogonality of the vectors $\hat{x}_i$ and the perpendicular metric, while the latter due to the properties of the supertrace of the orthosymplectic metric which satisfies $\operatorname{str} g_{d|2} = d - 2$. A similar decoupling works at the level of operators with spin. In this case to define operators on the trivial defect we must project the tensor indices to directions either orthogonal or parallel to $\mathcal{M}_{\hat{d}}$. The only $OSp(2|2)$-singlets which do not decouple have all indices along the trivial defect $\mathcal{M}_{\hat{d}}$. So in practice given a superfield $\mathcal{O}_Y$ that transforms in a representation of $OSp(d|2)$ labelled by a Young tableau $Y$ as in (19), the reduced field transforms in a representation of $SO(\hat{d})$ labelled by the same Young tableau $Y$. It can happen that the dimension of the representation $Y$ in $SO(\hat{d})$ is zero. When this happens, the dimensionally reduced field vanishes, so the superfield $\mathcal{O}_Y$ is projected to zero upon dimensional reduction. We will expand on this point in section 7.

## 2.5 Reduction and uplift of Lagrangians

There are explicit realizations of PS QFTs. An important class is defined through the following scalar action in superspace [1],

$$S_{d|2} = \int d^d x\, d\theta\, d\bar{\theta}\, \frac{1}{2} \partial^a \Phi(y) \partial_a \Phi(y) + V(\Phi(y)), \tag{32}$$

where $V$ is a given polynomial potential and the scalar superfield $\Phi$ can be decomposed in components as

$$\Phi(x, \theta, \bar{\theta}) = \varphi(x) + \theta \bar{\psi}(x) + \bar{\theta} \psi(x) + \theta \bar{\theta} \omega(x). \tag{33}$$

We can integrate out the Grassmann variables obtaining the action

$$S_{d|2} = \int d^d x\, \partial^\mu \varphi \partial_\mu \omega - \omega^2 + \partial^\mu \psi \partial_\mu \bar{\psi} + \omega V'(\varphi) + \psi \bar{\psi} V'(\varphi), \tag{34}$$

where we notice that the classical dimensions of the fields is

$$[\varphi] = \frac{d}{2} - 2, \qquad [\psi] = [\bar{\psi}] = \frac{d}{2} - 1, \qquad [\omega] = \frac{d}{2}. \tag{35}$$

This action can be dimensionally reduced to [1]

$$S_{\hat{d}} = \int d^{\hat{d}} \hat{x}\, \frac{1}{2} (\partial_\mu \hat{\phi}(\hat{x}))^2 + V(\hat{\phi}(\hat{x})). \tag{36}$$

In this case the dimensional reduction is more powerful than in (30). Indeed here we know that correlators on the *lhs* of (30) which are computed using the action (32) can be equivalently calculated by the prescription in the *rhs* defined using the action (36), e.g.

$$\langle \Phi(y_1) \dots \Phi(y_n) \rangle_{S_{d|2}} \Big|_{\mathcal{M}_{\hat{d}}} = \langle \hat{\phi}(\hat{x}_1) \dots \hat{\phi}(\hat{x}_n) \rangle_{S_{\hat{d}}}. \tag{37}$$

Therefore there exist two separate computations which give the same result. Because of this we can conclude that when a PS SUSY Lagrangian is available, not only dimensional reduction is available, but also the uplift. Indeed the equivalence (37) can be also read backwards, implying that every scalar action of the form (36) can be uplifted to (32) which can be used to compute the same observables. The prescription to uplift Lagrangians is thus trivial and it amounts to uplifting fields to superfields, and derivatives to superderivatives,

$$\hat{\phi}(\hat{x}) \to \Phi(y), \qquad \partial_\mu \to \partial_a. \tag{38}$$

This can be further generalized to any number or scalar fields and also to theories where the fundamental fields are not scalars, see for example the realizations for spinors and gauge fields presented in [28].

## 2.6 Uplift of CFTs

While we showed that given a Lagrangian, its uplift is somewhat straightforward, this result does not help much in defining the uplift of CFTs which typically do not have useful Lagrangian descriptions. However in CFT we can use the power of symmetry to uplift some observables for free. This is indeed a basic idea in CFT where it is known that one can use conformal symmetry to set the operator insertions to special locations. E.g. by knowing two and three-point functions of scalar operators on a line one can recover their full dependence on coordinates, and similarly four-point functions of scalar operators are completely determined by their expression on a plane. The same idea works for PS CFTs. For these theories the dimensionally reduced two and three-point functions of scalar operators determine the uplifted ones as long as $\hat{d} \geq 1$, while for four-point functions of scalar operators the uplifted correlator is fully reconstruct if $\hat{d} \geq 2$. The reconstruction in these cases is trivial and it amounts to replace

$$\hat{x} \to y, \tag{39}$$

in correlation functions. For example a two-point function $\langle \hat{O}_1(\hat{x}_1) \hat{O}_2(\hat{x}_2) \rangle = |\hat{x}_{12}|^{-2\Delta}$ is simply uplifted to $\langle \mathcal{O}_1(y_1) \mathcal{O}_2(y_2) \rangle = |y_{12}|^{-2\Delta}$, where now the distance is computed using the $g_{d|2}$ metric instead of a $g_{d-2}$ metric. Similarly for any scalar four-point function $\langle \hat{O}_1(\hat{x}_1) \hat{O}_2(\hat{x}_2) \hat{O}_3(\hat{x}_3) \hat{O}_4(\hat{x}_4) \rangle = F(\hat{x}_i \cdot \hat{x}_j)$, as long as $\hat{d} \geq 2$, then its uplift is just given by $F(y_i \cdot y_j)$ where all distances are replaced by superspace distances.

Of course this prescription cannot be used to fix all possible observables. In general for any fixed dimension $\hat{d}$, when the number of insertion is high enough and/or the operators have spin large enough, then a piece of the information of the correlation function cannot be reconstructed. One can easily see when this happens for any given observable by counting the number of cross ratios and invariant tensor structures [29]. E.g. in $\hat{d} = 1$ a four-point function is fixed in terms of a single cross ratio and therefore one cannot simply reconstruct a full PS CFT four-point function in $d = 3$, but only its restriction to a line (we expand on this point in section 7). This is also related to the fact that some operators of the PS CFT are projected to zero after dimensional reduction. This is easy to see, indeed in the PS CFT there can exist operators with an arbitrary number of graded antisymmetric indices [7], but in $\hat{d}$ dimensions one cannot antisymmetrize more that $\hat{d}$ indices otherwise the result is trivially zero. This will be further discussed in section 7.

In this paper we will mostly focus on scalar correlators of $n \leq 4$ insertions in dimensions $\hat{d} \geq 2$ so for these cases the uplift will trivially work. It is interesting to stress that this is a huge number of observables which know a lot about the theory. Indeed we are saying that all possible four-point functions of scalar operators will be exactly uplifted. The latter contain in their conformal block decomposition a lot of information about operators with spin. It is an interesting open question whether the information contained in this infinite set of correlators

is enough to fix the rest of the observables by e.g. applying bootstrap techniques. We leave this question for future investigations.

# 3 Extracting primary components of correlation functions in superspace

One of the results of this paper is to put on a firm footing the uplift of scalar $n$-point functions for $n \leq 4$. For the uplift to make sense all components of the superspace correlator must behave appropriately as correlators of a $CFT_d$. In the following we show how to obtain the different primary components from the superspace correlators for the cases of two, three and four-point functions. We explicitly show the form of all components and that they indeed have the correct covariance properties of correlators of primary operators. Moreover we will show that all the primary components can be fully reconstructed by the knowledge of the lowest component, which is fixed by the dimensionally reduced theory. This result will be important for the rest of the paper where we will input a four-point function of a $CFT_{\hat{d}}$ to obtain all the primary components of the uplifted four-point functions. Since this philosophy works for any $CFT_{\hat{d}}$ (at least when $\hat{d} > 1$) this result also supports the conjecture that the dimensional uplift works for any theory.

Given a scalar superprimary $\mathcal{O}(y_i)$ with dimension $\Delta_i$ inserted at a point $y_i$ in superspace, the correspondent primaries are defined as follows[4]

$$\mathcal{O}(y_i)|_0 = \mathcal{O}_0(x_i), \tag{40}$$

$$\partial_{\theta_i}\mathcal{O}(y_i)|_0 = \mathcal{O}_{\bar{\theta}}(x_i), \tag{41}$$

$$\partial_{\bar{\theta}_i}\mathcal{O}(y_i)|_0 = \mathcal{O}_{\theta}(x_i), \tag{42}$$

$$\mathbf{D}_{[\mathbf{i}]}\mathcal{O}(y_i)|_0 = \tilde{\mathcal{O}}_{\theta\bar{\theta}}(x_i), \tag{43}$$

where throughout the paper the notation $|_0$ will mean that we set to zero all $\theta_i$ and $\bar{\theta}_i$. The differential operator in (43) is defined as

$$\mathbf{D}_{[\mathbf{i}]} \equiv \partial_{x_i}^2 - (2\Delta_i - d + 2)\partial_{\bar{\theta}_i}\partial_{\theta_i}. \tag{44}$$

Notice that, while $\mathcal{O}_0, \mathcal{O}_\theta, \mathcal{O}_{\bar{\theta}}$ are the same as the components defined in (20), the tilded operator $\tilde{\mathcal{O}}_{\theta\bar{\theta}}$ differs from the component $\mathcal{O}_{\theta\bar{\theta}}$. The two operators are related since $\tilde{\mathcal{O}}_{\theta\bar{\theta}}$ is a linear combination of $\mathcal{O}_{\theta\bar{\theta}}$ with a descendant of $\mathcal{O}_0$. In practice we can think of $\tilde{\mathcal{O}}_{\theta\bar{\theta}}$ as an improved version of $\mathcal{O}_{\theta\bar{\theta}}$ which is primary; indeed $\mathcal{O}_{\theta\bar{\theta}}$ by itself is not a primary operator.

Correlation functions of a PS $CFT_d$ can be written in terms of superspace distances $y_{ij}^2$. By opportunely acting with the operators $\partial_{\theta_i}, \partial_{\bar{\theta}_i}, \mathbf{D}_{[\mathbf{i}]}$ on superspace correlators we can extract the correlators of $CFT_d$ primaries. The generic form is as follows

$$\mathbf{D}_{[\vec{s}]}\langle \mathcal{O}_1(y_1)\ldots\mathcal{O}_n(y_n)\rangle|_0, \tag{45}$$

where $\mathbf{D}_{[\vec{s}]}$ is a differential operator in $x_i, \theta_i, \bar{\theta}_i$ defined as follows

$$\mathbf{D}_{[\vec{s}]} \equiv \mathbf{D}_{[s_1\ldots s_m]} \equiv \mathbf{D}_{[s_1]}\cdots\mathbf{D}_{[s_m]}, \tag{46}$$

where each $S_k$ can be either $\mathbf{i}$ or $i\bar{j}$ and we define

$$\mathbf{D}_{[i\bar{j}]} \equiv \partial_{\bar{\theta}_j}\partial_{\theta_i} \qquad (i \neq j). \tag{47}$$

---

[4]One could change the normalization of the primaries and require that their two-point function is unit normalized. We did not do it here since we privileged having a simpler form for the differential operators. Moreover in our normalization $\Delta$ never appears at the denominator thus avoiding singularities in the differential operators at special values of $\Delta$.

We find it convenient to use $\mathbf{D}_{[i\bar{j}]}$ instead of the building blocks $\partial_{\theta_i}, \partial_{\bar{\theta}_j}$, because all observables must be $Sp(2)$ invariants, so if $\partial_{\theta_i}$ is used then we must also use $\partial_{\bar{\theta}_j}$ for some $j$. As an example the operators will be of the form $\mathbf{D}_{[1\bar{2}3\bar{4}]} = \mathbf{D}_{[1\bar{2}]}\mathbf{D}_{[3\bar{4}]}$ (of course this could be equivalently written in terms of $\mathbf{D}_{[1\bar{4}]}\mathbf{D}_{[3\bar{2}]}$), $\mathbf{D}_{[\mathbf{124}]} = \mathbf{D}_{[\mathbf{1}]}\mathbf{D}_{[\mathbf{2}]}\mathbf{D}_{[\mathbf{4}]}$, $\mathbf{D}_{[\mathbf{21}\bar{4}]} = \mathbf{D}_{[\mathbf{2}]}\mathbf{D}_{[1\bar{4}]}$.

It will also be useful to introduce some shift operators $\Sigma_{[\vec{S}]}$, which take into account that the dimension of the operator $\Delta_i$ is shifted by one unit if $\mathbf{D}_{[\vec{S}]}$ contains $\partial_{\bar{\theta}_i}$ or $\partial_{\theta_i}$ and by two units if $\mathbf{D}_{[\vec{S}]}$ contains $\partial_{x_i}^2 - (2\Delta_i - d + 2)\partial_{\bar{\theta}_i}\partial_{\theta_i}$. To be precise

$$\Sigma_{[\vec{S}]} \equiv \Sigma_{[S_1 \ldots S_m]} \equiv \Sigma_{[S_1]} \cdots \Sigma_{[S_m]}, \tag{48}$$

where $\Sigma_{[S_i]}$ implement the following shifts

$$\Sigma_{[\mathbf{i}]}g(\Delta_k) = g(\Delta_k + 2\delta_{ik}), \qquad \Sigma_{[i\bar{j}]}g(\Delta_k) = g(\Delta_k + \delta_{ik} + \delta_{jk}), \tag{49}$$

for any function of $g$ that depends on the conformal dimensions $\Delta_k$, where $k = 1, \ldots, n$ in formula (45).

## 3.1 Two-point functions

As we explained in section 2.3, the scalar two-point function in superspace is fixed as (22). Using the differential operators defined above, we can easily compute all the associated primary components as follows

$$\langle \mathcal{O}_0(x_1)\mathcal{O}_0(x_2) \rangle \equiv \langle \mathcal{O}(y_1)\mathcal{O}(y_2) \rangle|_0 = \frac{1}{(x_{12}^2)^{\Delta}}, \tag{50}$$

$$\langle \mathcal{O}_\theta(x_2)\mathcal{O}_{\bar{\theta}}(x_1) \rangle \equiv \mathbf{D}_{[1\bar{2}]}\langle \mathcal{O}(y_1)\mathcal{O}(y_2) \rangle|_0 = \frac{2\Delta}{(x_{12}^2)^{\Delta+1}}, \tag{51}$$

$$\langle \tilde{\mathcal{O}}_{\theta\bar{\theta}}(x_1)\tilde{\mathcal{O}}_{\theta\bar{\theta}}(x_2) \rangle \equiv \mathbf{D}_{[\mathbf{12}]}\langle \mathcal{O}(y_1)\mathcal{O}(y_2) \rangle|_0 = \frac{-4\Delta(\Delta+1)(-d+2\Delta+2)(-d+2\Delta+4)}{(x_{12}^2)^{\Delta+2}}. \tag{52}$$

We notice that all these two-point functions have the correct properties to be two-point functions of primary operators. Moreover we further checked that the two-point functions of different primaries are diagonal, namely $\langle \tilde{\mathcal{O}}_{\theta\bar{\theta}}\mathcal{O}_0 \rangle = 0$. Indeed one of the ways to fix the differential operator $\mathbf{D}_{[\mathbf{i}]}$ is to require that the latter is true. We notice that the normalization of the two-point functions above may be zero. This happens when $\Delta = 0$, which is the trivial case of the identity, but also at the free field dimension $\Delta = \frac{d-2}{2}$ and at $\Delta = \frac{d-4}{2}$ which is the dimension of a free field in the dimensionally reduced theory. In section (6) we will see that indeed a field with dimensions $\Delta = 1$ in $d = 4$ will appear, in that circumstance we will explain how the vanishing norm (52) will have important implications.

## 3.2 Three-point functions

We now focus on three-point functions of scalar operators, which in superspace are defined as (23). This observable contains 14 inequivalent three-point functions of primary operators corresponding to the different primary components of the supermultiplets. To capture the three-point functions of all components at once we use the following compact formula

$$\mathbf{D}_{[\vec{S}]}\langle \mathcal{O}_1(y_1)\mathcal{O}_2(y_2)\mathcal{O}_3(y_3) \rangle\Big|_0 = c_{[\vec{S}]}\lambda_{123}\Sigma_{[\vec{S}]}\frac{1}{(x_{12}^2)^{\frac{\Delta_1+\Delta_2-\Delta_3}{2}}(x_{13}^2)^{\frac{\Delta_1+\Delta_3-\Delta_2}{2}}(x_{23}^2)^{\frac{\Delta_2+\Delta_3-\Delta_1}{2}}}, \tag{53}$$

where $\mathbf{D}_{[\vec{S}]}$ are the differential operators defined in (46) and $\Sigma_{[\vec{S}]}$ implement the shifts in $\Delta_i$ according to (48). In this notation we only need to define what the coefficients $c_{[\vec{S}]}$ are.

They encode the relation between the OPE coefficient of the superprimary and the ones of its superdescendants. When $\vec{S}$ is the empty list, which corresponds to the lowest component, the coefficient equals one, namely $c_{[\ ]} = 1$. All other cases are below

$$c_{[\mathbf{1}]} = -\alpha_{13,2}\alpha_{12,3},$$
$$c_{[\mathbf{1\bar{2}}]} = -\alpha_{12,3},$$
$$c_{[\mathbf{12\bar{3}}]} = \alpha_{13,2}\alpha_{12,3}(\beta - d + 2),$$
$$c_{[\mathbf{3}]} = -\alpha_{31,2}\alpha_{32,1},$$
$$c_{[\mathbf{3\bar{1}}]} = -\alpha_{31,2},$$

$$c_{[\mathbf{2}]} = -\alpha_{23,1}\alpha_{21,3},$$
$$c_{[\mathbf{2\bar{3}}]} = -\alpha_{23,1},$$
$$c_{[\mathbf{23\bar{1}}]} = \alpha_{23,1}\alpha_{21,3}(\beta - d + 2),$$

$$c_{[\mathbf{31\bar{2}}]} = \alpha_{31,2}\alpha_{32,1}(\beta - d + 2),$$
$$c_{[\mathbf{12}]} = \alpha_{12,3}(\alpha_{12,3} + 2)(\beta - d + 2)(d - 4 - \alpha_{12,3}),$$
$$c_{[\mathbf{13}]} = \alpha_{13,2}(\alpha_{13,2} + 2)(\beta - d + 2)(d - 4 - \alpha_{13,2}),$$
$$c_{[\mathbf{23}]} = \alpha_{23,1}(\alpha_{23,1} + 2)(\beta - d + 2)(d - 4 - \alpha_{23,1}),$$
$$c_{[\mathbf{123}]} = \alpha_{23,1}\alpha_{12,3}\alpha_{13,2}(\beta - 2d + 6)(\beta - d + 2)(\beta - d + 4),$$

$$(54)$$

where $\alpha_{ij,k} \equiv \Delta_i + \Delta_j - \Delta_k$ and $\beta \equiv \Delta_1 + \Delta_2 + \Delta_3$. For the sake of clarity we show a couple of three-point functions in a more explicit form,

$$\langle \mathcal{O}_{0\,1}(x_1)\mathcal{O}_{0\,2}(x_2)\tilde{\mathcal{O}}_{\theta\bar{\theta}\,3}(x_3)\rangle = \frac{\lambda_{123}(\Delta_1 - \Delta_2 - \Delta_3)(\Delta_1 - \Delta_2 + \Delta_3)}{(x_{12}^2)^{\frac{\Delta_1+\Delta_2-\Delta_3-2}{2}}(x_{13}^2)^{\frac{\Delta_1+\Delta_3+2-\Delta_2}{2}}(x_{23}^2)^{\frac{\Delta_2+\Delta_3+2-\Delta_1}{2}}}, \quad (55)$$

$$\langle \mathcal{O}_{0\,1}(x_1)\mathcal{O}_{\bar{\theta}\,2}(x_2)\mathcal{O}_{\theta\,3}(x_3)\rangle = \frac{\lambda_{123}(\Delta_2 + \Delta_3 - \Delta_1)}{(x_{12}^2)^{\frac{\Delta_1+\Delta_2-\Delta_3}{2}}(x_{13}^2)^{\frac{\Delta_1+\Delta_3-\Delta_2}{2}}(x_{23}^2)^{\frac{\Delta_2+\Delta_3-\Delta_1+2}{2}}}. \quad (56)$$

As a comment, let us mention that it is a non-trivial consistency check that in all 14 cases these take the form of three-point functions of primary operators. The shifts $\Sigma_{[\vec{S}]}$ keep track of the correct scaling of the operators and thus the theory behaves correctly as a $\mathrm{CFT}_d$. The OPE coefficients of the (primary) superdescendants are just related to the ones of the superprimary by some coefficients $c_{[\vec{S}]}$ which are fixed by SUSY. Therefore by knowing the lowest component one can reconstruct all the other ones.

## 3.3 Four-point functions

Scalar four-point functions in superspace are written as (24), in terms of a single function $f(U,V)$ of superspace cross ratios $U, V$. Of course since the theory is a CFT, all components of this superspace four-point function should be written in terms of usual $d$-dimensional cross ratios $u, v$. The lowest component is just $f(u, v)$. The other components are obtained by acting with a differential operator in $u, v$ that acts on $f(u, v)$. The latter can be computed by acting with some differential operators on a four-point function, as follows

$$\mathbf{D}_{[\vec{S}]}K_{\Delta_i}(y_i)f(U,V)\big|_{\theta,\bar{\theta}=0} = (\Sigma_{[\vec{S}]}K_{\Delta_i}(x_i))D_{[\vec{S}]}f(u,v), \quad (57)$$

where $\mathbf{D}_{[\vec{S}]}$ and $\Sigma_{[\vec{S}]}$ are defined in (46) and (48) while $D_{[\vec{S}]}$ is a differential operator in $u, v$ which we computed for all 43 components. Some cases are too lengthy to be shown in a paper,[5] but they will be collected in a Mathematica file included with the publication. Below we show

---

[5]Indeed the maximal order of the differential operator in $\partial_u$ and $\partial_v$ is eight, and the coefficients are polynomials of the seven variables $d, \Delta_1, \Delta_2, \Delta_3, \Delta_4, u, v$ Because of this, the resulting expressions are sometimes several pages long.

their explicit form in a few instances. First the lowest component is the trivial differential operator $D_{[\ ]} = 1$. Let us now focus on the operators $D_{[\mathbf{i}]}$ for $i = 1, 2, 3, 4$,

$$
\begin{aligned}
D_{[\mathbf{1}]} \equiv{}& \left[-\Delta_{12}^2 + \Delta_{34}^2 u - \Delta_{34}\Delta_{12}(u+v-1) - 2\Delta_2\left(\Delta_{12} + \Delta_{34}(v-1)\right)\right] \\
&- 2u\left(\Delta_{12}(u+v-1) - \Delta_{34}(u+v-1) + 2\Delta_2 v - 2v\right)\partial_u \\
&- 2v\left(-2\left(\Delta_{34}+1\right)u + \Delta_{12}(u+v-1) + 2\Delta_2(v-1)\right)\partial_v \\
&+ 4uv(u+v-1)\partial_u\partial_v + 4u^2 v\partial_u^2 + 4uv^2\partial_v^2,
\end{aligned}
\tag{58}
$$

$$
\begin{aligned}
D_{[\mathbf{2}]} \equiv{}& \left(\Delta_1^2 - \Delta_2^2\right) - 4\left(\Delta_1 - 1\right)u\partial_u + \left(2\left(-\Delta_1 + \Delta_2 + 2\right)u - 2\left(\Delta_1 + \Delta_2\right)(v-1)\right)\partial_v \\
&+ 4u^2\partial_u^2 + 4uv\partial_v^2 + 4u(u+v-1)\partial_u\partial_v,
\end{aligned}
\tag{59}
$$

$$
\begin{aligned}
D_{[\mathbf{3}]} \equiv{}& \left(\Delta_4^2 - \Delta_3^2\right) - 4\left(\Delta_4 - 1\right)u\partial_u + \left(2\Delta_3(u-v+1) - 2\Delta_4(u+v-1) + 4u\right)\partial_v \\
&+ 4u^2\partial_u^2 + 4uv\partial_v^2 + 4u(u+v-1)\partial_u\partial_v,
\end{aligned}
\tag{60}
$$

$$
\begin{aligned}
D_{[\mathbf{4}]} \equiv{}& \left[\Delta_{12}\left(\Delta_{12} - \Delta_{34}\right)u + \left(\Delta_4 + \Delta_3\right)\left(\Delta_{34} + \Delta_{12}(v-1)\right)\right] + 4uv(u+v-1)\partial_u\partial_v \\
&- 2u\left(\Delta_{12}(u+v-1) + \Delta_{34}(-u+v+1) + 2\Delta_4 v - 2v\right)\partial_u + 4u^2 v\partial_u^2 \\
&+ 2v\left(-2\Delta_{12}u + \Delta_{34}(u-v+1) + 2u - 2\Delta_4(v-1)\right)\partial_v + 4uv^2\partial_v^2.
\end{aligned}
\tag{61}
$$

We further exemplify the form of $D_{[i\bar{j}]}$ for $1 \le i < j \le 4$,

$$
D_{[1\bar{2}]} \equiv \left(-\Delta_1 - \Delta_2\right) + 2u\partial_u,
\tag{62}
$$

$$
D_{[3\bar{4}]} \equiv \left(-\Delta_3 - \Delta_4\right) + 2u\partial_u,
\tag{63}
$$

$$
D_{[1\bar{3}]} \equiv -2u^{3/2}\partial_u - \Delta_{34}\sqrt{u} - 2\sqrt{u}v\partial_v,
\tag{64}
$$

$$
D_{[2\bar{4}]} \equiv -2u^{3/2}\partial_u + \Delta_{12}\sqrt{u} - 2\sqrt{u}v\partial_v,
\tag{65}
$$

$$
D_{[1\bar{4}]} \equiv \left(-\Delta_{12} + \Delta_{34}\right)\sqrt{u} + 2\sqrt{u}v\partial_v,
\tag{66}
$$

$$
D_{[2\bar{3}]} \equiv 2\sqrt{u}\partial_v.
\tag{67}
$$

It is easy to see that $D_{[i\bar{j}]} = -D_{[\bar{i}j]}$, so these components are redundant and are not taken into account in the counting. Let us also show the cases of four fermionic components. In this case there are only three independent operators given by

$$
\begin{aligned}
D_{[1\bar{2}3\bar{4}]} \equiv{}& \left(\Delta_1 + \Delta_2\right)\left(\Delta_3 + \Delta_4\right) + 4u^2\partial_u^2 + 2\left(\Delta_{12} - \Delta_{34} - 2\right)u\partial_v \\
&- 2\left(\Delta_1 + \Delta_2 + \Delta_3 + \Delta_4 - 2\right)u\partial_u - 4uv\partial_v^2,
\end{aligned}
\tag{68}
$$

$$
\begin{aligned}
D_{[1\bar{3}2\bar{4}]} \equiv{}& 2\left(-\Delta_{12} + \Delta_{34} + 2\right)u^2\partial_u + 4u^3\partial_u^2 + 8u^2 v\partial_u\partial_v - \Delta_{12}\Delta_{34}u \\
&- 2\left(\Delta_{12} - \Delta_{34} - 2\right)u(v-1)\partial_v + 4u(v-1)v\partial_v^2,
\end{aligned}
\tag{69}
$$

$$
\begin{aligned}
D_{[1\bar{2}4\bar{3}]} \equiv{}& \left(\left(\Delta_1 + \Delta_2\right)\left(\Delta_3 + \Delta_4\right) + \Delta_{12}\Delta_{34}u\right) + 2\left(\Delta_{12} - \Delta_{34} - 2\right)uv\partial_v \\
&+ 2\left(\left(\Delta_{12} - \Delta_{34} - 2\right)u - \left(\Delta_1 + \Delta_2 + \Delta_3 + \Delta_4 - 2\right)\right)u\partial_u \\
&- 8u^2 v\partial_u\partial_v + 4(1-u)u^2\partial_u^2 - 4uv^2\partial_v^2.
\end{aligned}
\tag{70}
$$

The other combinations are equivalent, e.g. $D_{[\bar{1}2\bar{3}4]} = D_{[1\bar{2}3\bar{4}]}$. Finally we show a single representative of the family $D_{[\mathbf{i}j\bar{k}]}$ since these are already quite lengthy,

$$
\begin{aligned}
D_{[12\bar{3}]} \equiv{}& +\left(\Delta_1 + \Delta_2\right)\Delta_{34}\sqrt{u}\left(-d + 2\Delta_1 + 2\right) - 2\left(\Delta_1 + \Delta_2 - \Delta_{34} - 2\right)u^{3/2}\left(d - 2(\Delta_1 + 1)\right)\partial_u \\
&+ \left(4u^{3/2}v\left(d - 3\Delta_1 - \Delta_2 + \Delta_{34} + 2\right) - 4\left(\Delta_{12} - \Delta_{34} - 2\right)(u-1)u^{3/2}\right)\partial_u\partial_v \\
&- 2\sqrt{u}\left[\left(\Delta_1 + \Delta_2\right)v\left(d - 2\Delta_1 + \Delta_{34}\right) + \left(\Delta_{12} - \Delta_{34} - 2\right)\left(\Delta_1 + \Delta_2 + \Delta_{34}u + 2u\right)\right]\partial_v \\
&+ 8u^{3/2}v^2\partial_v^3 + 8u^{3/2}v(v+u-1)\partial_u\partial_v^2 + 4u^{5/2}\left(d - 2(\Delta_1 + 1)\right)\partial_u^2 \\
&+ 8u^{5/2}v\partial_u^2\partial_v + 4\sqrt{u}v\left(\left(\Delta_1 + \Delta_2 - \Delta_{12}u + 2\Delta_{34}u + 6u\right) - \left(\Delta_1 + \Delta_2\right)v\right)\partial_v^2.
\end{aligned}
\tag{71}
$$

The operator (71) has an explicit dependence on $d$. This feature is shared with all the other operators which we are omitting from the text.

All differential operators are written in a form where the derivatives in $u$ and $v$ are on the right and thus act directly on the function. This formulation is the most convenient for computations, however there may exist a different formulation which makes the differential operators more compact. We did not invest time in searching for compact rewritings, which we postpone for future work.

We checked that the differential operators automatically satisfy crossing. For example the crossing $1 \leftrightarrow 3$ is written as

$$D_{[\vec{S}]} f(u,v)\big|_{\Delta_1 \leftrightarrow \Delta_3, u \leftrightarrow v} = \sigma_{\vec{S}'} \frac{D_{[\vec{S}']} f(u,v)}{\Sigma_{[\vec{S}']} v^{\frac{-\Delta_2-\Delta_3}{2}} u^{\frac{\Delta_1+\Delta_2}{2}}}, \tag{72}$$

where we assume that $f(u,v)$ is crossing covariant, namely $f(u,v) = v^{\frac{-\Delta_2-\Delta_3}{2}} u^{\frac{\Delta_1+\Delta_2}{2}} f(v,u)$ (which must be the case since it comes from a well defined CFT$_{d-2}$). Here we define $[\vec{S}'] = [\vec{S}]|_{1 \leftrightarrow 3}$ and $\sigma_{\vec{S}} = \pm 1$ is the fermionic signature of $\vec{S}$, namely it computes how many permutations of the fermionic operators are needed to restore the canonical order, e.g. $\sigma_{\bar{2}143} = \sigma_{\bar{2}134} = -\sigma_{1\bar{2}34} = -1$.

We further checked the crossing $1 \leftrightarrow 2$ and $1 \leftrightarrow 4$, which also work out correctly. These equations generically map $D_{[\vec{S}]}$ into a different $D_{[\vec{S}']}$, which is a check that all the differential operators are correct. Moreover this implies that all primary components of the superspace four-point functions automatically satisfy crossing in all channels. This is an important check for the uplift, indeed we find that given one four-point function in $d-2$ we get 43 new four-point functions in $d$ dimensions which are crossing covariant and consistent with supersymmetry. In the next section we explain that they can also be automatically decomposed in $d$-dimensional conformal blocks.

## 4 Conformal block expansion in PS CFTs

In this section we want to study how the different components of a four-point function are all compatible with the conformal block decomposition. This will give rise to 43 relations between conformal blocks in $d-2$ and $d$ dimensions.

As explained in section 2.3, the scalar four-point function in superspace is determined in terms of a single function $f(U,V)$ of super cross ratios which can be decomposed in superspace superconformal blocks $G_{\Delta\ell}^{(d|2)}$ as in (28). On the other hand $f(u,v)$ also defines the dimensionally reduced four-point function (which depends on the usual cross ratios $u,v$) and thus it can be decomposed in $(d-2)$-dimensional blocks

$$f(u,v) = \sum_{\Delta\ell} \tilde{a}_{\Delta\ell} g_{\Delta\ell}^{(d-2)}(u,v), \tag{73}$$

where in principle $\tilde{a}_{\Delta\ell}$ and $a_{\Delta\ell}$ of the two decompositions could be different. In [7] it was shown that they are actually the same $\tilde{a}_{\Delta\ell} = a_{\Delta\ell}$, since also the conformal blocks are the same function (29). This already explains that given a scalar four-point function in $d-2$ dimensions compatible with the conformal block decomposition (and other bootstrap axioms like crossing) one can define a scalar four-point function in $\mathbb{R}^{d|2}$ superspace which is compatible with the superconformal block decomposition (and the other bootstrap axioms).

However it is also important to check that the $d$-dimensional PS CFT behaves as a good CFT$_d$ and therefore its four-point functions are decomposed in terms of $d$-dimensional conformal blocks $g_{\Delta\ell}^{(d)}$. In particular this means that each component of $G_{\Delta\ell}^{(d|2)}(U,V)$ should be

decomposed in terms of a finite number of $g_{\Delta \ell}^{(d)}$ in a precise linear combination fixed by supersymmetry. In [7] we showed that this is indeed true for the lowest component. In particular since the lowest component of $G_{\Delta \ell}^{(d|2)}(U,V)$ is also equal to the block $g_{\Delta \ell}^{(d-2)}$ itself, we obtained the following beautiful formula relating conformal blocks in $d-2$ and $d$ dimensions [7]

$$g_{\Delta \ell}^{(d-2)} = g_{\Delta \ell}^{(d)} + c_{2,0}\, g_{\Delta+2\,\ell}^{(d)} + c_{1,-1}\, g_{\Delta+1\,\ell-1}^{(d)} + c_{0,-2}\, g_{\Delta\,\ell-2}^{(d)} + c_{2,-2}\, g_{\Delta+2\,\ell-2}^{(d)}, \tag{74}$$

where the generic scalar block in $d-2$ dimensions is written as a linear combination of only five blocks in $d$ dimensions. The coefficients can be written in closed form as follows:

$$
\begin{aligned}
c_{2,0} &= -\frac{(\Delta-1)\Delta(\Delta-\Delta_{12}+\ell)(\Delta+\Delta_{12}+\ell)\big(\Delta-\Delta_{34}+\ell\big)\big(\Delta+\Delta_{34}+\ell\big)}{4(d-2\Delta-4)(d-2\Delta-2)(\Delta+\ell-1)(\Delta+\ell)^2(\Delta+\ell+1)}, \\
c_{1,-1} &= -\frac{(\Delta-1)\Delta_{12}\Delta_{34}\ell}{(\Delta+\ell-2)(\Delta+\ell)(d-\Delta+\ell-4)(d-\Delta+\ell-2)}, \\
c_{0,-2} &= -\frac{(\ell-1)\ell}{(d+2\ell-6)(d+2\ell-4)}, \\
c_{2,-2} &= \frac{(\Delta-1)\Delta(\ell-1)\ell(d-\Delta-\Delta_{12}+\ell-4)(d-\Delta+\Delta_{12}+\ell-4)\big(d-\Delta-\Delta_{34}+\ell-4\big)\big(d-\Delta+\Delta_{34}+\ell-4\big)}{4(d-2\Delta-4)(d-2\Delta-2)(d+2\ell-6)(d+2\ell-4)(d-\Delta+\ell-5)(d-\Delta+\ell-4)^2(d-\Delta+\ell-3)}.
\end{aligned}
\tag{75}
$$

Because of this relation it is trivial to see that if $f(u,v)$ is decomposed in $g_{\Delta \ell}^{(d-2)}$, then it can be also decomposed in $g_{\Delta \ell}^{(d)}$ and the explicit form of the coefficients $c_{i,j}$ can be used to simply map the OPE coefficients of the $d-2$ dimensional decomposition, to the OPE coefficients of the $d$-dimensional one.

This relation however was obtained by only considering the lowest component of $f(U,V)$. Now we want to see how this relation generalizes for the other components.

## 4.1   43 relations between conformal blocks across dimensions

As we showed in section 3.3, inside a single superspace four-point functions there are 43 inequivalent four-point functions specified by the list $\vec{S}$. Using (57) we can reconstruct each primary four-point function by acting with a given differential operator $D_{[\vec{S}]}$ in the variables $u,v$ on the function $f(u,v)$ that specifies the lowest component. This is of course also true at the level of conformal blocks if we replace $f \to G_{\Delta \ell}^{(d|2)}$ in (57). In particular this replacement defines what are the superconformal blocks for each component. In other words $G_{\Delta \ell}^{(d|2)}(U,V)$ is the *superspace superconformal blocks*, while $D_{[\vec{S}]} G_{\Delta \ell}^{(d|2)}(u,v)$ defines what is usually called a *superconformal block*, which is just a function of $u,v$. Each $D_{[\vec{S}]} G_{\Delta \ell}^{(d|2)}(u,v)$ must be decomposed in a finite number of $d$-dimensional blocks. Using again formula (29), we can replace $G^{(d|2)} \to g^{d-2}$ and thus obtain 42 extra relations between blocks in $d-2$ and $d$ dimensions labelled by the component $\vec{S}$. We can capture all such relations by the following compact formula

$$\boxed{D_{[\vec{s}]} g_{\Delta \ell}^{(d-2)} = \sum_{(i,j)\in P_{[\vec{s}]}} c_{i,j}^{[\vec{S}]} \, \Sigma_{[\vec{S}]} \, g_{\Delta+i\,\ell+j}^{(d)}}, \tag{76}$$

where we recall that $\Sigma_{[\vec{S}]}$ just implements some shifts in the conformal dimensions $\Delta_k$ of the external operators by some units as defined in (48) and (49), while the differential operators $D_{[\vec{S}]}$ are defined in section 3.3 and explicitly given in a Mathematica file attached to the publication. We computed in a closed form all coefficients $c_{i,j}^{[\vec{S}]}$ for all 43 possible choices of $\vec{S}$ as we will show in the following section and in appendix A (for convenience in the Mathematica file we also include the full list of the coefficients together with a check of (76)).

Table 2: For all possible components $[\vec{S}]$ of four-point function we show which is their associated $P_{[\vec{S}]}$. The set $P^{(0)}$ appears in 26 instances, $P^{(1)}$ in 16 while $P^{(2)}$ only in a single one.

| $P_{[\vec{S}]}$ | # | $[\vec{S}]$ |
|---|---|---|
| $P^{(0)}$ | 26 | $[\,],[1\bar{2}],[3\bar{4}],[\mathbf{1}],[\mathbf{2}],[\mathbf{3}],[\mathbf{4}],[13\bar{4}],[23\bar{4}],[31\bar{2}],[41\bar{2}],[1\bar{2}3\bar{4}],[1\bar{2}4\bar{3}],$ $[\mathbf{12}],[\mathbf{13}],[\mathbf{14}],[\mathbf{23}],[\mathbf{24}],[\mathbf{34}],[123\bar{4}],[341\bar{2}],[\mathbf{123}],[\mathbf{124}],[\mathbf{134}],[\mathbf{234}],[\mathbf{1234}]$ |
| $P^{(1)}$ | 16 | $[1\bar{3}],[1\bar{4}],[2\bar{3}],[2\bar{4}],[231\bar{4}],[132\bar{4}],[241\bar{3}],[142\bar{3}],$ $[12\bar{3}],[12\bar{4}],[21\bar{3}],[21\bar{4}],[31\bar{4}],[32\bar{4}],[41\bar{3}],[42\bar{3}]$ |
| $P^{(2)}$ | 1 | $[1\bar{3}2\bar{4}]$ |

Finally it is crucial that the summation over $P_{[\vec{S}]}$ is finite; in particular it contains at most 5 terms depending on the choice of $\vec{S}$. Indeed $P_{[\vec{S}]}$ is defined as one the following three possible sets

$$P_{[\vec{S}]} = \begin{cases} P^{(0)} \equiv \{(0,0),(0,-2),(1,-1),(2,0),(2,-2)\}, & \text{if } Q_1[\vec{S}] + Q_2[\vec{S}] = 0, \\ P^{(1)} \equiv \{(0,-1),(1,0),(1,-2),(2,-1)\}, & \text{if } Q_1[\vec{S}] + Q_2[\vec{S}] = \pm 1, \\ P^{(2)} \equiv \{(1,-1)\}, & \text{if } Q_1[\vec{S}] + Q_2[\vec{S}] = \pm 2, \end{cases} \quad (77)$$

where $Q_k$ is the $Sp(2)$ R-symmetry charge (computed by $M^{\theta\bar{\theta}}$) of the operator at the position $k$. Namely if the $k$-th operator is a fermion then $Q_k = 1$, if it is an antifermion $Q_k = -1$, while if it is bosonic $Q_k = 0$. The charge of the $k$-th operator can be extracted from the list $\vec{S}$ in a very simple way, by defining $Q_k[\vec{S}] = Q_k[S_1, \ldots, S_n] = Q_k[S_1] + \cdots + Q_k[S_n]$ with

$$Q_k[i\bar{j}] = \delta_{ik} - \delta_{jk}, \qquad Q_k[\mathbf{i}] = 0. \quad (78)$$

In table 2 we present a table that summarizes all components $[\vec{S}]$ of the four-point function and their associated sets $P_{[\vec{S}]}$.

Of course it is natural that the *rhs* of (76) depends on the charge of the operators at position 1 and 2. Indeed, since the $Sp(2)$ charge is conserved, the charge of these two operators specifies which operator can flow in their OPE and thus which conformal blocks can be exchanged. Keeping this in mind let us now explain why the specific form of (77) arises.

Let us first consider the case $Q_1 + Q_2 = 0$ when bosons are exchanged. In this case there are 5 different primaries that can be exchanged for every spin $\ell$ super primary.[6] These are obtained from a single spin $\ell$ super primary $\mathcal{O}^{a_1\ldots a_\ell}$ as follows

$$\mathcal{O}_0^{a_1\ldots a_\ell}, \qquad \mathcal{O}_0^{a_1\ldots a_{\ell-2}\theta\bar{\theta}}, \qquad \mathcal{O}_{\theta\bar{\theta}}^{a_1\ldots a_\ell}, \qquad \mathcal{O}_{\theta}^{a_1\ldots a_{\ell-1}\bar{\theta}}(\mathcal{O}_{\bar{\theta}}^{a_1\ldots a_{\ell-1}\theta}), \qquad \mathcal{O}_{\theta\bar{\theta}}^{a_1\ldots a_{\ell-2}\theta\bar{\theta}}. \quad (79)$$

These have quantum numbers $(\Delta, \ell)$, $(\Delta+2, \ell)$, $(\Delta+1, \ell-1)$, $(\Delta+2, \ell)$, $(\Delta+2, \ell-2)$ as prescribed by (77) and (76). To be precise the operators above may not be all primaries, but they can be always improved to be primaries, so this argument is enough to explain the counting and the quantum numbers of the conformal blocks in the *rhs* of (76). Similarly when $Q_1 + Q_2 = \pm 1$ there are 4 primaries which are fermionic (or antifermionic), which are related to the following operators

$$\mathcal{O}_0^{a_1\ldots a_{\ell-1}\theta}, \qquad \mathcal{O}_{\theta}^{a_1\ldots a_\ell}, \qquad \mathcal{O}_{\theta}^{a_1\ldots a_{\ell-2}\theta\bar{\theta}}, \qquad \mathcal{O}_{\theta\bar{\theta}}^{a_1\ldots a_{\ell-1}\theta}, \quad (80)$$

---

[6]For this counting we consider $\ell$ large enough otherwise there may be less than 5 coefficients. The coefficients $c_{i,j}^{[\vec{S}]}$ have zeros which automatically take into account when some contributions do not arise. Similar remarks hold for the other cases below.

for $Q_1 + Q_2 = 1$ and similarly for the opposite charge. These corresponds to operators with quantum numbers $(\Delta, \ell - 1)$, $(\Delta + 1, \ell)$, $(\Delta + 1, \ell - 2)$, $(\Delta + 2, \ell - 1)$, as shown in (77). Finally when $Q_1 + Q_2 = \pm 2$ there is a single charge-2 (or $-2$) fermion

$$\mathcal{O}_\theta^{\alpha_1 \dots \alpha_{\ell-1} \theta} \left( \mathcal{O}_{\bar\theta}^{\alpha_1 \dots \alpha_{\ell-1} \bar\theta} \right), \tag{81}$$

with quantum numbers $(\Delta + 1, \ell - 1)$. This simple analysis fully explains the sets in (77) and thus the structure of equation (76). In what follows we explicitly show the form of $c_{i,j}^{[\vec{S}]}$.

## 4.2 Examples

In the previous section we introduced all ingredients for the equation (76) besides the coefficient $c_{i,j}^{[\vec{S}]}$. Here we want to show their form in a few examples (the rest can be found in appendix A). While doing it we will also provide a few explicit examples of equation (76) for some choices of $\vec{S}$.

Let us first consider the case of $\vec{S} = \mathbf{i}$ for $i = 1, \dots 4$, namely when there is a single level-2 superdescendant and all other operators are lowest components. Let us exemplify in this case all the ingredients entering (76). The differential operators $D_{[\mathbf{i}]}$ are defined in equations (58)-(61). The charges $Q_1$ and $Q_2$ defined in (78) are zero, thus $P_{[\mathbf{i}]} = P^{(0)}$ which contains 5 terms. We thus find that there are five $d$-dimensional conformal blocks appearing in the *rhs* of (76), with $\Delta_i$ shifted by two units (namely $\Delta_i \to \Delta_i + 2$). We therefore obtain that (76) can be unpacked as follows

$$D_{[\mathbf{i}]} g_{\Delta\ell}^{(d-2)} = c_{0,0}^{[\mathbf{i}]} \Sigma_{\mathbf{i}} g_{\Delta\ell}^{(d)} + c_{0,-2}^{[\mathbf{i}]} \Sigma_{\mathbf{i}} g_{\Delta\ell-2}^{(d)} + c_{1,-1}^{[\mathbf{i}]} \Sigma_{\mathbf{i}} g_{\Delta+1\ell-1}^{(d)} + c_{2,0}^{[\mathbf{i}]} \Sigma_{\mathbf{i}} g_{\Delta+2\ell}^{(d)} + c_{2,-2}^{[\mathbf{i}]} \Sigma_{\mathbf{i}} g_{\Delta+2\ell-2}^{(d)}, \tag{82}$$

where $i = 1, 2, 3, 4$. For $i = 1$ the exact form of the coefficients is as follows,

$$
\begin{aligned}
c_{0,0}^{[\mathbf{1}]} &= (-\Delta - \Delta_{12} - \ell)(-\Delta + \beta_{12} + \ell), \\
c_{0,-2}^{[\mathbf{1}]} &= \frac{(\ell-1)\ell(d - \Delta - \Delta_{12} + \ell - 2)(d + \Delta - \beta_{12} + \ell - 2)}{(d + 2\ell - 4)(d + 2\ell - 2)}, \\
c_{1,-1}^{[\mathbf{1}]} &= \frac{(1-\Delta)\Delta_{34}\ell(d - \beta_{12})(\Delta + \Delta_{12} + \ell)(-d + \Delta + \Delta_{12} - \ell + 2)}{(\Delta + \ell - 2)(\Delta + \ell)(-d + \Delta - \ell)(-d + \Delta - \ell + 2)}, \\
c_{2,0}^{[\mathbf{1}]} &= \frac{(1-\Delta)\Delta(\Delta + \Delta_{12} + \ell)(\Delta + \Delta_{12} + \ell + 2)(\Delta - \Delta_{34} + \ell)(\Delta + \Delta_{34} + \ell)(-d + \Delta + \Delta_{12} - \ell + 2)(-d + \Delta + \beta_{12} + \ell)}{4(2\Delta - d)(-d + 2\Delta + 2)(\Delta + \ell - 1)(\Delta + \ell)^2(\Delta + \ell + 1)}, \\
c_{2,-2}^{[\mathbf{1}]} &= \frac{(1-\Delta)\Delta(\ell-1)\ell(\Delta + \Delta_{12} + \ell)(d - \Delta - \Delta_{12} + \ell - 4)(d - \Delta - \Delta_{12} + \ell - 2)(2d - \Delta - \beta_{12} + \ell - 2)(d - \Delta - \Delta_{34} + \ell - 2)(d - \Delta + \Delta_{34} + \ell - 2)}{4(d - 2\Delta - 2)(d - 2\Delta)(d + 2\ell - 4)(d + 2\ell - 2)(d - \Delta + \ell - 3)(d - \Delta + \ell - 2)^2(d - \Delta + \ell - 1)},
\end{aligned}
\tag{83}
$$

where we introduced the notation $\beta_{ij} \equiv \Delta_i + \Delta_j$.

All other coefficients $c_{i,j}^{[\mathbf{k}]}$ (for $k = 2, 3, 4$) are obtained by a simple map of the coefficient $c_{i,j}^{[\mathbf{1}]}$, namely

$$c_{i,j}^{[\mathbf{2}]} = \pi_{(12)(34)} c_{i,j}^{[\mathbf{1}]}, \qquad c_{i,j}^{[\mathbf{3}]} = \pi_{(13)(24)} c_{i,j}^{[\mathbf{1}]}, \qquad c_{i,j}^{[\mathbf{4}]} = \pi_{(23)(14)} c_{i,j}^{[\mathbf{1}]}, \tag{84}$$

where $\pi$ implements permutations of the labels $\Delta_i$. In particular

$$\pi_{(ij)} F(\Delta_k) \equiv F(\Delta_k)|_{\Delta_i \leftrightarrow \Delta_j}, \qquad \pi_{(i_1 j_1) \dots (i_n j_n)} F(\Delta_k) \equiv \pi_{(i_1 j_1)} \cdots \pi_{(i_n j_n)} F(\Delta_k), \tag{85}$$

for any function $F$ of the conformal dimensions $\Delta_k$.

Let us mention a common feature that we will see for most of choices of $\vec{S}$: the relations (76) can be though as equations for the conformal blocks, but the latter only depend on the differences $\Delta_{12}$ and $\Delta_{34}$, while the coefficients $c_{i,j}^{[\vec{S}]}$ (see e.g. (83)) as well as the differential operators $D_{[\vec{S}]}$ (see e.g. (58)) depend also on $\beta_{12}$, $\beta_{34}$ (polynomially). Therefore one can expand the equations (76) in $\beta_{12}$ and $\beta_{34}$ and each term of the expansion can be understood as a valid equation for the blocks.

Let us now consider the case of $\vec{S} = i\bar{j}$ ($i \neq j$), which corresponds to two level-one superdescendants and two lowest components. There are two distinct situations. First we consider $(i, j) = (1, 2), (3, 4)$. In this case $P_{[i\bar{j}]}$ contains 5 terms. Therefore the action of $D_{[i\bar{j}]}$ (defined in (62) and (63)) on the $d-2$-dimensional conformal blocks can be written in terms of a sum of five conformal blocks in $d$-dimensions,

$$D_{[i\bar{j}]}g_{\Delta\ell}^{(d-2)} = c_{0,0}^{[i\bar{j}]}g_{\Delta\ell}^{(d)} + c_{0,-2}^{[i\bar{j}]}g_{\Delta\ell-2}^{(d)} + c_{1,-1}^{[i\bar{j}]}g_{\Delta+1\ell-1}^{(d)} + c_{2,0}^{[i\bar{j}]}g_{\Delta+2\ell}^{(d)} + c_{2,-2}^{[i\bar{j}]}g_{\Delta+2\ell-2}^{(d)}. \quad (86)$$

Here the conformal blocks are not shifted since they only depend on $\Delta_{12}$ and $\Delta_{34}$ and thus the shift act trivially. The form of the coefficients is as follows,

$$
\begin{aligned}
c_{0,0}^{[1\bar{2}]} &= \Delta - \beta_{12} - \ell, & (87)\\
c_{0,-2}^{[1\bar{2}]} &= \frac{(\ell-1)\ell(\beta_{12}-d-\Delta-\ell+2)}{(d+2\ell-4)(d+2\ell-2)}, \\
c_{1,-1}^{[1\bar{2}]} &= \frac{(\Delta-1)\Delta_{12}\Delta_{34}\ell(\beta_{12}-d)}{(\Delta+\ell-2)(\Delta+\ell)(-d+\Delta-\ell)(-d+\Delta-\ell+2)}, \\
c_{2,0}^{[1\bar{2}]} &= \frac{(\Delta-1)\Delta(\Delta-\Delta_{12}+\ell)(\Delta+\Delta_{12}+\ell)(\Delta-\Delta_{34}+\ell)(\Delta+\Delta_{34}+\ell)(-d+\Delta+\beta_{12}+\ell)}{4(2\Delta-d)(-d+2\Delta+2)(\Delta+\ell-1)(\Delta+\ell)^2(\Delta+\ell+1)}, \\
c_{2,-2}^{[1\bar{2}]} &= \frac{(\Delta-1)\Delta(\ell-1)\ell(d-\Delta+\Delta_{12}+\ell-2)(\Delta_{12}-d+\Delta-\ell+2)(\beta_{12}-2d+\Delta-\ell+2)(\Delta-d-\Delta_{34}-\ell+2)(\Delta-d+\Delta_{34}-\ell+2)}{4(d-2\Delta)(d-2(\Delta+1))(d+2\ell-4)(d+2\ell-2)(d-\Delta+\ell-3)(d-\Delta+\ell-2)^2(d-\Delta+\ell-1)}.
\end{aligned}
$$

The other case $[3\bar{4}]$ is defined by

$$c_{i,j}^{[3\bar{4}]} \equiv \pi_{(13)(24)}c_{i,j}^{[1\bar{2}]}. \quad (88)$$

In the cases $(i, j) = (1, 3), (1, 4), (2, 3), (2, 4)$ the set $P_{[i\bar{j}]}$ contains 4 terms. We thus find that the action of $D_{[i\bar{j}]}$ (defined in (64)-(67)) on the $d-2$-dimensional conformal blocks can be written in terms of a sum of only four conformal blocks in $d$-dimensions, with $\Delta_i$ and $\Delta_j$ shifted by one unit,

$$D_{[i\bar{j}]}g_{\Delta\ell}^{(d-2)} = c_{0,-1}^{[i\bar{j}]}\Sigma_{ij}g_{\Delta\ell-1}^{(d)} + c_{1,0}^{[i\bar{j}]}\Sigma_{ij}g_{\Delta+1\ell}^{(d)} + c_{1,-2}^{[i\bar{j}]}\Sigma_{ij}g_{\Delta+1\ell-2}^{(d)} + c_{2,-1}^{[i\bar{j}]}\Sigma_{ij}g_{\Delta+2\ell-1}^{(d)}. \quad (89)$$

The coefficient for $[1\bar{3}]$ takes the following form

$$
\begin{aligned}
c_{0,-1}^{[1\bar{3}]} &= -\ell, \\
c_{1,0}^{[1\bar{3}]} &= -\frac{(\Delta-1)(\Delta+\Delta_{12}+\ell)(\Delta+\Delta_{34}+\ell)}{2(\Delta+\ell-1)(\Delta+\ell)}, \\
c_{1,-2}^{[1\bar{3}]} &= \frac{(\Delta-1)(\ell-1)\ell(d-\Delta-\Delta_{12}+\ell-2)(d-\Delta-\Delta_{34}+\ell-2)}{2(d+2\ell-4)(d+2\ell-2)(d-\Delta+\ell-2)(d-\Delta+\ell-1)}, \\
c_{2,-1}^{[1\bar{3}]} &= \frac{(\Delta-1)\Delta\ell(\Delta+\Delta_{12}+\ell)(\Delta+\Delta_{34}+\ell)(d-\Delta-\Delta_{12}+\ell-2)(d-\Delta-\Delta_{34}+\ell-2)}{4(2\Delta-d)(-d+2\Delta+2)(\Delta+\ell-1)(\Delta+\ell)(-d+\Delta-\ell+1)(-d+\Delta-\ell+2)}.
\end{aligned}
\quad (90)
$$

The other three coefficients are defined by

$$c_{i,j}^{[2\bar{4}]} = \pi_{(14)(23)}c_{i,j}^{[1\bar{3}]}, \qquad c_{i,j}^{[1\bar{4}]} = (-1)^j\pi_{(34)}c_{i,j}^{[1\bar{3}]}, \qquad c_{i,j}^{[1\bar{3}]} = (-1)^j\pi_{(12)}c_{i,j}^{[1\bar{3}]}, \quad (91)$$

where $(-1)^j$ changes the sign of the coefficients $c_{0,-1}$ and $c_{2,-1}$ (notice that these correspond to operators with spin $\ell-1$, thus they have an extra sign with respect to operators with spin $\ell$ or $\ell-2$).

Let us consider in details also the three cases when all four operators are level-one superdescendant. In the first case $1\bar{2}3\bar{4}$, the set $P_{[1\bar{2}3\bar{4}]}$ contains 5 terms and thus

$$D_{[1\bar{2}3\bar{4}]}g_{\Delta\ell}^{(d-2)} = c_{0,0}^{[1\bar{2}3\bar{4}]}g_{\Delta\ell}^{(d)} + c_{0,-2}^{[1\bar{2}3\bar{4}]}g_{\Delta\ell-2}^{(d)} + c_{1,-1}^{[1\bar{2}3\bar{4}]}g_{\Delta+1\ell-1}^{(d)} + c_{2,0}^{[1\bar{2}3\bar{4}]}g_{\Delta+2\ell}^{(d)} + c_{2,-2}^{[1\bar{2}3\bar{4}]}g_{\Delta+2\ell-2}^{(d)}, \quad (92)$$

where $D_{[1\bar{2}3\bar{4}]}$ is defined in (68) and the coefficients read

$$c_{0,0}^{[1\bar{2}3\bar{4}]} = (\beta_{12} - \Delta + \ell)(\beta_{34} - \Delta + \ell), \tag{93}$$

$$c_{0,-2}^{[1\bar{2}3\bar{4}]} = \frac{(\ell-1)\ell(d + \Delta - \beta_{12} + \ell - 2)(\beta_{34} - d - \Delta - \ell + 2)}{(d + 2\ell - 4)(d + 2\ell - 2)},$$

$$c_{1,-1}^{[1\bar{2}3\bar{4}]} = \frac{(\Delta-1)\ell\left(\Delta_{12}\Delta_{34}(-d+\beta_{12})(d-\beta_{34}) + (\Delta+\ell-2)(\Delta+\ell)(d-\Delta+\ell-2)(d-\Delta+\ell)\right)}{(\Delta+\ell-2)(\Delta+\ell)(d-\Delta+\ell-2)(d-\Delta+\ell)},$$

$$c_{2,0}^{[1\bar{2}3\bar{4}]} = \frac{(\Delta-1)\Delta(\Delta_{12}-\Delta-\ell)(\Delta+\Delta_{12}+\ell)(\Delta-\Delta_{34}+\ell)(\Delta+\Delta_{34}+\ell)(-d+\Delta+\beta_{12}+\ell)(-d+\Delta+\beta_{34}+\ell)}{4(2\Delta-d)(-d+2\Delta+2)(\Delta+\ell-1)(\Delta+\ell)^2(\Delta+\ell+1)},$$

$$c_{2,-2}^{[1\bar{2}3\bar{4}]} = \frac{(\Delta-1)\Delta(\ell-1)\ell(d-\Delta-\Delta_{12}+\ell-2)(2d-\Delta-\beta_{12}+\ell-2)(d-\Delta+\Delta_{12}+\ell-2)(d-\Delta-\Delta_{34}+\ell-2)(2d-\Delta-\beta_{34}+\ell-2)(d-\Delta+\Delta_{34}+\ell-2)}{4(d-2\Delta)(d-2(\Delta+1))(d+2\ell-4)(d+2\ell-2)(d-\Delta+\ell-3)(d-\Delta+\ell-2)^2(d-\Delta+\ell-1)}.$$

The case $1\bar{2}4\bar{3}$ is similar and the coefficients are related to the ones above by $c_{i,j}^{[1\bar{2}4\bar{3}]} = (-1)^j \pi_{(34)} c_{i,j}^{[1\bar{2}3\bar{4}]}$.

We now consider the case $1\bar{3}2\bar{4}$. This is the only case where the set $P_{[1\bar{3}2\bar{4}]}$ contains only one term. The single coefficient is defined as $c_{1,-1}^{[1\bar{3}2\bar{4}]} = 2(\Delta - 1)\ell$, therefore the final formula takes the very simple form

$$D_{[1\bar{3}2\bar{4}]} g_{\Delta\,\ell}^{(d-2)} = 2(\Delta - 1)\ell\, g_{\Delta+1\,\ell-1}^{(d)}, \tag{94}$$

where $D_{[1\bar{3}2\bar{4}]}$ is defined in (69). The differential operator $D_{[1\bar{3}2\bar{4}]}/(2(\Delta-1)\ell)$ is very useful since it can be applied to define conformal blocks in higher dimensions knowing the ones in lower dimensions.

Indeed the relation (94) was already found in equation (4.37) of [30][7] by searching for a differential operator that shifts the dimension of the conformal blocks. Now we can give a physical interpretation to this relation as arising from supersymmetry.

We avoid the exemplifications of all other cases, but it should be clear that all 43 relations in (76) can be easily implemented using the differential operators and the coefficients defined in the Mathematica file attached to the publication. For completeness in appendix A we also define the remaining coefficients.

## 4.3 Poles in the conformal blocks and SUSY

In this section we aim at studying the possible singularities in $\Delta$ of conformal blocks. Below we warm up by considering a unitary CFT and show that in this case the blocks are always finite even when they naively look singular. Then we turn to a PS CFT and consider the superconformal blocks, which by equation (76) are written as a linear combination of conformal blocks. We will show that in this case it may also happen that some blocks in the linear combination diverge, but the specific linear combination provided by (76) is fine tuned to cancel all singularities and gives a finite result. On one hand this remark can be understood as a consistency check of (76). On the other hand it will also uncover some interesting features which will play an important role in the conformal block decompositions of the following sections.

To start let us consider the conformal block decomposition of a four-point correlation function in usual (compact unitary) CFTs. In this case it is expected that both the OPE coefficients and the conformal blocks appearing in the decomposition are finite quantities and sum up to a finite result. The fact that conformal blocks are finite is non trivial since they have poles in $\Delta$ as shown in (12), which for convenience we rewrite here

$$g_{\Delta\ell} \sim \frac{R_A}{\Delta - \Delta_A^*} g_{\Delta_A \ell_A}, \tag{95}$$

where $\Delta_A^*$, $\Delta_A = \Delta_A^* + n_A$, $\ell_A$ are defined in table 1 and the coefficient $R_A$ is defined in (13). So the conformal blocks seem to diverge for the exchange of an operator with labels $\Delta = \Delta_A^*, \ell$.

---

[7]The normalization of the blocks is different $g_{\Delta,\ell}^{(d),here} = \frac{(-2)^\ell\left(\frac{d-2}{2}\right)_\ell}{(d-2)_\ell} g_{\Delta,\ell}^{(d),there}$.

At a first sight this might be puzzling since these labels include the ones of a free scalar ($\Delta = d/2-1, \ell = 0$) and a conserved tensors (with $\Delta = d + \ell - 2$ for any spin $\ell = 1, 2, \ldots$). So naively it may seem that free scalars and conserved tensors cannot be exchanged. How can this be compatible with the fact that these operators exist in physical theories? The answer is simple. In unitary theories when $\Delta, \ell$ equals the ones of a free scalar or a conserved tensor then the associated residue $R_A$ vanishes to cancel the pole. This gives a non trivial restriction on the possible set of correlation functions that exchange such operators. For example for a free scalar exchange the conformal block diverges as

$$g_{\Delta\,\ell=0} \sim \frac{R_{\mathrm{III},1}|_{\ell=0}}{\Delta - (d/2-1)} g_{\Delta+2,\ell=0}, \qquad \text{for} \qquad \Delta \to d/2-1. \qquad (96)$$

The residue at $\Delta = d/2-1$ vanishes when

$$R_{\mathrm{III},1}|_{\ell=0} = \frac{(d/2-1-\Delta_{12})(d/2-1+\Delta_{12})(d/2-1-\Delta_{34})(d/2-1+\Delta_{34})}{4(d-2)d} = 0, \qquad (97)$$

which implies that either $\Delta_{12} = \pm(d/2-1)$ or $\Delta_{34} = \pm(d/2-1)$. This is indeed what happens in free theory: in order to exchange a free field in an OPE, the dimensions of the (external) operators in the correlation function must differ by $\pm(d/2-1)$, e.g. $\phi^n \times \phi^{n-1} \sim \phi$.

Similarly when we exchange a conserved tensor of spin $\ell$ the conformal block diverges as

$$g_{\Delta\,\ell} \sim \frac{R_{\mathrm{II},1}}{\Delta - (d+\ell-2)} g_{\Delta+1,\ell-1}, \qquad \text{for} \qquad \Delta \to d+\ell-2. \qquad (98)$$

Thus we need to require that the residue at $\Delta = d + \ell - 2$ has to vanish, namely

$$R_{\mathrm{II},1} = \frac{\Delta_{12}\Delta_{34}\ell!(d+\ell-3)}{2(\ell-1)!(d+2\ell-4)(d+2\ell-2)} = 0, \qquad (99)$$

which is possible only for either $\Delta_{12} = 0$ or $\Delta_{34} = 0$. This condition is indeed required by Ward identities of the conserved tensors $\langle \phi_{\Delta_1} \phi_{\Delta_2} T^{\mu_1 \ldots \mu_\ell}\rangle \propto \delta_{\Delta_1 \Delta_2}$.

We thus found that in unitary theories the singularities of the conformal blocks are avoided because the residue $R_A$ is zero. Let us discuss what this means physically. Let us consider a conformal multiplet/module labelled by a primary with dimension $\Delta$ and spin $\ell$. Now we want to see what happens to this multiplet when changing $\Delta$. Using the notation of equation (95), when $\Delta = \Delta_A^*$ the module becomes singular/reducible, namely it contains a descendant labelled by $\Delta_A, \ell_A$ which becomes primary. The primary descendant together with all its descendants define the submodule. All states of the submodule have vanishing norm. These are responsible for the pole in the conformal block. The condition $R_A = 0$ is equivalent to the fact that there is a shortening condition that the primary descendant must satisfy. For example $\Box\phi = 0$ for a free scalar or $\partial_\mu J^{\mu\mu_2\cdots\mu_\ell} = 0$ for a spin $\ell$ conserved current. These shortening conditions can be understood as a way to mod out the primary descendant and its associated multiplet. This is necessary in a unitary theory to avoid non-trivial states with vanishing norms. In the following we see how this does not happen in PS CFTs.

**Avoiding poles with Parisi-Sourlas SUSY**

Above we showed that a tuning of the conformal blocks is required when operators at the unitarity bound are exchanged. Here we exemplify how this is not necessary in PS CFTs. This will also generalize to exchanges below the unitarity bound which are naively singular (as we detail in appendix B).

Let us exemplify the new mechanism for the case of an exchanged scalar superblock with $\Delta = d/2 - 1$. We further think of this superblock as arising from the uplift of a $\hat{d} = d - 2$-dimensional block. Since in $\hat{d}$ dimensions nothing special happens for exchanges of operators of dimensions $\Delta = d/2 - 1 = \hat{d}/2$, we would expect that the same is true for the superblocks of the uplifted theory, however this is non-trivial since in $d$-dimensions the blocks have a pole at this value of $\Delta$ (unless (97) is satisfied). Let us show how this pole is erased in the case of the lowest component of a four-point function. The lowest component of a scalar ($\ell = 0$) superblock takes the form

$$g_{\Delta\,\ell=0} + c_{2,0}^{(\ell=0)} g_{\Delta+2\,\ell=0}\,, \qquad \text{where} \qquad c_{2,0}^{(\ell=0)} = -\frac{(\Delta - \Delta_{12})(\Delta + \Delta_{12})(\Delta - \Delta_{34})(\Delta + \Delta_{34})}{4\Delta(\Delta+1)(d - 2\Delta - 4)(d - 2\Delta - 2)}\,. \quad (100)$$

We can explicitly see that the coefficient $c_{2,0}^{(\ell=0)}$ in front of $g_{\Delta+2\,\ell=0}$ diverges as follows

$$c_{2,0} \sim -\frac{R_{\text{III},1}}{\Delta - (d/2 - 1)}\,, \qquad \text{for} \qquad \Delta \to d/2 - 1\,. \quad (101)$$

The contribution of $c_{2,0} g_{\Delta+2\,\ell=0}$ exactly cancels the pole at $\Delta \to d/2 - 1$ of $g_{\Delta\,\ell=0}$ defined in formula (12), and thus we get that the combination provided by (the lowest component of) the superblock

$$\tilde{g}_{d/2-1\,\ell=0} \equiv \lim_{\Delta \to d/2 - 1} g_{\Delta\,\ell=0} + c_{2,0} g_{\Delta+2\,\ell=0} \quad (102)$$

is finite. Therefore in a PS CFT the pole at $\Delta = d/2 - 1$ in the conformal block cancels thanks to supersymmetry and the conformal block $\tilde{g}_{d/2-1\,\ell=0}$ can always be exchanged independently on the value of $\Delta_{12}$ and $\Delta_{34}$. This mechanism is unconventional because the pole of the block is cancelled because of a divergent OPE coefficient (to be precise it is only the kinematic part $c_{2,0}$ of the OPE which diverges). In all theories with finite OPE coefficients this can never happen and the only mechanism allowed is the one of the cancellation of the residues $R_A$.

The fact that $R_A \neq 0$ means that the module of the primary descendant is not modded out and thus it will be part of the spectrum. Another consequence of this is that we can consider these primary descendants as insertions in a correlation function without getting a vanishing result, because no shortening condition has to be satisfied (this is in contrast with the unitary cases described above where if one considers e.g. $\square\phi$ as an insertion in any free boson correlation function, the resulting correlator vanishes). We will see how all these facts will play a role in the study of the uplifted minimal models.

There are other important cases where this mechanism takes place. Indeed, besides for exchanges of operators at the unitarity bound, the conformal blocks diverge for infinitely more values of $\Delta = \Delta_A^*$ (the unitary values appear only for the type III, $n = 1$ when $\ell = 0$ and for II, $n = 1$ when $\ell > 0$, but in general one could consider any of the three types I, II, III for any $n$). All these extra cases lye below the unitarity bounds, so these exchanges do not arise in unitary theories. However these may arise in non-unitary theories, like the PS CFTs. For such theories there should be a mechanism to cancel the singularities whenever one of these operators is exchanged.

First it is interesting to ask which are the possible problematic exchanges which arise in the Parisi-Sourlas uplift of a unitary $\hat{d} \equiv d - 2$-dimensional theory (these are the theories that we consider in the rest of the paper). The answer is easy: the operators of such uplifts satisfy bounds which arise by uplifting the unitarity-bounds of the lower dimensional theory. Namely they satisfy $\Delta \geq \frac{d}{2} - 2$ for scalar operators and $\Delta \geq d + \ell - 3$ for operators of spin $\ell \geq 1$. In particular in these theories one can exchange scalar operators with dimensions $\Delta = \frac{d}{2} - 2, \frac{d}{2} - 1$ and operators with dimensions $\Delta = d + \ell - 3, d + \ell - 2, d + \ell - 1$ and spin $\ell$. For all these values the blocks have singularities. For the uplift to make sense it is crucial that all these singularities are resolved. In appendix B we show that all these poles are erased, sometimes because $R_A$

vanishes and sometimes because of the special linear combination of the blocks provided by supersymmetry which gives rise to new regularized blocks that exactly cancels the singularity as we showed for (102). The specific form of the regularized blocks computed in appendix B is going to play a role in the next sections, when we will decompose correlations functions of the uplifted minimal models.

## 5 Bootstrapping GFF using PS SUSY

In the previous sections we determined a set of kinematic properties shared by all PS CFTs. We now want to apply the PS CFT framework to the simple theory of a generalized free boson field (GFF). Below we will start by a basic introduction to GFF to set some conventions. We define it in $\hat{d} = d - 2 > 1$ dimensions (the case $\hat{d} = 1$ is studied in section 7.2) so that the uplifted theory lives in $d > 3$ dimensions (for the purposes of this section one could also drop the hat of $\hat{d}$ in every formula). In the next subsection we will define the uplifted theory. We will show that, besides being well defined, the PS uplift of GFF can be used to infer some properties of the original GFF, thus helping to bootstrap some observables. As an example we will be able to compute, without doing any Wick contraction, the conformal block decomposition of $\langle \phi \phi^n \phi^m \phi^m \rangle$ for any power $n, m$, which is a new result in GFF. We will also explain how the uplift can be used to constrain perturbative computations around GFF.

GFF can be defined through a non-local $\hat{d}$-dimensional Lagrangian $\phi \Box^\xi \phi$, where $\xi$ is a real parameter. The theory is Gaussian and therefore is solvable. An equivalent way to define the theory without introducing a Lagrangian is as follows. The defining property of GFF is that correlation function of fields $\phi$ can be computed through Wick contractions where the propagators are $\langle \phi(x) \phi(0) \rangle = |x|^{-2\Delta_\phi}$. Since the theory enjoys a $\mathbb{Z}_2$ symmetry (which acts as $\phi \to -\phi$) correlation functions of an odd number of $\phi$ fields vanish.

When $\Delta_\phi = \hat{d}/2 - 1$ (which corresponds to $\xi = 1$ in the Lagrangian) the theory is the usual free theory, which has local equations of motions $\Box \phi = 0$ and infinitely many conserved (higher spin) currents. However we shall study the more generic case of $\Delta_\phi$ being a generic real parameter. In this case the field $\phi$ does not satisfy any local shortening condition. Similarly no local composite of $\phi$ (built out of an integer number of $\phi$ dressed with an integer number of derivatives) can play the role of a conserved current/tensor, simply because it would not have integer conformal dimensions (which is required by representation theory). So naively we do not expect shortening conditions for any operator or in other words we do not expect the correlation functions of GFF to satisfy any local differential equation. In the next subsection we shall see that such differential equations can be defined because of the existence of the PS uplift. Before that let us continue the review of GFF by describing some observables.

In GFF it is straightforward to compute correlation functions by Wick contractions. A simple example is

$$\langle \phi(x_1)\phi(x_2)\phi(x_3)\phi(x_4) \rangle = \frac{1}{x_{12}^{2\Delta_\phi} x_{34}^{2\Delta_\phi}} \left[ 1 + u^{\Delta_\phi} + u^{\Delta_\phi} v^{-\Delta_\phi} \right]. \tag{103}$$

The OPE in GFF can also be fixed by Wick contractions, for example

$$\phi(x)\phi(0) = \frac{1}{|x|^{2\Delta_\phi}} + \sum_{n,\ell} \lambda_{n,\ell} |x|^{2n+\ell} [\phi\phi]_{n,\ell}, \tag{104}$$

where the first term is obtained as by Wick contraction of $\phi(x)\phi(0)$ and is associated to the exchange of the identity operator, while $[\phi\phi]_{n,\ell}$ are called double twist operators and $\lambda_{n,\ell}$ are the associated OPE coefficients. The double twists are obtained by Taylor expansion of

$:\phi(x)\phi(0):$ around $x = 0$ and are defined as

$$[\phi\phi]_{n,\ell} \equiv\, :\phi\Box^n\partial^{\mu_1}\cdots\partial^{\mu_\ell}\phi: +\dots, \tag{105}$$

where the dots take into account terms that make the operator symmetric and traceless in the indices $\mu_1\dots\mu_\ell$ together with terms that make the operator primary. By taking the OPE in a four-point function we obtain

$$\langle\phi(x_1)\phi(x_2)\phi(x_3)\phi(x_4)\rangle = \frac{1}{x_{12}^{2\Delta_\phi}x_{34}^{2\Delta_\phi}}\Big[1 + \sum_{n,\ell}a_{n,\ell}\,g^{(\hat{d})}_{2\Delta_\phi+2n+\ell\,\ell}\Big]. \tag{106}$$

Since we know the exact form of the four-point function we can easily read off the coefficients $a_{n,\ell} = \lambda_{n,\ell}^2$ by simply expanding the conformal blocks and the four-point functions in powers of the cross ratios and match the coefficients. There are more sophisticated ways to compute $a_{n,\ell}$ from the four-point function, e.g. the inversion formula of [31], which however will not be needed for this work.

The same logic can be applied for any four-point function. In the following we show how one can use the PS uplift as a tool to compute the conformal block decomposition of a four-point function of $\langle\phi\phi^{n_2}\phi^{n_3}\phi^{n_4}\rangle$ for generic $n_2, n_3, n_4$. In this case the conformal block expansion takes the form

$$\langle\phi(x_1)\phi^{n_2}(x_2)\phi^{n_3}(x_3)\phi^{n_4}(x_4)\rangle = K_{\Delta_i}(x_i)\Big[a^{n_2,n_3,n_4}g^{(\hat{d})}_{\Delta=-\Delta_{12}\,\ell=0} + \sum_{n,\ell}a_{n,\ell}^{n_2,n_3,n_4}g^{(\hat{d})}_{\Delta=\Delta_1+\Delta_2+2n+\ell\,\ell}\Big], \tag{107}$$

where $\Delta_1 = \Delta_\phi$, $\Delta_i = n_i\Delta_\phi$ ($i = 2, 3, 4$) and the contribution at $\Delta = -\Delta_{12} = (n_2 - 1)\Delta_\phi$ comes by taking a Wick contraction of $\phi$ with the operator $\phi^{n_2}$. Very interestingly PS supersymmetry gives a set of recurrence relations for $a_{n,\ell}$, which we were able to solve in the case of $n_3 = n_4$. Besides the specific result it is very interesting that just kinematic considerations of PS supersymmetry can be used to fix the dynamical data $a_{n,\ell}$ with the only input of the Ansatz (107).

Before explaining how this bootstrap problem works, let us present the simplest set of GFF correlators. Because of the tower of double twist operator, in a four-point function we typically expect the exchange of an infinite number of operators. However there is a set of "extremal" four-point function where only one operator is exchanged. These take the simple form

$$\langle\phi^{n_1}(x_1)\phi^{n_2}(x_2)\phi^{n_3}(x_3)\phi^N(x_4)\rangle = K_{\Delta_i}(x_i)u^{-\frac{\Delta_{34}}{2}}N! = K_{\Delta_i}(x_i)N!\,g^{(\hat{d})}_{\Delta=-\Delta_{34}\,\ell=0}(u,v), \tag{108}$$

where $N = n_1 + n_2 + n_3$ and $\Delta_{34} = (n_3 - N)\Delta_\phi$. The factorial is simply obtained from the combinatorics. Nicely enough this actually corresponds to a single conformal block exchange.[8] Whenever $n_1 = 1$ and $1 + n_2 + n_3 = N$ we can compare the result (108) with the Ansatz (107), which shows that all the coefficients $a_{n,\ell}^{n_2,n_3,n_4}$ of the double twist operators vanish, leaving as only contribution $a^{n_2,n_3,N} = N!$.

Below we want to study $a_{n,\ell}^{n_2,n_3,n_4}$ for generic values of $n_i$.

## 5.1 Bootstrapping GFF using PS supersymmetry

Given the GFF in $d - 2$ dimensions, we can obtain its relative PS $\text{CFT}_d$ by explicitly uplifting the Lagrangian of the theory, namely $\phi(\partial^\mu\partial_\mu)^\xi\phi \to \Phi(\partial^a\partial_a)^\xi\Phi$. Like GFF, its uplifted version

---

[8]Similarly if the operator $\phi^N$ is at position $x_1$ we get $g^{(\hat{d})}_{\Delta=\Delta_{12}\,\ell=0}(u,v) = u^{\frac{\Delta_{12}}{2}}$. If it is at $x_2$, the single block is $g^{(\hat{d})}_{\Delta=-\Delta_{12}\,\ell=0}(u,v) = u^{-\frac{\Delta_{12}}{2}}v^{\frac{1}{2}(\Delta_{12}-\Delta_{34})}$. While if it is at $x_3$, the single block is $g^{(\hat{d})}_{\Delta=-\Delta_{34}\,\ell=0}(u,v) = u^{\frac{\Delta_{34}}{2}}v^{\frac{1}{2}(\Delta_{12}-\Delta_{34})}$. So in particular these exchanges are crossing covariant, meaning that a single exchange in the $s$-channel is mapped to a single exchange in the $t$-channel.

can be defined by saying that all correlators are computed by Wick contractions, where now the two-point function lives in superspace $\langle \Phi(y_1)\Phi(y_2) \rangle = |y_{12}|^{-2\Delta_\phi}$, where $\Delta_\phi$ is the same as in the dimensionally reduced theory. When $\Delta_\phi = d/2 - 2$, the model becomes the uplift of free theory but we will keep $\Delta_\phi$ generic.[9] Composite operators are mapped by following the rules that $\phi \to \Phi$ and $\partial_\mu \to \partial_a$.

Now we show how the existence of the uplift can be used to constrain (the dimensionally reduced) GFF. For example to constrain GFF correlation functions, we can leverage the simple fact that

$$\mathbf{D}_{[1]}\langle \Phi(y_1)\Phi(y_2) \rangle|_0 = 0, \tag{109}$$

where $\mathbf{D}_{[1]}\Phi(y_1)|_0 \equiv \tilde{\Phi}_{\theta\bar{\theta}}(x_1)$ is the superdescendant of $\Phi_0$. This two-point function vanishes simply because it involves two different primaries, namely $\langle \tilde{\Phi}_{\theta\bar{\theta}}(x_1)\Phi_0(x_2) \rangle = 0$.[10] Indeed any correlation function of the following form vanishes,

$$\mathbf{D}_{[1]}\langle \Phi(y_1)\Phi^{n_2}(y_2)\ldots\Phi^{n_k}(y_k) \rangle|_0 = 0, \tag{110}$$

for generic $k$ and $n_1, \ldots, n_k$. This is true because the correlation function in GFF is computed by Wick contraction and therefore it is equal to as a sum of terms that contain a vanishing term $\mathbf{D}_{[1]}\langle \Phi(y_1)\Phi(y_i) \rangle|_0 = 0$. For the sake of clarity let us show how the vanishing result appears in the simplest four-point function

$$\begin{aligned}\mathbf{D}_{[1]}\langle \Phi(y_1)\Phi(y_2)\Phi(y_3)\Phi(y_4) \rangle|_0 = &+\langle \Phi(y_3)\Phi(y_4) \rangle \mathbf{D}_{[1]}\langle \Phi(y_1)\Phi(y_2) \rangle|_0 \\ &+ \langle \Phi(y_2)\Phi(y_4) \rangle \mathbf{D}_{[1]}\langle \Phi(y_1)\Phi(y_3) \rangle|_0 \\ &+ \langle \Phi(y_2)\Phi(y_3) \rangle \mathbf{D}_{[1]}\langle \Phi(y_1)\Phi(y_4) \rangle|_0 = 0.\end{aligned} \tag{111}$$

Equation (110) can be generalized to other differential operators: all those that do not involve $\mathbf{D}_{[i]}$ at other points $i \neq 1$. Let us exemplify this for four-point functions,

$$\mathbf{D}_{[\vec{S}]}\langle \Phi(y_1)\Phi^{n_2}(y_2)\Phi^{n_3}(y_3)\Phi^{n_4}(y_4) \rangle|_0 = 0, \qquad [\vec{S}] = [\mathbf{1}], [13\bar{4}], [12\bar{4}], [12\bar{3}]. \tag{112}$$

Moreover for other differential operators the result is not zero but it greatly simplifies. E.g. for $S = [\mathbf{12}], [123\bar{4}], [1\bar{2}]$ the field at position 1 must be contracted with the one at position 2 and no possible contraction $1-3$ and $1-4$ are allowed: the latter are the contractions responsible to double twist contributions, which for these correlation functions are thus absent. In this case we obtain that the correlator is written in terms of a single conformal block

$$\mathbf{D}_{[\vec{S}]}\langle \Phi(y_1)\Phi^{n_2}(y_2)\Phi^{n_3}(y_3)\Phi^{n_4}(y_4) \rangle|_0 = N_{[\vec{S}]}^{n_2,n_3,n_4}\left(\Sigma_{[\vec{S}]}K_{\Delta_i}(x_i)\right)g_{\Delta=-\Delta_{12}\,\ell=0}^{(d)}, \tag{113}$$

$$[\vec{S}] = [\mathbf{12}], [123\bar{4}], [1\bar{2}],$$

where $N_{[\vec{S}]}^{n_2,n_3,n_4}$ is an overall constant and $g_{\Delta=-\Delta_{12}\,\ell=0}^{(d)}(u,v) = u^{-\frac{\Delta_{12}}{2}}v^{\frac{\Delta_{12}-\Delta_{34}}{2}}$. Because of the absence of double twist operators, the correlators (113) behave like the extremal correlators defined in (108).[11]

In all cases (112) and (113) we found that the double twist contributions are annihilated by the differential operators. Below we explain how to exploit this fact to find recurrence relations for the OPE coefficients in the conformal block decomposition of the correlators. In section 5.3 we also show how to use (112) to find the correlators themselves.

---

[9]For the connection of uplifted free theory and random field models see [32].

[10]Notice that $\tilde{\Phi}_{\theta\bar{\theta}} = (d - 2 - 2\Delta_\phi)\omega + \Box\varphi$. This primary is identically zero in the uplift of local free theory because $\Box\varphi = -2\omega$ and $\Delta_\phi = (d-4)/2$. On the contrary it is non vanishing in GFF, where we do not impose the latter constraints. In fact in GFF the operator $\tilde{\Phi}_{\theta\bar{\theta}}$ has a non vanishing two-point function.

[11]E.g. in a simple case $\mathbf{D}_{[12]}\langle \Phi\Phi\Phi^2\Phi^2 \rangle|_0 = -8\Delta_\phi\left(\Delta_\phi + 1\right)\left(-d + 2\Delta_\phi + 2\right)\left(-d + 2\Delta_\phi + 4\right)\left(\Sigma_{[12]}K_{\Delta_i}(x_i)\right)$, which is constant in $u, v$ and only exchanges the identity.

## 5.2 Bootstrapping OPE coefficients

We now show how to use (112) and (113) to find recurrence relations for the OPE coefficients in the conformal block decomposition of $\langle \phi \phi^{n_2} \phi^{n_3} \phi^{n_4} \rangle$. We further show how to solve these recurrence relations when $n_3 = n_4$, giving rise to new closed form expressions for these OPE coefficients.

To start we uplift the Ansatz (107) and act with the operators $\mathbf{D}_{[\vec{S}]}$ with $[\vec{S}] = [\mathbf{1}], [\mathbf{13\bar{4}}], [\mathbf{12\bar{4}}], [\mathbf{12\bar{3}}]$. According to (112) we find

$$
\begin{aligned}
0 &= \left( \Sigma_{[\vec{S}]} K_{\Delta_i}(x_i) \right)^{-1} \mathbf{D}_{[\vec{S}]} \langle \Phi(y_1) \Phi^{n_2}(y_2) \Phi^{n_3}(y_3) \Phi^{n_4}(y_4) \rangle |_0 \\
&= a^{\{n_k\}} D_{[\vec{S}]} g^{(d-2)}_{\Delta=-\Delta_{12}\,\ell=0}(u,v) + \sum_{n,\ell} a^{\{n_k\}}_{n,\ell} D_{[\vec{S}]} g^{(d-2)}_{\Delta=\beta_{12}+2n+\ell\,\ell}(u,v),
\end{aligned}
\tag{114}
$$

where we use the notation $\beta_{12} \equiv \Delta_1 + \Delta_2 = (1 + n_2)\Delta_\phi$. From equation (76) it is easy to see that $D_{[\vec{S}]} g^{(d-2)}_{\Delta=-\Delta_{12}\,\ell=0}(u,v) = 0$ for $[\vec{S}] = [\mathbf{1}], [\mathbf{13\bar{4}}], [\mathbf{12\bar{4}}], [\mathbf{12\bar{3}}]$. On the other hand $D_{[\vec{S}]}$ does not trivially vanish when acting on each double twist conformal block. Miraculously there is a very non trivial cancellation which arises in the sum. Using (76) we find that

$$
\sum_{n,\ell} a^{\{n_k\}}_{n,\ell} \sum_{(i,j) \in P_{[\vec{S}]}} \left( c^{[\vec{S}]}_{i,j} \, \Sigma_{[\vec{S}]} \, g^{(d)}_{\Delta+i\,\ell+j}(u,v) \right)_{\Delta=\beta_{12}+2n+\ell} = 0.
\tag{115}
$$

We set $\Delta = \beta_{12} + 2n + \ell$ because the coefficients $c^{[\vec{S}]}_{i,j}$ are functions of $\Delta$, which should be evaluated to the correct value depending on the exchanged block. Because of the shifts in $i$ and $j$, now the blocks at different values of $n$ and $\ell$ can mix. It is however simple to rearrange the formula above collecting the contributions of $d$-dimensional blocks with given $\Delta$ and $\ell$. Let us do it first for the cases $[\vec{S}] = [\mathbf{1}], [\mathbf{13\bar{4}}]$ (we will see that this can be extended to $[\vec{S}] = [\mathbf{12}], [\mathbf{123\bar{4}}], [\mathbf{1\bar{2}}]$) which have $P_{[\vec{S}]} = P^{(0)}$,

$$
0 = \sum_{n,\ell} \left( a^{\{n_k\}}_{n,\ell} \tilde{c}^{[\vec{S}]}_{0,0} + a^{\{n_k\}}_{n-1,\ell+2} \tilde{c}^{[\vec{S}]}_{0,-2} + a^{\{n_k\}}_{n-1,\ell+1} \tilde{c}^{[\vec{S}]}_{1,-1} + a^{\{n_k\}}_{n-1,\ell} \tilde{c}^{[\vec{S}]}_{2,0} + a^{\{n_k\}}_{n-2,\ell+2} \tilde{c}^{[\vec{S}]}_{2,-2} \right) \Sigma_{[\vec{S}]} g^{(d)}_{\Delta\,\ell} \Bigg|_{\Delta=\beta_{12}+2n+\ell},
\tag{116}
$$

where $a^{\{n_k\}}_{n<0,\ell} \equiv 0$ and the tilded coefficients $\tilde{c}$ are related to $c$ by simple shifts of $\Delta$ and $\ell$, namely

$$
\tilde{c}^{[\vec{S}]}_{i,j} \equiv c^{[\vec{S}]}_{i,j} |_{\Delta \to \Delta-i\,\ell \to \ell-j}.
\tag{117}
$$

Similarly we can write this expansion for the cases $[\vec{S}] = [\mathbf{12\bar{4}}], [\mathbf{12\bar{3}}]$ which have $P_{[\vec{S}]} = P^{(1)}$,

$$
0 = \sum_{n,\ell} \left( a^{\{n_k\}}_{n+1,\ell} \tilde{c}^{[\vec{S}]}_{0,-1} + a^{\{n_k\}}_{n,\ell} \tilde{c}^{[\vec{S}]}_{1,0} + a^{\{n_k\}}_{n-1,\ell+2} \tilde{c}^{[\vec{S}]}_{1,-2} + a^{\{n_k\}}_{n-1,\ell+1} \tilde{c}^{[\vec{S}]}_{2,-1} \right) \Sigma_{[\vec{S}]} g^{(d)}_{\Delta\,\ell} \Bigg|_{\Delta=\beta_{12}+2n+\ell+1}.
\tag{118}
$$

Now let us discuss the cases $[\vec{S}] = [\mathbf{12}], [\mathbf{123\bar{4}}], [\mathbf{1\bar{2}}]$ of equation (113). In practice the argument goes the same way besides that in these cases $D_{[\vec{S}]} g^{(d-2)}_{\Delta=-\Delta_{12}\,\ell=0}(u,v) \propto g^{(d)}_{\Delta=-\Delta_{12}\,\ell=0}(u,v)$. But we can forget about this contribution in (114) and focus on the sum of double traces that vanishes, thus equation (115) still holds for these cases. Now since the blocks in (116) and (118) are linearly independent,[12] in order to get zero the terms in the parentheses must be vanishing. In summary with this simple argument we have found seven recurrence relations

---

[12] If the blocks were linearly dependent, the conformal block decomposition would not be unique. In the case of (116) and (118) it is easy to see that this is not the case by expanding the blocks in powers of the cross ratios.

for $a_{n,\ell}^{\{n_k\}}$ which take the form

$$
\begin{aligned}
&0 = a_{n,\ell}^{\{n_k\}} \tilde{c}_{0,0}^{[\vec{S}]} + a_{n-1,\ell+2}^{\{n_k\}} \tilde{c}_{0,-2}^{[\vec{S}]} + a_{n-1,\ell+1}^{\{n_k\}} \tilde{c}_{1,-1}^{[\vec{S}]} + a_{n-1,\ell}^{\{n_k\}} \tilde{c}_{2,0}^{[\vec{S}]} + a_{n-2,\ell+2}^{\{n_k\}} \tilde{c}_{2,-2}^{[\vec{S}]} \Big|_{\Delta=\beta_{12}+2n+\ell}, & [\vec{S}] = &\begin{matrix}[1],[13\bar{4}],\\[12],[123\bar{4}],[1\bar{2}]\end{matrix}, \\
&0 = a_{n+1,\ell}^{\{n_k\}} \tilde{c}_{0,-1}^{[\vec{S}]} + a_{n,\ell}^{\{n_k\}} \tilde{c}_{1,0}^{[\vec{S}]} + a_{n-1,\ell+2}^{\{n_k\}} \tilde{c}_{1,-2}^{[\vec{S}]} + a_{n-1,\ell+1}^{\{n_k\}} \tilde{c}_{2,-1}^{[\vec{S}]} \Big|_{\Delta=\beta_{12}+2n+\ell+1}, & [\vec{S}] = &[12\bar{4}],[12\bar{3}].
\end{aligned}
\tag{119}
$$

Notice that the coefficients $\tilde{c}_{i,j}^{[\vec{S}]}$ are all known in a closed form as shown in section 4.2. There-fore these relations are very explicit. Let us show a couple of examples for concreteness. A simple recursion can be obtained by summing the relation labelled by $[\vec{S}] = [12\bar{4}]$ with the one of $[\vec{S}] = [12\bar{3}]$. The result is

$$
\begin{aligned}
&\frac{2(\ell+1)(n_3-n_4)\Delta_\phi(-\beta_{12}+d-n-2)(\beta_{12}+\ell+2n-2)(\beta_{12}+\ell+2n-1)(2\beta_{12}-d+2\ell+2n+2)(d-2\Delta_\phi-2n-2)(\Delta_\phi+\ell+n)}{(-\beta_{12}+d-2n-2)(-\beta_{12}+d-2n-1)(\beta_{12}+2\ell+2n-1)(\beta_{12}+2\ell+2n)(2\beta_{12}-d+2\ell+4n)}a_{n-1,\ell+1}^{\{n_k\}} \\
&+\frac{2(\ell+1)(\ell+2)(d+2\ell+2n-2)(-\beta_{12}+d-n-2)(\beta_{12}+\ell+2n-1)(d-2\Delta_\phi-2n-2)}{(d+2\ell-2)(d+2\ell)(-\beta_{12}+d-2n-1)}a_{n-1,\ell+2}^{\{n_k\}} \\
&+\frac{4n(\beta_{12}+\ell+2n-1)(2\beta_{12}-d+2\ell+2n+2)(\Delta_\phi+\ell+n)}{\beta_{12}+2\ell+2n-1}a_{n,\ell}^{\{n_k\}} = 0.
\end{aligned}
\tag{120}
$$

Another nice combination is obtained by subtracting $[\vec{S}] = [12\bar{4}]$ with $[\vec{S}] = [12\bar{3}]$ (we also divide by the factor $2(\ell+1)(\beta_{12}+\ell+2n-1)(d-2\Delta_\phi-2(n+1))$ to shorten the expression, this is normally allowed since we are taking $\Delta_\phi$ to be an arbitrary real number),

$$
\begin{aligned}
&\frac{(-\beta_{12}+d-n-2)(\beta_{12}+\ell+2n-2)(-2\beta_{12}+d-2(\ell+n+1))(\Delta_\phi+\ell+n)\big((-\beta_{12}+d-2(n+1))(\beta_{12}+2(\ell+n))-(n_3-n_4)^2\Delta_\phi^2\big)}{(-\beta_{12}+d-2n-1)(-\beta_{12}+d-2(n+1))(\beta_{12}+2\ell+2n-1)(\beta_{12}+2(\ell+n))(-2\beta_{12}+d-2(\ell+2n))(-2\beta_{12}+d-2(\ell+2n+1))}a_{n-1,\ell+1}^{\{n_k\}} \\
&-\frac{(\ell+2)(n_3-n_4)\Delta_\phi(d+2(\ell+n-1))(-\beta_{12}+d-n-2)}{(d+2\ell-2)(d+2\ell)(-\beta_{12}+d-2n-1)(-\beta_{12}+d-2(n+1))}a_{n-1,\ell+2}^{\{n_k\}} \\
&+\frac{2n(n_3-n_4)\Delta_\phi(-2\beta_{12}+d-2(\ell+n+1))(\Delta_\phi+\ell+n)}{(\ell+1)(\beta_{12}+2\ell+2n-1)(\beta_{12}+2(\ell+n))(-d+2\Delta_\phi+2n+2)}a_{n,\ell}^{\{n_k\}} \\
&+\frac{2n(d+2(\ell+n-1))}{(\beta_{12}+\ell+2n-1)(-d+2\Delta_\phi+2n+2)}a_{n,\ell+1}^{\{n_k\}} = 0.
\end{aligned}
\tag{121}
$$

Similar nice combinations can be defined using the remaining five recurrence relations, but we will omit them here to avoid clutter. One can in principle try to solve this set of seven relations for generic $n_i$. We did not invest enough time in this, however we want to point out that the solution of the recursion is straightforward when $n_3 = n_4$. Indeed in this case the two recursions above greatly simplify (they only involve two terms) and can be solved by[13]

$$
\begin{aligned}
a_{n,\ell}^{n_2,n_3,n_3} =&\, a_0^{n_2,n_3} \frac{\left(\frac{\beta_{12}-1}{2}\right)_{\frac{\ell}{2}}\left(\frac{\beta_{12}}{2}\right)_{\frac{\ell}{2}}\left(\frac{\Delta_\phi}{2}\right)_{\frac{\ell}{2}}\left(\frac{\Delta_\phi+1}{2}\right)_{\frac{\ell}{2}}}{\ell!\left(\frac{\beta_{12}-1}{4}\right)_{\frac{\ell}{2}}\left(\frac{\beta_{12}+1}{4}\right)_{\frac{\ell}{2}}} \\
&\times \frac{(-d+\beta_{12}+3)_n\left(-\frac{d}{2}+\Delta_\phi+2\right)_n\left(\frac{\ell+\beta_{12}-1}{2}\right)_n\left(\frac{\ell+\beta_{12}}{2}\right)_n(\ell+\Delta_\phi)_n\left(-\frac{d}{2}+\ell+\beta_{12}+1\right)_n}{2^{4n}n!\left(\frac{d}{2}+\ell-1\right)_n\left(\frac{-d+\beta_{12}+3}{2}\right)_n\left(\ell+\frac{\beta_{12}}{2}-\frac{1}{2}\right)_n\left(\frac{-d+2\ell+2\beta_{12}+2}{4}\right)_n\left(\frac{-d+2\ell+2\beta_{12}+4}{4}\right)_n}.
\end{aligned}
\tag{123}
$$

Here we solved the recursion only for $\ell$ even since the equality $n_3 = n_4$ selects only even spin operators in the OPE. Notice that $n_2$ enters in the definition of $\beta_{12} = (1+n_2)\Delta_\phi$, while $n_3$ only

---

[13]The recurrence relations take the following form which can be trivially solved

$$
A_n = A_{n-1}\prod_{i=1}^{j}(p_i+q_i n)^{r_i} \implies A_n = A_0\prod_{i=1}^{j}\left[q_i^n\left(\frac{p_i}{q_i}+1\right)_n\right]^{r_i},
\tag{122}
$$

for any set of constants $p_i, q_i, r_i$ and any $j \in \mathbb{N}$. The second line of (123) comes by solving (5.2) in $n$ while the first line comes by solving (120) in $\ell$ (using the dependence in $n$ just computed).

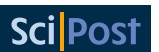

appears in the overall constant $a_0^{n_2,n_3}$, which is the seed of the recursion for $\ell = 0$ and $n = 0$. We checked that this solution automatically satisfies the remaining five recurrence relations.

Interestingly we found in a closed form all the OPEs of double traces exchanges of any correlator $\langle \phi \phi^{n_2} \phi^{n_3} \phi^{n_3} \rangle$ without ever computing any correlation function. The expression is given up to an overall constant which is the seed of the recursion and can be fixed from explicit computations of the simplest double twist exchange, $\phi^{n_2+1}$. Indeed the computation of such coefficient is a simple combinatorial exercise which is solved by[14]

$$ a_0^{n_2,n_3} = n_3! \binom{n_2+1}{\frac{n_2+1}{2}} \left( \frac{1-n_2+2n_3}{2} \right)_{\frac{n_2+1}{2}}, \tag{124} $$

where $n_2$ must be even, otherwise the correlator vanishes. We also notice that the correlator is extremal when $n_2 = 1+2n_3$ and thus $a_0^{n_2,n_3} = 0$ for all $n_2 \geq 1+2n_3$ as it should. The other OPE coefficient which is missing in the Ansatz (107) is the one of the exchange of $\phi^{n_2-1}$ which can also be trivially computed by combinatorics $a^{n_2,n_3,n_3} = n_2 a_0^{n_2-2,n_3}$, indeed the combinatorics is the same as the one of $a_0^{n_2,n_3}$ but with two less fields $n_2$, and adding an extra factor of $n_2$ to take into account the left OPE $\phi \times \phi^{n_2} \sim n_2 \phi^{n_2-1}$.

As a consistency check, for the case of the four-point function of $\langle \phi \phi \phi \phi \rangle$ these coefficients were computed in [33]; it is easy to see that our solution matches theirs once we restrict to $n_2 = 1$ and we shift $d \to d + 2$ (recall that the original GFF leaves in two less dimension).

In summary this method solves for the OPE coefficients that are harder to compute (the ones of double twist operators which contain derivatives), while the remaining ones can be obtained from a one-line combinatoric computation. Moreover, at a more abstract level, PS SUSY provides a novel way to understand why double twist families (in GFF) are tied together: one cannot simply modify one of the OPE coefficients in the family without violating the recursion relations provided by supersymmetry.

## 5.3 Bootstrapping correlators and extension to perturbation theory

In the previous section we found a sophisticated way to extract information from supersymmetry. There is also another simple way to make use of these constraints. Indeed in (112) we found that a differential operator was annihilating the correlators, therefore we obtained a differential equation that the correlators must satisfy. Below we show how to use this information to fix the form of the GFF correlators $\langle \phi \phi^{n_2} \phi^{n_3} \phi^{n_4} \rangle$ up to a few constants. Moreover we will see that this logic can be also used to bootstrap the form of correlation functions in perturbation theory around GFF.

Let us first see how to use (112) to fix the correlation function. We require that the GFF correlators are written as finite sums of powers $u^a v^b$. This is always true if the correlator contains a finite number of fields and derivatives. Then we notice that the differential operators $D_{[\vec{S}]}$ acting on $u^a v^b$ produce a sum of terms with different powers, e.g. $D_{[\mathbf{1}]} u^a v^b$ is a linear combination of $u^a v^b, u^{a+1} v^b, u^a v^{b+1}$. Because of this reason and the fact that the correlator is written as a finite sum of powers, the action $D_{[\vec{S}]}$ must annihilate separately each power. By requiring

$$ D_{[\vec{S}]} u^a v^b = 0, \tag{125} $$

we obtain a constraint on the possible powers $a$ and $b$. If we apply this logic to $\langle \phi \phi^{n_2} \phi^{n_3} \phi^{n_4} \rangle$ using e.g. $[\vec{S}] = [\mathbf{1}]$ we find that there are only three allowed powers and thus the correlator

---

[14]The rationale is the following: we need to find $\phi^{n_2+1}$ in the OPE of $\phi^{n_3} \times \phi^{n_3}$, we thus choose $\frac{n_2+1}{2}$ fields out of the $n_3$ of both $\phi^{n_3}$ fields, which gives $\binom{n_3}{\frac{n_2+1}{2}}^2$ (these form the field $\phi^{n_2+1}$), the remaining $n_3 - \frac{n_2+1}{2}$ fields of each $\phi^{n_3}$ should be contracted giving a $(n_3 - \frac{n_2+1}{2})!$. Moreover we need to introduce an extra factor of $(n_2 + 1)!$ to account for the normalization of the operator $\phi^{n_2+1}$. Putting the factors together and simplifying the expression, we obtain (124).

must take the form

$$\langle \phi(x_1)\phi^{n_2}(x_2)\phi^{n_3}(x_3)\phi^{n_4}(x_4)\rangle = K_{\Delta_i}(x_i)\left[ k_1 u^{\frac{n_2-1}{2}\Delta_\phi} v^{-\frac{n_2+n_3-n_4-1}{2}\Delta_\phi} \right.$$
$$\left. + k_2 u^{\frac{n_2+1}{2}\Delta_\phi} v^{-\frac{n_2+n_3-n_4-1}{2}\Delta_\phi} + k_3 u^{\frac{n_2+1}{2}\Delta_\phi} v^{-\frac{n_2+n_3-n_4+1}{2}\Delta_\phi} \right], \tag{126}$$

were $k_i$ are undetermined coefficients. In this case the computation of the four-point function is actually quite simple, indeed by some combinatorics it is easy to find that the four-point function indeed matches the form above and that the coefficients are[15]

$$k_1 = n_2 C_{n_2-1,n_3,n_4}, \qquad k_2 = n_3 C_{n_2,n_3-1,n_4}, \qquad k_3 = n_4 C_{n_2,n_3,n_4-1}, \tag{127}$$

where we defined $C_{m_1,m_2,m_3} \equiv \frac{m_1! m_2! m_3!}{\left(\frac{m_1+m_2-m_3}{2}\right)! \left(\frac{m_1-m_2+m_3}{2}\right)! \left(\frac{-m_1+m_2+m_3}{2}\right)!}$.

In this example we were able to obtain the Ansatz (126) without doing any Wick contraction. This may not seem a very impressive result since the direct computation of the four-point function is quite straightforward. However this idea can be applied to more generic cases, where the computation is more involved as we show below.

Let us give an example of how to use this same logic for perturbative computations. We consider the one-loop integral of a $\phi^4$ perturbation around GFF,

$$F(x_i) \equiv \int d^{\hat{d}}x_0 \langle \phi(x_1)\phi(x_2)\phi(x_3)\phi(x_4)\phi^4(x_0)\rangle = \int d^{\hat{d}}x_0 \prod_{i=1}^{4} |x_{i0}|^{-2\Delta_\phi}. \tag{128}$$

Instead of trying to compute the integral in $\mathbb{R}^{\hat{d}}$ we uplift the correlator to $\mathbb{R}^{d|2}$,

$$F(y_i) \equiv \int d^d x_0 d\theta_0 d\bar{\theta}_0 \langle \Phi(y_1)\Phi(y_2)\Phi(y_3)\Phi(y_4)\Phi^4(y_0)\rangle. \tag{129}$$

The fermionic part of the integral has the effect of taking the highest component of $\Phi^4$ which is written in terms of $\Phi_\theta \Phi_{\bar\theta} \Phi_0^2$ and $\Phi_{\theta\bar\theta} \Phi_0^3$. Moreover can rewrite $\Phi_{\theta\bar\theta}$ in terms of the primary $\tilde{\Phi}_{\theta\bar\theta}$ and $\partial_x^2 \Phi_0$. So $F(y_i)$ is obtained as a linear combination of the following five-point functions integrated over $x_0$,

$$\langle \Phi\Phi\Phi\Phi[\tilde{\Phi}_{\theta\bar\theta}\Phi_0^3](x_0)\rangle, \qquad \langle \Phi\Phi\Phi\Phi[(\partial_x^2\Phi_0)\Phi_0^3](x_0)\rangle, \qquad \langle \Phi\Phi\Phi\Phi[\Phi_\theta\Phi_{\bar\theta}\Phi_0^2](x_0)\rangle, \tag{130}$$

where we suppressed the position $y_1, y_2, y_3, y_4$ of the first four $\Phi$ fields to shorten the notation. Looking at the correlators above, just from factorization and the fact that two-point functions of different primaries are diagonal, we see that they are annihilated every time we act with at least two differential operators of the form $\mathbf{D}_{[\mathbf{i}]}$ or $\mathbf{D}_{[i\bar{j}]}$, namely

$$\mathbf{D}_{[\vec{S}]}F(y_i)|_0 = 0, \qquad [\vec{S}] = [\mathbf{ij}], [\mathbf{i}j\bar{k}], [i\bar{j}k\bar{l}], [\mathbf{ijk}], [\mathbf{1234}], \tag{131}$$

where $i, j, k, l$ are all different and run over $1, 2, 3, 4$.

The same logic can be generalized to the correlator of $n$ different generalized free bosons $\phi_i$ with dimension $\Delta_i$, coupled by the interaction $\phi_1 \cdots \phi_n$ where at one loop we find

$$F_{\Delta_1,\dots,\Delta_n}(x_i) \equiv \int d^{\hat{d}}x_0 \langle \phi_1(x_1)\dots\phi_n(x_n)[\phi_1\cdots\phi_n](x_0)\rangle = \int d^{\hat{d}}x_0 \prod_{i=1}^{n} |x_{i0}|^{-2\Delta_{\phi_i}}. \tag{132}$$

---

[15]The OPE coefficients in the previous section are related to these coefficients as $a_0^{n_2,n_3} = 2n_3 C_{n_2,n_3,n_3-1}$ and $a^{n_2,n_3,n_3} = n_2 C_{n_2-1,n_3,n_3}$.

Again the correlator can be uplifted to superspace and it is easy to see that for the same reasons of above, it should satisfy a large set of differential equations. In particular by acting with two or more differential operators of the type $\mathbf{D}_{[\mathbf{i}]}$ or $\mathbf{D}_{[j\bar{k}]}$ the uplifted correlator is annihilated.

Let us now focus of the case $\hat{d} = \sum_{i=1}^{n} \Delta_i$, when the integral is conformal [34]. We further consider a four-point function where by conformal symmetry we can write

$$F_{\Delta_1,\Delta_2,\Delta_3,\Delta_4}(x_i) = K_{\Delta_i}(x_i) f_{\Delta_1,\Delta_2,\Delta_3,\Delta_4}(u,v), \tag{133}$$

where $K_{\Delta_i}$ is the kinematic factor of (5). The set of equations (131) can then be written directly in terms of $u$ and $v$ using the differential operator $D_{[\vec{S}]}$,

$$D_{[\vec{S}]} f_{\Delta_1,\Delta_2,\Delta_3,\Delta_4}(u,v) = 0, \qquad [\vec{S}] = [\mathbf{ij}], [ij\bar{k}], [i\bar{j}k\bar{l}], [\mathbf{ijk}], [\mathbf{1234}]. \tag{134}$$

These types of integrals are very common in the literature. Indeed $F$ computes the so called $D$-function which defines a contact Witten diagram in AdS (while $f$ computes the $\bar{D}$-function) see for example [23]).[16]

The fact that $D$-functions satisfy many relations is a well known fact in the literature. These can be obtained by manipulating the integrals (e.g. by taking derivatives in the propagators) and re-expressing the result as $D$-functions with shifted $\Delta_i$, see for example formulae (C.4), (C.5) and (C.7) in [23]. Using the latter formulae we could prove that (134) is indeed correct (the proof is included in the Mathematica file attached to the publication). It is however interesting that for us the relations (134) came without doing any computation. In practice it was sufficient to notice that the integral in $\theta_0, \bar{\theta}_0$ cannot create more than one term $\tilde{\Phi}_{\theta\bar{\theta}}$ or one term $\Phi_\theta \Phi_{\bar{\theta}}$, which automatically implies that the external operators have vanishing Wick contractions with the vertex.

Let us also mention that the relations (134) impose strong constraint on the shape of the correlator. For example let us show what happens in the simplest case of $\Delta_i = 1$ and $\hat{d} = 4$, where the form of the $\bar{D}$ function is known [23]

$$f_{1,1,1,1}(u,v) \propto \frac{z\bar{z}}{z-\bar{z}} \left( \log\left( \frac{1-\bar{z}}{1-z} \right) \log(z\bar{z}) + 2\mathrm{Li}_2(\bar{z}) - 2\mathrm{Li}_2(z) \right). \tag{136}$$

First it is easy to check that (136) is indeed annihilated by all the differential operators in (134). Moreover we found that (134) can be used to bootstrap the correlator. In particular one can consider the Ansatz made by a generic linear combination of the three terms $\frac{z\bar{z}}{z-\bar{z}}\{\log\left(\frac{1-\bar{z}}{1-z}\right)\log(z\bar{z}), \mathrm{Li}_2(\bar{z}), \mathrm{Li}_2(z)\}$ in (136).[17] We noticed that requiring that the Ansatz is annihilated by any of $D_{[1\bar{2}3\bar{4}]}$, $D_{[1\bar{2}4\bar{3}]}$ or $D_{[1\bar{3}2\bar{4}]}$ (since $D_{[1\bar{2}3\bar{4}]} - D_{[1\bar{2}4\bar{3}]} - D_{[1\bar{3}2\bar{4}]} = 0$, these are just two independent differential equations) is sufficient to fix the relative coefficients of the three terms.

This section was a very basic sample of applications of the idea of bootstrapping correlators using PS supersymmetry, which should be understood more as a proof of concept. One could extend this to more sophisticated cases where the final answer is not known and leverage PS SUSY to obtain new results. We leave this direction for future investigations.

---

[16]Let us match our conventions with the definition of $D$ and $\bar{D}$ in [23],

$$F_{\Delta_1,...,\Delta_n}(x_i) = \frac{2\pi^{\hat{d}/2}}{\Gamma(\sum_{i=1}^{n} \frac{\Delta_i}{2} - \frac{\hat{d}}{2})} D_{\Delta_1,...,\Delta_n}(x_i), \qquad f_{\Delta_1,\Delta_2,\Delta_3,\Delta_4}(u,v) = \frac{\pi^{\hat{d}/2} u^{\Delta_1+\Delta_2}}{\prod_{i=1}^{4} \Gamma(\Delta_i)} \bar{D}_{\Delta_1,\Delta_2,\Delta_3,\Delta_4}(u,v). \tag{135}$$

[17]This Ansatz can be motivated by the degree of trascendentality of the functions appearing in a one-loop integral. Of course one could find a more restrictive Ansatz or even the full solution $f_{1,1,1,1}$ by requiring that it satisfies extra conditions (e.g. we could impose that $f_{1,1,1,1}$ is invariant under $z \to \bar{z}$ and/or is a single-valued function of the cross ratios). Here we only wanted to show that (134) is also a powerful condition.

# 6 The 4d Parisi-Sourlas uplift of diagonal minimal models

The minimal models are an infinite set of solvable two-dimensional CFTs. They can be completely fixed through bootstrap constraints which yield the exact spectrum of conformal dimensions and OPE coefficients. Their correlation functions can be also computed exactly giving rise to the best known examples of interacting CFTs. In the following we will start by a quick introduction to these models following [35] and then we will give an argument for why these should have a Parisi-Sourlas uplift to four dimensions. In the next subsection we will further investigate the uplift of the know correlation functions and show that they do not possess any inconsistency.

Minimal models are defined by requiring that their spectrum only contains a finite number of Virasoro primaries. They are denoted as $\mathcal{M}(p, p')$ and are labelled by two coprime integers $p, p'$ with $p, p' \geq 2$ which determine the central charge [36]

$$c = 1 - 6\frac{(p - p')^2}{pp'}. \tag{137}$$

The holomorphic weight of the Virasoro primaries is labelled by

$$h_{r,s} = \frac{(pr - p's)^2 - (p - p')^2}{4pp'}, \tag{138}$$

with $1 \leq r \leq p' - 1$ and $1 \leq s \leq p - 1$. We will be mostly interested in the diagonal minimal models defined by $|p - p'| = 1$, where all Virasoro primaries have identical holomorphic and antiholomorphic weights, namely they are labelled by $h = \bar{h} = h_{r,s}$ or equivalently $\Delta = 2h_{r,s}, \ell = 0$.

Minimal models are known [37] (see also section 7.4.7 of [35]) to have an effective description in terms of the following Landau-Ginzburg Lagrangian,

$$S = \int d^{\hat{d}}x \, \frac{1}{2}(\partial^a \phi)^2 + V(\phi), \tag{139}$$

where $\hat{d} = 2$. By choosing a $\mathbb{Z}_2$ invariant polynomial potential $V(\phi)$ of degree $2m$ ($m = 2, 3, \dots$) we obtain a multicritical system in the universality class defined by the diagonal minimal models $(m + 2, m + 1)$ where $m = 2$ is Ising, $m = 3$ is the tricritical, etc. Of course it is hard to use this action for actual computation since there is no small coupling. Typically one performs computation in $\epsilon$-expansion (namely one works in $\hat{d} = d_{uc} - \epsilon$ dimensions where $d_{uc} = 2m/(m-1)$ is the upper critical dimension) such that the interaction $\phi^{2m}$ is weakly relevant. One can then tune all the strongly relevant perturbation $\phi^{2i}$ with $i < m$ and study the IR fixed point in perturbation theory for small $\epsilon$. To recover the 2-dimensional results one must then set $\epsilon = d_{uc} - 2$. Similarly by introducing also $\mathbb{Z}_2$-odd terms in the potential one can study a class of non-unitary minimal models, e.g. the cubic potential is in the same universality class as the Lee-Yang minimal model which corresponds to $\mathcal{M}(5, 2)$.

The Lagrangian description (139) is not particularly useful to compute observables since minimal models can be solved exactly through bootstrap methods. On the other hand this formulation has a clear benefit for our purposes since it can be easily uplifted to $d = \hat{d} + 2$ dimension as we showed in section 2.5. Indeed the action (139) can be trivially uplifted to

$$S = \int d^d x \, d\theta \, d\bar{\theta} \, \frac{1}{2}(\partial^a \Phi)^2 + V(\Phi). \tag{140}$$

One can consider this action in $d = 4$ which reduces to (139) in $\hat{d} = 2$, where both actions are strongly coupled.[18] Alternatively it is also possible to consider (140) in $d = d_{uc} + 2 - \epsilon$

---

[18]Notice that when written in components the action becomes (34), where one can easily see that all polynomial interactions are relevant since $[\varphi] = 0$.

dimensions which corresponds to the uplift the $\epsilon$-expansion of the model (139). Since the superspace action (140) dimensionally reduces to the $\hat{d} = d - 2$ action (139) for any $\hat{d}$, then the $\epsilon$-expansion of (140) matches the one of (139) for every $\epsilon$. Therefore it is natural to assume that $\epsilon$-expansion of (140) when continued to $d = 4$, dimensionally reduce to minimal models. The action (140) in $d = 4$ therefore can be thought as a definition for the Parisi-Sourlas uplift of the diagonal minimal models. While this Lagrangian description of the uplift is not particularly useful (as it was the case for the usual minimal models), at least it tells us that the uplift of the diagonal minimal models should exist. This argument is also valid for the non-unitary minimal models which have a Lagrangian description (like Lee-Yang) and even for the Liouville model which is also defined through a scalar field action. In the following we assume that indeed the Parisi-Sourlas uplift of the diagonal minimal models exists, and we see if any problem arises by looking at the concrete example of the Ising minimal model.

## 6.1 Example: Parisi-Sourlas uplift of the Ising minimal model

In this section we test the dimensional uplift on the explicit correlators of the $2d$ critical Ising model. This model contains the identity and two scalar Virasoro primaries called $\sigma$ and $\epsilon$, which respectively have conformal dimensions $\Delta = \frac{1}{8}$ and $\Delta = 1$. The model has a $\mathbb{Z}_2$ symmetry under which $\sigma$ is odd while $\epsilon$ and the identity are even.

All correlators of Virasoro primaries can be computed in a closed form, see e.g. [38]. In particular the four-point functions in the notation (4) take the form

$$
\begin{aligned}
f_{\sigma\sigma\sigma\sigma} &= \left| \frac{1}{(1-\rho^2)^{1/4}} \right|^2 + \left| \frac{\sqrt{\rho}}{(1-\rho^2)^{1/4}} \right|^2, & f_{\sigma\sigma\epsilon\epsilon} &= \left| \frac{1+\rho^2}{1-\rho^2} \right|^2, \\
f_{\sigma\epsilon\epsilon\sigma} &= \left| \frac{\rho^{1/16}(1+6\rho+\rho^2)}{2^{7/8}(1-\rho)^2(1+\rho)^{1/8}} \right|^2, & f_{\epsilon\epsilon\epsilon\epsilon} &= \left| \frac{1+14\rho^2+\rho^4}{(1-\rho^2)^2} \right|^2,
\end{aligned}
\tag{141}
$$

where we wrote $f$ in terms of the cross ratios $\rho, \bar{\rho}$ defined in (6) to get more compact expressions. The subscripts of $f$ specify the correspondent four-point functions.

Being four-point functions of scalar operators, all correlators in (141) can be trivially up-lifted to $4d$ using the prescription (39). Moreover, as explained in section 3, for each of these four-point functions we obtain 43 different four-point functions in four dimensions, which can be simply computed using the differential operators $D_{[\vec{S}]}$ defined in (57). As an example we present the action of these differential operators on the correlator of four $\epsilon$. We start by showing in table 3 the components[19] in $P^{(0)}$. In table 3 we show all the 16 components in $P^{(1)}$. Finally let us show the remaining single component in $P^{(2)}$ which takes the form $D_{[1\bar{3}2\bar{4}]}f_{\epsilon\epsilon\epsilon\epsilon} = 4uv^{-2}(v-1)\left(u^2-u(v+1)+v^2+1\right)$. These results show that for any given four-point function it is straightforward to compute all its 43 components. Of course many of the entries in tables 3 and 4 are redundant since they could be obtained by permutations of the external operators. However we decided to present all the components in order to show that they are compatible with crossing. E.g. the component $[\mathbf{1}]$ and $[\mathbf{3}]$ are related by the crossing $1 \leftrightarrow 3$ which amounts to take the component $[\mathbf{1}]$, change $u \leftrightarrow v$ and multiply by $u^{\frac{1}{2}(\Delta_1+\Delta_2)}v^{\frac{1}{2}(-\Delta_2-\Delta_3)} = u^2 v^{-1}$.

We did repeat this exercise for the other correlators but we do not show the results here because the expression are lengthy, however we stress that this can be easily done in all cases using the differential operators $D_{[\vec{S}]}$ computed in section 3.3.

---

[19]The component $[\mathbf{1234}]$ did not fit the table so for cosmetic reasons we present it here:

$$
\begin{aligned}
D_{[\mathbf{1234}]}f_{\epsilon\epsilon\epsilon\epsilon} =4^5 v^{-3}\big[&\left(8u^2+9u+8\right)v^4-\left(4u^3+u^2+u+4\right)v^3+\left(8u^4-u^3-u+8\right)v^2 \\
&-\left(10u^5-9u^4+u^3+u^2-9u+10\right)v+2(u-1)^2\left(2u^4-u^3-u+2\right)-10(u+1)v^5+4v^6\big]
\end{aligned}
$$

Table 3: For the components $[\vec{S}] \in P^{(0)}$ we present the explicit action $D_{[\vec{s}]}f_{\epsilon\epsilon\epsilon\epsilon}$.

| $[\vec{S}]$ | $D_{[\vec{s}]}f_{\epsilon\epsilon\epsilon\epsilon}$ |
|---|---|
| [ ] | $v^{-1}\big(u^2-uv-u+v^2-v+1\big)$ |
| [**1**] | $4v^{-1}\big(u^2(v+1)-u^3+u\big(v^2+1\big)-(v-1)^2(v+1)\big)$ |
| [**2**] | $4v^{-2}\big(u^2(v+1)-u^3+u\big(v^2+1\big)-(v-1)^2(v+1)\big)$ |
| [**3**] | $4v^{-2}\big(u^2(v+1)-u^3+u\big(v^2+1\big)-(v-1)^2(v+1)\big)$ |
| [**4**] | $4v^{-1}\big(u^2(v+1)-u^3+u\big(v^2+1\big)-(v-1)^2(v+1)\big)$ |
| [1$\bar{2}$] | $2v^{-1}\big(u^2-v^2+v-1\big)$ |
| [3$\bar{4}$] | $2v^{-1}\big(u^2-v^2+v-1\big)$ |
| [1$\bar{2}$3$\bar{4}$] | $-4v^{-2}(u-v)((u-1)u+(v-1)v+1)$ |
| [1$\bar{2}$4$\bar{3}$] | $-4v^{-1}(-1+u)(1+u^2-(1+u)v+v^2)$ |
| [13$\bar{4}$] | $-8v^{-1}\big(-u^2(v+1)+u^3+u\big(v^2+1\big)-(v-1)^2(v+1)\big)$ |
| [23$\bar{4}$] | $-8v^{-2}\big(-u^2(v+1)+u^3+u\big(v^2+1\big)-(v-1)^2(v+1)\big)$ |
| [31$\bar{2}$] | $-8v^{-2}\big(-u^2(v+1)+u^3+u\big(v^2+1\big)-(v-1)^2(v+1)\big)$ |
| [41$\bar{2}$] | $-8v^{-1}\big(-u^2(v+1)+u^3+u\big(v^2+1\big)-(v-1)^2(v+1)\big)$ |
| [**12**] | $16v^{-2}\big(u^2\big(v^2+1\big)-u^3(v+1)+u^4+u\big(-3v^3+v^2+v-3\big)+2(v-1)^2\big(v^2+1\big)\big)$ |
| [**13**] | $16v^{-2}\big(u^2\big(4v^2+v+1\big)-u^3(4v+3)+2u^4+u\big(-4v^3+v^2-1\big)+(v-1)^2\big(2v^2+v+1\big)\big)$ |
| [**14**] | $16v^{-1}\big((u^2+1)v^2+\big(-3u^3+u^2+u-3\big)v+2(u-1)^2\big(u^2+1\big)-(u+1)v^3+v^4\big)$ |
| [**23**] | $16v^{-3}\big((u^2+1)v^2+\big(-3u^3+u^2+u-3\big)v+2(u-1)^2\big(u^2+1\big)-(u+1)v^3+v^4\big)$ |
| [**24**] | $16v^{-2}\big(u^2\big(4v^2+v+1\big)-u^3(4v+3)+2u^4+u\big(-4v^3+v^2-1\big)+(v-1)^2\big(2v^2+v+1\big)\big)$ |
| [**34**] | $16v^{-2}\big(u^2\big(v^2+1\big)-u^3(v+1)+u^4+u\big(-3v^3+v^2+v-3\big)+2(v-1)^2\big(v^2+1\big)\big)$ |
| [**12**3$\bar{4}$] | $-32v^{-2}\big(u^2\big(v^2+1\big)+u^3(v+1)-u^4+u\big(-3v^3+v^2+v-3\big)+2(v-1)^2\big(v^2+1\big)\big)$ |
| [1$\bar{2}$**34**] | $-32v^{-2}\big(u^2\big(v^2+1\big)+u^3(v+1)-u^4+u\big(-3v^3+v^2+v-3\big)+2(v-1)^2\big(v^2+1\big)\big)$ |
| [**123**] | $-256v^{-3}\big(u^2-u(v+1)+(v-1)v+1\big)\big(-u^2(v+1)+u^3-u\big(v^2+1\big)+(v-1)^2(v+1)\big)$ |
| [**124**] | $-256v^{-2}\big(u^2-u(v+1)+(v-1)v+1\big)\big(-u^2(v+1)+u^3-u\big(v^2+1\big)+(v-1)^2(v+1)\big)$ |
| [**134**] | $-256v^{-2}\big(u^2-u(v+1)+(v-1)v+1\big)\big(-u^2(v+1)+u^3-u\big(v^2+1\big)+(v-1)^2(v+1)\big)$ |
| [**234**] | $-256v^{-3}\big(u^2-u(v+1)+(v-1)v+1\big)\big(-u^2(v+1)+u^3-u\big(v^2+1\big)+(v-1)^2(v+1)\big)$ |

Notice also that the expressions (141) determine all correlators of Virasoro primaries, but from these it is also possible to obtain the correlators global primaries by studying descendants under Virasoro. In particular we could build an infinite set of scalar global primaries by acting on each of these correlators with some appropriate choices of Virasoro generators (see appendix C). All four-point functions of scalar global primaries can be equally trivially uplifted to 4$d$ and for any such correlator it is straightforward to obtain the 43 components as above.

From this exercise we therefore conclude that we know a huge amount of information for the uplifted Ising minimal model. In this model we can in principle obtain all possible correlators of global scalar primaries and we can easily uplift any of them and obtain all its components as shown above. All components will automatically transform correctly under crossing, respect supersymmetry, and —because of equation (76)— they will also have a good four-dimensional conformal block decomposition. In what follows we will show some explicit conformal blocks decompositions of the uplifts of the Ising correlators.

## 6.2 Conformal block decomposition of the uplifted correlators

We now consider some four-point functions of the uplifted Ising minimal model and show how they decompose in $d = 4$ conformal blocks. In principle this exercise is automatic since it is

Table 4: For all components $[\vec{S}] \in P^{(1)}$ we show the explicit action $D_{[\vec{S}]} f_{\epsilon\epsilon\epsilon\epsilon}$.

| $[\vec{S}]$ | $D_{[\vec{S}]} f_{\epsilon\epsilon\epsilon\epsilon}$ |
|---|---|
| $[1\bar{3}]$ | $-2\sqrt{u}v^{-1}\left(u^2 - uv + v^2 - 1\right)$ |
| $[1\bar{4}]$ | $2\sqrt{u}v^{-1}\left(-u^2 + u + v^2 - 1\right)$ |
| $[2\bar{3}]$ | $-2\sqrt{u}v^{-2}\left((u-1)u - v^2 + 1\right)$ |
| $[2\bar{4}]$ | $-2\sqrt{u}v^{-1}\left(u^2 - uv + v^2 - 1\right)$ |
| $[12\bar{3}]$ | $8\sqrt{u}v^{-2}\left(-u^2(v+1) + u^3 + u\left(v^2 - 1\right) - (v-1)\left(v^2 + 1\right)\right)$ |
| $[12\bar{4}]$ | $8\sqrt{u}v^{-1}\left(-u^2(v+1) + u^3 - uv^2 + u + (v-1)\left(v^2 + 1\right)\right)$ |
| $[21\bar{3}]$ | $8\sqrt{u}v^{-2}\left(-u^2(v+1) + u^3 - uv^2 + u + (v-1)\left(v^2 + 1\right)\right)$ |
| $[21\bar{4}]$ | $8\sqrt{u}v^{-2}\left(-u^2(v+1) + u^3 + u\left(v^2 - 1\right) - (v-1)\left(v^2 + 1\right)\right)$ |
| $[31\bar{4}]$ | $8\sqrt{u}v^{-2}\left(-u^2(v+1) + u^3 + u\left(v^2 - 1\right) - (v-1)\left(v^2 + 1\right)\right)$ |
| $[32\bar{4}]$ | $8\sqrt{u}v^{-2}\left(-u^2(v+1) + u^3 - uv^2 + u + (v-1)\left(v^2 + 1\right)\right)$ |
| $[41\bar{3}]$ | $8\sqrt{u}v^{-1}\left(-u^2(v+1) + u^3 - uv^2 + u + (v-1)\left(v^2 + 1\right)\right)$ |
| $[42\bar{3}]$ | $8\sqrt{u}v^{-2}\left(-u^2(v+1) + u^3 + u\left(v^2 - 1\right) - (v-1)\left(v^2 + 1\right)\right)$ |
| $[132\bar{4}]$ | $-32\sqrt{u}v^{-2}\left(u^2\left(4v^2 + v + 1\right) - u^3(4v + 3) + 2u^4 + u\left(-4v^3 + v^2 + 1\right) + (v-1)\left((2v-1)v^2 + 1\right)\right)$ |
| $[142\bar{3}]$ | $-32\sqrt{u}v^{-2}\left(\left(u^2 + 1\right)v^2 + \left(-3u^3 + u^2 + u - 3\right)v + 2(u-1)^2\left(u^2 + 1\right) + (u+1)v^3 - v^4\right)$ |
| $[231\bar{4}]$ | $-32\sqrt{u}v^{-3}\left(\left(u^2 + 1\right)v^2 + \left(-3u^3 + u^2 + u - 3\right)v + 2(u-1)^2\left(u^2 + 1\right) + (u+1)v^3 - v^4\right)$ |
| $[241\bar{3}]$ | $-32\sqrt{u}v^{-2}\left(u^2\left(4v^2 + v + 1\right) - u^3(4v + 3) + 2u^4 + u\left(-4v^3 + v^2 + 1\right) + (v-1)\left((2v-1)v^2 + 1\right)\right)$ |

easy to compute the decomposition in two-dimensional conformal blocks (see appendix C.1) and for each such block one can use formula (76) to obtain the four-dimensional counterpart. However we find it useful to exemplify what happens in a few instances in order to discuss some features of the theory. As we explained above, there are in principle infinitely many correlators of scalar primary operators which we could consider and for each of them we could further study 43 components. For the following examples we find it enough to only consider the uplift of all four-point functions of Virasoro primaries. Moreover we will restrict our attention to their lowest components, which in practice are exactly the ones in (141). In appendix D we further show the decomposition of a few higher components of the four-point function of $\epsilon$.

**Correlator $\epsilon\epsilon\epsilon\epsilon$**

In $\hat{d} = 2$ the correlation function of four $\epsilon$ only exchanges a single Virasoro block associated to the identity, which can be written in terms of infinitely many global conformal primaries which contain all spin $\ell$ conserved currents (with $\ell$ even, since the odd $\ell$ operators are never exchanged in the OPE of two equal operators). Let us see how is the $d = 4$ counterpart of these exchanges

$$f_{\epsilon\epsilon\epsilon\epsilon} = \sum_{\Delta = 2\mathbb{N}_{\geq 0}} \sum_{\ell = 0,2,\dots,\Delta} a_{\Delta\ell}\, g_{\Delta\ell}^{(d=4)}, \tag{142}$$

where the conformal blocks have $\Delta_{12} = \Delta_{34} = 0$. To make this formula more transparent in table 5 we explicitly list all the coefficients up to $\Delta \leq 10$. As a first comment, we notice that for this correlator $a_{\Delta\ell}$ are just square of OPE coefficients. In a unitary theory $a_{\Delta\ell}$ should be always positive, while here we also find negative values, which signals the non unitarity of the theory. This will be a recurrent feature also of the next decompositions.

Let us spell out some of the operators with interesting features. The first operator is simply the identity with $\Delta = 0$ and spin zero. Another important operator is the stress tensor with $\Delta = 4$ and $\ell = 2$. We notice that it is accompanied by a spin 2 lower dimensional counterpart with $\Delta = 2$, which lies below the unitarity bounds. Indeed the stress tensor multiplet takes

Table 5: Decomposition of $\epsilon\epsilon\epsilon$: All coefficients for exchanged operators with dimension $\Delta \le 10$.

| $(\Delta,\ell)$ | (0,0) | (2,0) | (2,2) | (4,0) | (4,2) | (4,4) | (6,0) |
|---|---|---|---|---|---|---|---|
| $a_{\Delta\ell}$ | 1 | $-1$ | 4 | 1 | $\frac{-2}{3}$ | $\frac{8}{5}$ | $\frac{-1}{6}$ |
| $(\Delta,\ell)$ | (6,2) | (6,4) | (6,6) | (8,0) | (8,2) | (8,4) | (8,6) |
| $a_{\Delta\ell}$ | $\frac{2}{5}$ | $\frac{-8}{35}$ | $\frac{32}{63}$ | $\frac{1}{60}$ | $\frac{-2}{35}$ | $\frac{8}{63}$ | $\frac{-16}{231}$ |
| $(\Delta,\ell)$ | (8,8) | (10,0) | (10,2) | (10,4) | (10,6) | (10,8) | (10,10) |
| $a_{\Delta\ell}$ | $\frac{64}{429}$ | $\frac{-1}{700}$ | $\frac{1}{189}$ | $\frac{-4}{231}$ | $\frac{16}{429}$ | $\frac{-128}{6435}$ | $\frac{512}{12155}$ |

the form

$$\mathcal{T}^{ab}(x,\theta,\bar\theta) = \mathcal{T}_0^{ab}(x) + \theta\,\mathcal{T}_{\bar\theta}^{ab}(x) + \bar\theta\,\mathcal{T}_{\theta}^{ab}(x) + \theta\bar\theta\,\mathcal{T}_{\theta\bar\theta}^{ab}(x), \tag{143}$$

where $\mathcal{T}_0^{\mu\nu}$ has $\Delta = 2$ and $\ell = 2$ and dimensionally reduces to the two-dimensional stress tensor. On the other hand $\mathcal{T}_{\theta\bar\theta}^{\mu\nu}$ has $\Delta = 4$ and $\ell = 2$ and is related to the $4d$ stress tensor (to be precise this operator should be improved to be a primary, which we call $\tilde{\mathcal{T}}_{\theta\bar\theta}^{\mu\nu}$). Similarly this happens for all higher conserved currents. We have

$$\mathcal{J}^{a_1\dots a_\ell}(x,\theta,\bar\theta) = \mathcal{J}_0^{a_1\dots a_\ell}(x) + \theta\,\mathcal{J}_{\bar\theta}^{a_1\dots a_\ell}(x) + \bar\theta\,\mathcal{J}_{\theta}^{a_1\dots a_\ell}(x) + \theta\bar\theta\,\mathcal{J}_{\theta\bar\theta}^{a_1\dots a_\ell}(x), \tag{144}$$

where $\mathcal{J}_0^{\mu_1\dots\mu_\ell}$ has dimension $\Delta = \ell$ and dimensionally reduces to a $2d$ conserved current. This operator is below the unitarity bound in $4d$ and appears always accompanied with both $\mathcal{J}_{\theta\bar\theta}^{\mu_1\dots\mu_\ell}$ and $\mathcal{J}_0^{\mu_1\dots\mu_{\ell-2}\theta\bar\theta}$ which have dimension that satisfy $\Delta = \ell + 2$ and thus define $4d$ conserved currents when opportunely improved to be primary operators which we shall call $\tilde{\mathcal{J}}_{\theta\bar\theta}^{\mu_1\dots\mu_\ell}$ and $\tilde{\mathcal{J}}_0^{\mu_1\dots\mu_{\ell-2}\theta\bar\theta}$. It is interesting that these two operators correspond to $4d$ currents of spin $\ell$ and $\ell-2$ which appear inside the same spin $\ell$ supercurrent. In particular since there are infinitely many supercurrents of increasing $\ell$ it means that there are typically two $4d$ higher-spin conserved current for each spin. E.g. there is a stress-tensor-like field appearing as $\tilde{\mathcal{T}}_{\theta\bar\theta}^{\mu\nu}$, but there is also one from $\tilde{\mathcal{J}}_0^{\mu\nu\theta\bar\theta}$ which appears inside the spin-four supercurrent. Both of them have spin equal to two and dimension $\Delta = 4$. This fact also means that the respective coefficients $a_{\Delta\ell}$ in table 144 will contain an admixture of such contributions.

**Correlator $\sigma\sigma\epsilon\epsilon$**

Another $\hat{d} = 2$ correlator which only exchanges the Virasoro identity block is the one of $\sigma\sigma\epsilon\epsilon$. Let us show how this can be decomposed in $d = 4$,

$$f_{\sigma\sigma\epsilon\epsilon} = \sum_{\Delta=2\mathbb{N}_{\ge 0}} \sum_{\ell=0,2,\dots,\Delta} a_{\Delta\ell}\, g_{\Delta\ell}^{(d=4)}. \tag{145}$$

Examples of the coefficients $a_{\Delta\ell}$ for all $\Delta \le 10$ can be found in table 6. As expected the same supercurrents are exchanged and therefore we obtain two towers of operators with dimensions $\Delta = \ell$ and $\Delta = \ell + 2$.

**Correlator $\sigma\epsilon\epsilon\sigma$**

The correlation function $\sigma\epsilon\epsilon\sigma$ in $\hat{d} = 2$ is written in terms of the contribution of the single Virasoro block associated to $\sigma$. Let us see how this contribution is written in the Parisi-Sourlas minimal model in $d = 4$. We get a decomposition of this form (here the conformal blocks

Table 6: Decomposition of $\sigma\sigma\epsilon\epsilon$: All coefficients for exchanged operators with dimension $\Delta \leq 10$.

| $(\Delta,\ell)$ | $(0,0)$ | $(2,0)$ | $(2,2)$ | $(4,0)$ | $(4,2)$ | $(4,4)$ | $(6,0)$ |
|---|---|---|---|---|---|---|---|
| $a_{\Delta\ell}$ | $1$ | $-\frac{1}{8}$ | $\frac{1}{2}$ | $\frac{1}{64}$ | $-\frac{5}{96}$ | $\frac{3}{40}$ | $\frac{-5}{3072}$ |
| $(\Delta,\ell)$ | $(6,2)$ | $(6,4)$ | $(6,6)$ | $(8,0)$ | $(8,2)$ | $(8,4)$ | $(8,6)$ |
| $a_{\Delta\ell}$ | $\frac{3}{1280}$ | $\frac{-19}{2240}$ | $\frac{5}{336}$ | $\frac{1}{16384}$ | $\frac{-19}{71680}$ | $\frac{5}{10752}$ | $\frac{-205}{118272}$ |
| $(\Delta,\ell)$ | $(8,8)$ | $(10,0)$ | $(10,2)$ | $(10,4)$ | $(10,6)$ | $(10,8)$ | $(10,10)$ |
| $a_{\Delta\ell}$ | $\frac{175}{54912}$ | $\frac{-57}{22937600}$ | $\frac{25}{2064384}$ | $\frac{-205}{3784704}$ | $\frac{175}{1757184}$ | $\frac{-497}{1317888}$ | $\frac{441}{622336}$ |

Table 7: Decomposition of $\sigma\epsilon\epsilon\sigma$: All coefficients for exchanged operators with dimension $\Delta \leq 10$.

| $(\Delta,\ell)$ | $(1/8,0)$ | $(17/8,0)$ | $(25/8,1)$ | $(25/8,3)$ | $(33/8,2)$ | $(41/8,1)$ | $(41/8,3)$ | $(41/8,5)$ |
|---|---|---|---|---|---|---|---|---|
| $a_{\Delta\ell}$ | $\frac{1}{4}$ | $\frac{4}{7}$ | $\frac{1}{51}$ | $\frac{-4}{51}$ | $\frac{32}{495}$ | $\frac{16}{357}$ | $\frac{332}{38335}$ | $\frac{-16}{1045}$ |
| $(\Delta,\ell)$ | $(49/8,0)$ | $(49/8,4)$ | $(49/8,6)$ | $(57/8,3)$ | $(57/8,5)$ | $(57/8,7)$ | $(65/8,0)$ | $(65/8,2)$ |
| $a_{\Delta\ell}$ | $\frac{1}{2601}$ | $\frac{9496}{759525}$ | $\frac{32}{215865}$ | $\frac{64}{7315}$ | $\frac{1262672}{766768275}$ | $\frac{-14400}{4411463}$ | $\frac{-83}{1955085}$ | $\frac{4}{53295}$ |
| $(\Delta,\ell)$ | $(65/8,4)$ | $(65/8,6)$ | $(65/8,8)$ | $(73/8,1)$ | $(73/8,3)$ | $(73/8,5)$ | $(73/8,7)$ | $(73/8,9)$ |
| $a_{\Delta\ell}$ | $\frac{-128}{1511055}$ | $\frac{3896384}{1472798433}$ | $\frac{256}{4373439}$ | $\frac{142}{246685725}$ | $\frac{-8}{11009115}$ | $\frac{57600}{30880241}$ | $\frac{52942784}{157026025485}$ | $\frac{-265984}{366285997}$ |

have $\Delta_{12} = -\Delta_{34} = -7/8$),

$$f_{\sigma\epsilon\epsilon\sigma} = \sum_{\Delta=\frac{1}{8}+\mathbb{N}_{\geq 0}} \sum_{\ell=0}^{\Delta-\frac{1}{8}} a_{\Delta\ell}\, g_{\Delta\ell}^{(d=4)}. \tag{146}$$

In table 7 we show all the non-vanishing coefficients with $\Delta \leq 10$. We see that for each spin there exists one contribution below the unitarity bound. In particular the scalar bound $\Delta \geq 1$ is violated by an operator with dimension $\frac{1}{8}$. This in $\hat{d}=2$ corresponds to the field $\sigma$ itself. In $d=4$ we instead have a scalar superprimary

$$\mathcal{S}(x,\theta,\bar{\theta}) = \mathcal{S}_0(x) + \theta\mathcal{S}_{\bar{\theta}}(x) + \bar{\theta}\mathcal{S}_{\theta}(x) + \theta\bar{\theta}\mathcal{S}_{\theta\bar{\theta}}(x), \tag{147}$$

which has lowest component $\mathcal{S}_0$ with dimensions $\frac{1}{8}$. The bounds for operators with spin are $\Delta \geq \ell + 2$. We see that there is a tower of operators below this bound which have spin $\ell$ and dimensions $\Delta = \frac{1}{8} + \ell$ (for $\ell = 3,5,6,7,\dots$). All other operators are instead above unitarity bounds.

**Correlator $\sigma\sigma\sigma\sigma$**

In $\hat{d}=2$ the correlator of four $\sigma$ exchanges two Virasoro multiplets: the one of the identity which we already saw in previous cases, and the one of $\epsilon$, which will give rise to new features in the uplifted theory. Let us write the conformal block decomposition on the lowest component of the uplifted correlator of four $\sigma$ operators in $d=4$,

$$f_{\sigma\sigma\sigma\sigma} = \sum_{\Delta=0,2,\dots} \sum_{\ell=0,2,\dots\Delta} a_{\Delta\ell} g_{\Delta\ell}^{(d=4)} + \sum_{\Delta=5,7,\dots} \sum_{\ell=0,2,\dots\Delta-2} a_{\Delta\ell} g_{\Delta\ell}^{(d=4)} + a_{10}\tilde{g}_{\Delta=1\,\ell=0}^{(d=4)} + \sum_{\Delta=5,7,\dots} a_{\Delta\,\Delta-1}\tilde{g}_{\Delta\,\ell=\Delta-1}^{(d=4)}, \tag{148}$$

Table 8: Decomposition of $\sigma\sigma\sigma\sigma$: All coefficients for exchanged operators with even dimension $\Delta \leq 10$.

| $(\Delta,\ell)$ | $(0,0)$ | $(2,0)$ | $(2,2)$ | $(4,0)$ | $(4,2)$ | $(4,4)$ | $(6,0)$ |
|---|---|---|---|---|---|---|---|
| $a_{\Delta\ell}$ | $1$ | $\frac{-1}{64}$ | $\frac{1}{16}$ | $\frac{1}{4096}$ | $\frac{-31}{6144}$ | $\frac{9}{2560}$ | $\frac{-31}{1572864}$ |
| $(\Delta,\ell)$ | $(6,2)$ | $(6,4)$ | $(6,6)$ | $(8,0)$ | $(8,2)$ | $(8,4)$ | $(8,6)$ |
| $a_{\Delta\ell}$ | $\frac{9}{655360}$ | $\frac{-381}{1146880}$ | $\frac{25}{57344}$ | $\frac{93}{335544320}$ | $\frac{-381}{293601280}$ | $\frac{25}{14680064}$ | $\frac{-21653}{484442112}$ |
| $(\Delta,\ell)$ | $(8,8)$ | $(10,0)$ | $(10,2)$ | $(10,4)$ | $(10,6)$ | $(10,8)$ | $(10,10)$ |
| $a_{\Delta\ell}$ | $\frac{15527}{224919552}$ | $\frac{-3429}{751619276800}$ | $\frac{775}{22548578304}$ | $\frac{-21653}{124017180672}$ | $\frac{15527}{57579405312}$ | $\frac{-1600069}{215922769920}$ | $\frac{251145}{20392706048}$ |

Table 9: Decomposition of $\sigma\sigma\sigma\sigma$: All coefficients for exchanged operators with odd dimension $\Delta \leq 10$.

| $(\Delta,\ell)$ | $(1,0)$ | $(5,2)$ | $(5,4)$ | $(7,4)$ | $(7,6)$ | $(9,0)$ | $(9,6)$ | $(9,8)$ |
|---|---|---|---|---|---|---|---|---|
| $a_{\Delta\ell}$ | $\frac{1}{8}$ | $\frac{-1}{16384}$ | $\frac{1}{4096}$ | $\frac{-29}{1048576}$ | $\frac{1}{20480}$ | $\frac{1}{1073741824}$ | $\frac{-5501}{1006632960}$ | $\frac{1125}{117440512}$ |

where the tilded conformal blocks are as follows: the block $\tilde{g}^{(d=4)}_{\Delta=1\,\ell=0}$ is defined as in (102), while the blocks $\tilde{g}^{(d=4)}_{\Delta\,\ell=\Delta-1}$ are defined as in (B.5) with $[\vec{S}]$ being trivial, *i.e.* the lowest component. More about the tilded blocks below. We explicitly computed the coefficients $a_{\Delta\ell}$ of (148) for all operators with $\Delta \leq 10$. All operators have integer conformal dimensions. There is a tower of even dimensional operator whose coefficients are shown in table 8. There is also a tower of odd dimensional operator whose coefficients are presented in table 9. All other coefficient with $\Delta \leq 10$ vanish.

We notice that the unitarity bound $\Delta \geq \frac{d}{2}-1 = 1$ for scalars and $\Delta \geq \ell + d - 2 = \ell + 2$ are not always respected. Many special operators appear in this decomposition. Let us start by analyzing the spectrum of operators with even scaling dimensions. These correspond in 2$d$ to the contributions of the Virasoro multiplet associated to the identity. Like for the correlation function of $\sigma\sigma\epsilon\epsilon$ and $\epsilon\epsilon\epsilon\epsilon$ above we recognize the exchange of the identity, the super stress tensor $\mathcal{T}^{ab}$ and all possible conserved super currents $\mathcal{J}^{a_1\cdots a_\ell}$ with even spin $\ell$. In particular we consistently see the pattern that each $d = 4$ conserved tensors $\Delta = \ell + 2$ are always accompanied with a lower dimensional operators with $\Delta = \ell$ according to (144).

Let us now focus on the operators with odd scaling dimensions which in 2$d$ correspond to contributions of the Virasoro multiplet of $\epsilon$. First we find a scalar with dimension $\Delta = 1$ which has the same quantum numbers of the 2$d$ field $\epsilon$. In $d = 4$ this corresponds to a scalar superfield

$$\mathcal{E}(x,\theta,\bar{\theta}) = \mathcal{E}_0(x) + \theta\mathcal{E}_{\bar{\theta}}(x) + \bar{\theta}\mathcal{E}_\theta(x) + \theta\bar{\theta}\mathcal{E}_{\theta\bar{\theta}}(x), \tag{149}$$

where the lowest component $\mathcal{E}_0$ has dimension $\Delta = 1$. Since $\mathcal{E}_0$ has the quantum number of a free field in $d = 4$ we would expect a singularity in the respective conformal block with $\Delta = 1, \ell = 0$ (see e.g. formula (96) with $\Delta_{12} = 0 = \Delta_{34}$). However —as explained in section 4.3— supersymmetry works by exactly subtracting the singularity. Meaning that the conformal block for the exchange of $\mathcal{E}_0$ is divergent, but the singularity is exactly cancelled by the block of the exchange of the primary built out of $\mathcal{E}_{\theta\bar{\theta}}$, namely

$$\tilde{g}^{(d=4)}_{\Delta=1\,\ell=0} \equiv \lim_{\Delta\to 1}\left[ g^{(d=4)}_{\Delta\,\ell=0} - \frac{\Delta^2}{16(\Delta-1)(\Delta+1)} g^{(d=4)}_{\Delta+2\,\ell=0} \right], \tag{150}$$

which, because of formula (74), is equal to its two-dimensional counterpart

$\tilde{g}^{(d=4)}_{\Delta=1\,\ell=0} = g^{(d=2)}_{\Delta=1\,\ell=0}$. To be precise, according to (43), the actual primary is defined as $\tilde{\mathcal{E}}_{\theta\bar{\theta}} = [\partial_x^2 \mathcal{E} - (2\Delta_{\mathcal{E}} - d + 2)\partial_{\bar{\theta}}\partial_\theta \mathcal{E}]_{\theta=0} = \partial_x^2 \mathcal{E}_0$, which is just the level-two descendant of $\mathcal{E}_0$ with dimension $\Delta = 3$ and spin zero. This operator is both a primary and a descendant, and it has zero norm as it can be easily seen by considering the form of the two-point function (52) with $\Delta = 1, d = 4$. It is however crucial to notice that while being an operator with zero norm, $\tilde{\mathcal{E}}_{\theta\bar{\theta}}$ should not be modded out from the spectrum. Indeed even if the two-point function of $\tilde{\mathcal{E}}_{\theta\bar{\theta}}$ is zero, its higher point correlation functions with generic operators are typically non-vanishing. We can explicitly see this in the tables 3 and 4, by considering any $[\vec{S}]$ that contain a bold entry. E.g. the case $[\mathbf{1}]$ is just the four-point function $\langle \tilde{\mathcal{E}}_{\theta\bar{\theta}} \mathcal{E}_0 \mathcal{E}_0 \mathcal{E}_0 \rangle$, which is clearly non vanishing. This can also be seen in full generality at the level of three-point functions by looking at the explicit formulae of section 3.2, e.g. equation (55) with $\Delta_i = 1$.

In fact the presence of $\tilde{\mathcal{E}}_{\theta\bar{\theta}}$ is crucial since it plays the role of artificially shortening the multiplet of $\mathcal{E}_0$. By this we mean that the pole in the block of $\mathcal{E}_0$ is associated to the exchange of the descendant $\partial_x^2 \mathcal{E}_0$, but this contribution is cancelled by the exchange of the supersymmetric partner $\tilde{\mathcal{E}}_{\theta\bar{\theta}}$. So we can conclude that the tilded block is in practice encoding the exchange of a shortened multiplet of $\mathcal{E}_0$ where one removes the submultiplet of $\partial_x^2 \mathcal{E}_0$ because of a tuned addition of the multiplet of $\tilde{\mathcal{E}}_{\theta\bar{\theta}}$.

As an observation, we find it interesting that in two-dimensions the field $\epsilon$ can be considered as a product of chiral free fermions $\epsilon = \psi\bar{\psi}$. On the other hand in $d = 4$ the field $\mathcal{E}_0$ has the dimension of a free boson. In both cases this operator looks like a free field in a strongly coupled theory. However the reason why this happens in the two situations is somewhat different. In $\hat{d} = 2$ the reasoning is that the operator $\sigma$ does not have a simple local expression in terms of $\psi$, so correlators involving $\sigma$ cannot be obtained by Wick theorem. On the other hand in $d = 4$ the operator $\mathcal{E}_0$ only shares the quantum numbers of a free field, but it does not satisfy the same equations of motion, namely $\partial_x^2 \mathcal{E}_0 = \tilde{\mathcal{E}}_{\theta\bar{\theta}} \neq 0$, where $\tilde{\mathcal{E}}_{\theta\bar{\theta}}$ is a good primary operator in the spectrum.

There is also another type of tilded block in (148), namely $\tilde{g}^{(d=4)}_{\Delta\,\ell=\Delta-1}$ for all odd $\Delta \geq 5$. These correspond to the exchange of primaries in the supermultiplet of

$$\mathcal{Y}^{a_1 \cdots a_\ell}(x, \theta, \bar{\theta}) = \mathcal{Y}_0^{a_1 \cdots a_\ell}(x) + \theta \mathcal{Y}_{\bar{\theta}}^{a_1 \cdots a_\ell}(x) + \bar{\theta} \mathcal{Y}_{\theta}^{a_1 \cdots a_\ell}(x) + \theta\bar{\theta} \mathcal{Y}_{\theta\bar{\theta}}^{a_1 \cdots a_\ell}(x), \tag{151}$$

where the lowest component $\mathcal{Y}_0^{\mu_1 \cdots \mu_\ell}$ had dimension $\Delta = \ell + 1$ and lies below the unitarity bounds.[20]

In particular the tilded block $\tilde{g}^{(d=4)}_{\ell+1\,\ell}$ arises because of the exchange of $\mathcal{Y}_0^{\mu_1 \cdots \mu_\ell}$ combined with the primary built out of $\mathcal{Y}_{\theta\bar{\theta}}^{\mu_1 \cdots \mu_{\ell-2}\theta\bar{\theta}}$ (which we shall call $\tilde{\mathcal{Y}}_{\theta\bar{\theta}}^{\mu_1 \cdots \mu_{\ell-2}\theta\bar{\theta}}$) that has dimensions $\Delta = \ell + 3$ and spin $\ell - 2$. In this case the block of $\mathcal{Y}_0^{\mu_1 \cdots \mu_\ell}$ has a pole due to the primary-descendant $\partial_{\mu_1}\partial_{\mu_2}\mathcal{Y}_0^{\mu_1\mu_2 \cdots \mu_\ell}$, which is erased thanks to the exchange of $\tilde{\mathcal{Y}}_{\theta\bar{\theta}}^{\mu_1 \cdots \mu_{\ell-2}\theta\bar{\theta}}$. In particular,

$$\tilde{g}^{(d=4)}_{\ell+1\,\ell} \equiv \lim_{\Delta \to \ell+1} \left[ g^{(d=4)}_{\Delta\,\ell} + \frac{(\ell - \Delta)^2}{64(-\Delta + \ell - 1)(-\Delta + \ell + 1)} g^{(d=4)}_{\Delta+2\,\ell-2} \right]. \tag{152}$$

The resulting regularized block can be written in a closed form (making use of formula (74)) and gives

$$\begin{aligned}
\tilde{g}^{(d=4)}_{\ell+1\,\ell} = \frac{8\ell\,(\bar{z} - 2)\bar{z}^{\frac{1}{2}} z^{\ell - \frac{1}{2}} K(\bar{z})}{\pi(-2)^\ell (2\ell - 1)(z - \bar{z})} \bigg[ & (z - 2)\,{}_2F_1\left(\ell - \frac{1}{2}, \ell - \frac{1}{2}; 2\ell - 1; z\right) \\
& - 2(z - 1)\,{}_2F_1\left(\ell - \frac{1}{2}, \ell + \frac{1}{2}; 2\ell - 1; z\right) \bigg] + z \leftrightarrow \bar{z},
\end{aligned} \tag{153}$$

---

[20]These exchanges only happen for $\ell \geq 4$, and in particular it is missing the operator with $\Delta = 3$ and spin $\ell = 2$. This is expected since this operator is also missing in the $\hat{d} = 2$ decomposition because by BPZ equations $L_{-2}\epsilon = \frac{3}{4}L_{-1}^2\epsilon$ is a descendant [36]. Similar considerations apply to the module of $\sigma$ which satisfies $L_{-2}\sigma = \frac{4}{3}L_{-1}^2\sigma$.

where $K(x)$ is the complete elliptic integral of the first kind.

It is interesting that the operators in (151) come from the uplift of Virasoro descendants of $\epsilon$ in $\hat{d} = 2$ (e.g. $L_{-k}\epsilon$ has dimension $\Delta = 1 + k$ and spin $\ell = k$). On the other hand $\mathcal{Y}_0^{\mu_1\dots\mu_\ell}$ in $d = 4$ have the same quantum numbers of generalized currents, namely they satisfy $\Delta = \ell + d - 3$. Operators with these quantum numbers typically satisfy a conservation equation of second order in the derivative, however in this case the conservation does not hold namely $\partial_{\mu_1}\partial_{\mu_2}\mathcal{Y}_0^{\mu_1\mu_2\dots\mu_\ell} \neq 0$ (as in the case of $\mathcal{E}_0$ the shortening condition is artificial and due to the exchange of $\tilde{\mathcal{Y}}_{\theta\bar{\theta}}^{\mu_1\dots\mu_{\ell-2}\theta\bar{\theta}}$). So we see that, as $\epsilon$ is uplifted to an operator with the same quantum number of a free field, the uplift of the Virasoro descendant of $\epsilon$ also have the flavour of an infinite set of (generalized) currents which one expects to find in free theories. However in both the cases the uplifted operators do not satisfy the shortening conditions which one expects in free theory and therefore are fundamentally different.

In the conformal block decomposition (148) there are also other operator exchanges in the same supermultiplet of $\mathcal{Y}^{a_1\dots a_\ell}$, e.g. $\mathcal{Y}_0^{\mu_1\dots\mu_{\ell-2}\theta\bar{\theta}}$ which has dimensions $\Delta = \ell + 1$ and spin $\ell - 2$. These other operators lye above the unitarity bounds and are associated to non-singular conformal blocks.

## 6.3 Comments on the operator spectrum

Let us summarize some features of the spectrum of the uplifted Parisi-Sourlas minimal models. First let us review some features of the original two dimensional models. In $\hat{d} = 2$, given a Virasoro primary $O$, one can build all possible Virasoro descendants applying the Virasoro generators $L_n, \bar{L}_n$ as follows $\bar{L}_{-k}^{\bar{n}_k}\cdots\bar{L}_{-1}^{\bar{n}_1}L_{-k}^{n_k}\cdots L_{-1}^{n_1}O$. The resulting operators have quantum numbers $h = \sum_{m=1}^{k} n_m m + h_O$ and $\bar{h} = \sum_{m=1}^{\bar{k}} \bar{n}_m m + \bar{h}_O$ (in the case of the two dimensional Ising model $O = 1, \sigma, \epsilon$ and thus we get three towers of operators with respectively $(h_O, \bar{h}_O) = (0,0), (\frac{1}{16}, \frac{1}{16}), (\frac{1}{2}, \frac{1}{2})$). If the module is non-degenerate all the Virasoro descendants are independent, however in minimal models all modules are singular, namely there are relations between Virasoro descendants —the BPZ equations [36]— which should be modded out. Between the independent Virasoro states one can generate all the global primaries by requiring that they are annihilated by $L_1$ and $\bar{L}_1$. It is easy to see that the number of the global primaries exponentially grows with the Virasoro level. In appendix C we show the construction of such primaries and their counting more explicitly in the Ising case.

Every global primary in $\hat{d} = 2$ has an uplifted superprimary counterpart which itself has various components. We could see the presence of such operators directly in the conformal block decompositions above. We noticed that any holomorphic or antiholomorphic global primary in the identity module (namely any operator built out of $L_{-k}^{n_k}\cdots L_{-2}^{n_2}1$ such that $L_1$ annihilates it and similarly for the antiholomorphic case, as shown in appendix C) gives rise to a superfield $\mathcal{J}^{a_1\dots a_\ell}$ with superdimension $\Delta = \ell$. The number of such supercurrents grows exponentially in $\ell$ (see appendix C). The supermultiplet of each $\mathcal{J}^{a_1\dots a_\ell}$ in contains various interesting primaries which we explicitly observe in the conformal block decompositions.[21] The primaries $\tilde{\mathcal{J}}_{\theta\bar{\theta}}^{\mu_1\dots\mu_\ell}$ and $\tilde{\mathcal{J}}_0^{\mu_1\dots\mu_{\ell-2}\theta\bar{\theta}}$ play the role of four-dimensional conserved currents of respectively spin $\ell$ and $\ell - 2$. These appeared in the decompositions above. Moreover there are also other important components of the same supermultiplet, like the fermionic currents $\tilde{\mathcal{J}}_0^{\mu_1\dots\mu_{\ell-1}\theta}$ and $\tilde{\mathcal{J}}_\theta^{\mu_1\dots\mu_\ell}$ which appear in the decomposition in appendix D. They have dimension $\Delta = \ell + 1$ and spin respectively equal to $\ell - 1$ and $\ell$. These operators satisfy a generalized conservation equation of the type $\partial_{\mu_1}\partial_{\mu_2}\tilde{\mathcal{J}}_0^{\mu_1\dots\mu_{\ell-1}\theta} = 0$ and $\partial_{\mu_1}\partial_{\mu_2}\tilde{\mathcal{J}}_\theta^{\mu_1\dots\mu_\ell} = 0$. Finally there exists also a component $\tilde{\mathcal{J}}_\theta^{\mu_1\dots\mu_{\ell-1}\theta}$ which has charge two under $Sp(2)$, and satisfies $\Delta = \ell + 2$

---

[21]It is worth noticing that the Virasoro identity multiplet can be exchanged only in the OPE of equal operators, which is consistent with the fact that the higher spin currents can also only appear in the OPE of equal operators.

with spin $\ell - 1$. This also satisfies a conservation equation $\partial_{\mu_1} \tilde{\mathcal{J}}_\theta^{\mu_1 \dots \mu_{\ell-1} \theta} = 0$. With each such current one can in principle construct topological operators that in turn define conserved charges (whose number will also grow exponentially in $\ell$). This property is shared by all minimal models since it only relies on the existence of the Virasoro multiplet of the identity.

Let us also stress that in $\hat{d} = 2$ all the different global currents discussed above can be packaged in a unique Virasoro multiplet. This structure should play a role also in the uplifted theory. However in the uplifted theory for every current one can build more charges than in $\hat{d} = 2$. The simplest example is that of (super)translations. Indeed in $\hat{d} = 2$ we can build a topological operator by integrating the stress tensor to define the translations $L_{-1}$ and $\bar{L}_{-1}$. When we do the same using the super stress tensor in $d = 4$ we can build six different types of supertranslations: four bosonic and two fermionic. Only two of the bosonic ones are related to $L_{-1}$ and $\bar{L}_{-1}$, while the other four are new charges that cannot be defined in $\hat{d} = 2$. If we believe that these charges will be packaged with all the ones of the higher spin tensors inside a single algebra, we should imagine that the resulting algebra must contain infinitely many more charges than Virasoro. It would be very interesting to better understand this infinite-dimensional algebra in the uplifted theory.

It is also important to mention that the existence of higher spin conserved currents in $d = 4$ is in contrast with the no-go theorem of [39], which showed that under some assumptions the only theory with higher spin symmetry in $d \geq 3$ is free theory. However this is not a paradox since one of the assumptions of [39] is unitarity. Therefore [39] does not apply to the PS uplift of the minimal models.

The rest of the operator content depends on the chosen minimal model. It happens that the case of the Ising model is somewhat special since it also contains the operator $\epsilon$ which has dimension one. Because of this fact we find that in the supermultiplet of its uplifted counterpart $\mathcal{E}$ there exists the operator $\mathcal{E}_0$ which has the same quantum numbers of a free scalar but does not satisfy equation of motions. Similarly in the same Virasoro multiplet of $\epsilon$ there are all the (anti) holomorphic global primaries which again they are defined by $L_{-k}^{n_k} \cdots L_{-1}^{n_1} \epsilon$ (and barred version of it) which are annihilated by $L_1$ ($\bar{L}_1$) as in appendix C. The uplift of the latter give rise to the superfields $\mathcal{Y}^{a_1 \dots a_\ell}$ with superdimension $\Delta = \ell + 1$ (again the number of such superfields grows exponentially with $\ell$). In their supermultiplet there exist the primaries $\mathcal{Y}_0^{\mu_1 \dots \mu_\ell}$ which have $\Delta = \ell + d - 3$. Operators with these quantum numbers typically satisfy the generalized conservation equations $\partial_{\mu_1} \partial_{\mu_2} \mathcal{Y}_0^{\mu_1 \mu_2 \dots \mu_\ell} = 0$, but in this case this equation does not hold so these currents are not conserved. A similar story works for other components of $\mathcal{Y}$, like $\mathcal{Y}_\theta^{\mu_1 \mu_2 \dots \mu_\ell}$ and $\mathcal{Y}_0^{\mu_1 \mu_2 \dots \mu_{\ell-1} \theta}$ which have the quantum numbers of higher spin conserved currents but they do not satisfy shortening conditions as one can infer from (B.7) and (B.8). The fact that all these operators do not satisfy shortening conditions implies that the theory contains zero norm states which are not modded out. This would in principle be a problem in the OPE, since the exchange of null states would give rise to poles. However because of supersymmetry other operators in the same supermultiplet are perfectly tuned to cancel the poles. This is an instance of the general discussion of section 4.3.

Finally for the case of the Ising model one can study the uplift of the Virasoro multiplet of the spin operator $\sigma$. Since the latter has rational dimensions it does not give rise to shortened global representations. However we can point out that the uplifted field $\mathcal{S}$ has lowest component $\mathcal{S}_0$ with dimension $\frac{1}{8}$ which lyes below the unitarity bound and similarly that the uplift of all the holomorphic global primaries in the multiplet of $\sigma$ have lowest component with dimension $\Delta = \frac{1}{8} + \ell$ which lye below the unitarity bound.

To complete the discussion on the spectrum we should mention that in the uplifted theory there should exist new operators which do not exist in the dimensionally reduced theories. It would be interesting to see if it is possible to reconstruct them by performing a bootstrap analysis of correlators that cannot be trivially reconstructed from their lower dimensional counter-

parts, e.g. by considering the four-point functions of spinning operators (e.g. stress tensors) or by increasing the number of insertions (e.g. the six-point function of $\sigma$ or $\epsilon$ in the Ising model). This might be possible since we know a huge amount of exact information for the model. Another possible strategy to fill out this missing information is to better understand the underlying infinite dimensional symmetry of the uplifted theory and see if this can be used to completely solve the model, in the same spirit as it was done for its two-dimensional counterparts.

# 7 Comments on the loss of information in the reduction

In this paper we always considered cases when the uplift works trivially. In particular we could simply use the prescription (39), which replace the insertions in $\mathbb{R}^{d-2}$ with insertions in $\mathbb{R}^{d|2}$. This is possible when the symmetry of the observable is sufficient to reconstruct the full kinematic dependence of the 2|2 extra dimensions. However in some cases this complete reconstruction cannot be done. Here we want to mention a couple of explicit examples.

## 7.1 On the operators that dimensionally reduce to zero

In Parisi-Sourlas theories one is allowed to define some operators in $OSp(d|2)$ representations which cannot possibly be reduced to non vanishing operators in $\hat{d} = d - 2$ dimensions. The representations of $OSp(d|2)$ allow for superprimaries labelled by a Young tableau of $[d/2]$ rows and one arbitrarily long column as in (19) (where rows are graded symmetrized and columns graded antisymmetrized). In particular one can consider a superprimary with an arbitrary number of graded antisymmetric indices. This for sure cannot have any dimensionally reduced counterpart since there are no available representations of $SO(d-2)$ labelled by these Young tableau. Namely it is not possible to antisymmetrize an arbitrary number of indices of a $SO(d-2)$-tensor and therefore such operators are projected to zero in the dimensionally reduced theory. In order to make this less abstract we want to give a very explicit construction of such class of PS CFT operators which projects to zero.

A simple way to study this problem is to consider a Parisi-Sourlas $O(n)$ vector model defined by the action,

$$\int d^d x \, d\theta \, d\bar{\theta} \, \frac{1}{2} (\partial^a \Phi_i)^2 + V(\Phi_i^2), \tag{154}$$

where the summation of the vector index $i = 1, \dots, n$ is understood and $\Phi_i = \varphi_i + \bar{\theta}\psi_i + \theta\bar{\psi}_i + \theta\bar{\theta}\omega_i$. This model can be reduced to

$$\int d^{\hat{d}} x \, \frac{1}{2} (\partial^\mu \hat{\phi}_i)^2 + V(\hat{\phi}_i^2). \tag{155}$$

For simplicity we shall work in the free case $V = 0$, but similar considerations would also apply in perturbation theory.

As we explained in section 2.5, the map between the superprimaries of the PS CFT and the primaries of the reduced theory is given by replacing $\Phi_i \to \hat{\phi}_i$ and $\partial_a \to \partial_\alpha$. For example this maps works perfectly when considering

$$\Phi_i \to \hat{\phi}_i, \qquad \Phi_i^2 \to \hat{\phi}_i^2, \qquad \Phi_{[i}\partial^a \Phi_{j]} \to \phi_{[i}\partial^\alpha \hat{\phi}_{j]}. \tag{156}$$

These are different operators, the first is a scalar $O(n)$-vector, the second is a scalar $O(n)$-singlet, while the third is the current, which is a vector operator in the rank-two antisymmetric representation of $O(n)$. We could continue writing infinitely many of such examples which

have a $1 \leftrightarrow 1$ map between the reduced theory and the uplift. Let us however consider the superprimary[22]

$$\mathcal{O}^{a_1 \dots a_{n-1}} = \epsilon^{i_1 \dots i_n} \Phi_{i_1} \partial^{[a_1} \Phi_{i_2} \cdots \partial^{a_{n-1}]} \Phi_{i_n}, \tag{157}$$

which is a $O(n)$-(pseudo)-singlet which transforms in the rank-$(n-1)$ graded antisymmetric representation of $OSp(d|2)$ (namely it is defined by a Young tableau (19) with a single column with $n-1$ boxes). If we set all indices $a_i$ of (157) to $\theta$ and take its lowest component, we see that this operator just becomes

$$\mathcal{O}_0^{\theta \dots \theta} \propto \sum_{i=1}^{n} (-1)^i \varphi_i \psi_1 \cdots \psi_{i-1} \psi_{i+1} \cdots \psi_n. \tag{158}$$

This component can be written down in any dimension $d$ and it is clearly non vanishing, so the superprimary $\mathcal{O}^{a_1 \dots a_{n-1}}$ is always non-trivial. On the other hand the prescription to dimensionally reduce this operator to $\hat{d} = d - 2$ dimensions does not always give a non-zero result. Indeed it gives

$$\mathcal{O}^{a_1 \dots a_{n-1}} \rightarrow \hat{O}^{\alpha_1 \dots \alpha_{n-1}} = \epsilon^{i_1 \dots i_n} \hat{\phi}_{i_1} \partial^{[\alpha_1} \phi_{i_2} \cdots \partial^{\alpha_{n-1}]} \hat{\phi}_{i_n}, \tag{159}$$

which again is a $O(n)$-(pseudo)-singlet in the rank-$(n-1)$ antisymmetric representation of $SO(\hat{d})$. When $n - 1 \leq \hat{d}$ the operator $\hat{O}^{\alpha_1 \dots \alpha_{n-1}}$ is non-vanishing.[23] Conversely for every $n - 1 > \hat{d}$, the operator $\hat{O}^{\alpha_1 \dots \alpha_{n-1}}$ vanishes because it is not possible to antisymmetrize $n-1$ indices that take less than $n-1$ possible values. In the latter case we thus conclude that the superprimary $\mathcal{O}^{a_1 \dots a_{n-1}}$ is projected to zero in the dimensionally reduced theory. This means that it will not be part of the spectrum of the reduced theory and thus that when we uplift the reduced theory we will not be able to get any information about such operator.

To conclude, (157) gives a very explicit example of a non-trivial operator that always dimensionally reduces to zero independently on the chosen dimension $d$ (when $n$ is large enough). While this was a fairly specific construction, we expect that the uplifted theory will generically contain a larger spectrum with respect to the reduced one, where the extra operators project to zero when dimensionally reduced. Of course any correlation function which contains at least one of these operators will dimensionally reduce to zero. A less trivial statement is that these operators strongly affect the dimensional reduction also when they can be exchange in the OPE. In the following we give a precise example of this phenomenon in $d = 3$.

## 7.2 On the uplift of one-dimensional GFF

In this paper we studied the cases when the prescription 5.1 fully reconstructs the uplifted observables, however sometimes this is insufficient. The simplest example is when we uplift a $\hat{d} = 1$ four-point function to $d = 3$. This procedure cannot reconstruct the full $3d$ correlator because the latter is a functions of two cross ratios while in $1\hat{d}$ only a single independent cross ratio is available. Let us show in more details what happens in this case. In particular we focus on the case of GFF, where we know both the $1\hat{d}$ GFF and the $3d$ PS GFF and we can explicitly see which part of the uplifted theory is reconstructed by the prescription of section 5.1.

It is easy to define the PS uplift of $1\hat{d}$ GFF. Indeed following the definition of section 5.1 there is a natural way to uplift the $1\hat{d}$ action $\int d^x \hat{\phi} (\partial^2)^\xi \hat{\phi}$ to $3d$ by considering $\int d^3x \, d\theta \, d\bar{\theta} \, \Phi (\partial^a \partial_a)^\xi \Phi$. Using this we can compute the four-point function of $\Phi$ which is equal

---

[22]The operator $\epsilon^{i_1 \dots i_n} \partial^{[a_1} \Phi_{i_1} \partial^{a_2} \Phi_{i_2} \cdots \partial^{a_n]} \Phi_{i_n}$ would provide a simpler example, but it is a superdescendant of $\mathcal{O}^{a_1 \dots a_{n-1}}$ and for this reason we preferred the latter.

[23]For $n - 1 \leq [\hat{d}/2]$ the operator $\hat{O}^{\alpha_1 \dots \alpha_{n-1}}$ automatically transforms in a standard $SO(\hat{d})$ representation, while for $[\hat{d}/2] < n - 1 \leq \hat{d}$ one should dualize the operator using the $SO(\hat{d})$ epsilon tensor. E.g. in the limiting case of $n - 1 = \hat{d}$ we can dualized it to a (pseudo) scalar $\epsilon_{\alpha_1 \dots \alpha_{\hat{d}}} \hat{O}^{\alpha_1 \dots \alpha_{\hat{d}}}$.

to $1 + U^{\Delta_\phi} + U^{\Delta_\phi} V^{-\Delta_\phi}$ (where $\Delta_\phi = 3/2 - \xi$), which is indeed the answer of the Parisi-Sourlas uplift of GFF in any dimension $d > 1$. Let us consider the lowest component of this four-point function, namely of $\langle \Phi_0 \Phi_0 \Phi_0 \Phi_0 \rangle$, which is just $A^{\text{uplift}} = 1 + u^{\Delta_\phi} + u^{\Delta_\phi} v^{-\Delta_\phi}$. Its superblock decomposition reads

$$A^{\text{uplift}} = 1 + \sum_{\ell = 0,2\dots} \sum_{\Delta = 2\Delta_\phi + \ell + 2\mathbb{N}_{\geq 0}} a_{\Delta\ell} \left[ g^{(3)}_{\Delta\ell} + c_{2,0}\, g^{(3)}_{\Delta+2\,\ell} + c_{0,-2}\, g^{(3)}_{\Delta\,\ell-2} + c_{2,-2}\, g^{(3)}_{\Delta+2\,\ell-2} \right], \quad (160)$$

where $c_{i,j}$ are the coefficients (75) of the superblocks computed for $d = 3$ and $\Delta_{12} = \Delta_{34} = 0$. We dropped the term proportional to $c_{1,-1}$ which is zero for equal external operators. The coefficients in the decomposition read

$$a_{2\Delta_\phi + 2n + \ell, \ell} = \frac{2^{\ell+1} \left(\Delta_\phi + \frac{1}{2}\right)_n^2 \left(\Delta_\phi\right)_{n+\ell}^2}{n!\,\ell!\left(\ell + \frac{1}{2}\right)_n \left(n + 2\Delta_\phi\right)_n \left(n + \ell + 2\Delta_\phi - \frac{1}{2}\right)_n \left(2n + \ell + 2\Delta_\phi - 1\right)_\ell}. \quad (161)$$

These coefficients are actually equal to the ones computed for GFF in generic dimensions $\hat{d}$ when restricted to $\hat{d} = 1$. For example (161) can be obtained from (123) setting $n_2 = n_3 = 1$ and $d = 3$.

Let us now consider the setup for CFTs in $1\hat{d}$. First let us mention that in $1\hat{d}$ a correlation function of equal operators takes the simpler form

$$\langle \hat{\phi}(x_1) \hat{\phi}(x_2) \hat{\phi}(x_3) \hat{\phi}(x_4) \rangle = \frac{A(z)}{|x_{12}|^{2\Delta_\phi} |x_{34}|^{2\Delta_\phi}}, \quad (162)$$

where $z = \frac{x_{12} x_{34}}{x_{13} x_{24}}$. In principle one could also define a second cross ratio $\zeta = \frac{x_{14} x_{23}}{x_{13} x_{24}}$, but it is easy to see that in one dimension $z + \zeta = 1$, so the two cross ratios are not independent. Notice that the $\hat{d} = 1$ setup can be recovered from higher dimensions where $\frac{x_{12}^2 x_{34}^2}{x_{13}^2 x_{24}^2} = u = z\bar{z}$ by considering the diagonal limit $z = \bar{z}$ of the cross ratios.

In the next we shall focus on $1\hat{d}$ GFF, for which

$$A(z) = 1 + \left(\frac{z}{1-z}\right)^{2\Delta_\phi} + z^{2\Delta_\phi}. \quad (163)$$

Let us now discuss how to possibly uplift this result to $d = 3$. If we are only given the final expression for the correlator (without knowing to which uplifted theory it corresponds), our only way to define the uplifted correlator becomes the replacement (39). This however is ambiguous. Indeed $z^2$ uplifts to the cross ration $U$ and $\zeta^2$ to $V$, but since in $1\hat{d}$ it is always possible to replace $z \leftrightarrow 1-\zeta$ we conclude that the uplifted correlation function is known only on the slice $\sqrt{U} = 1 - \sqrt{V}$ (which corresponds to $Z = \bar{Z}$). Therefore the reconstruction of the uplift provided by the prescription (39) is incomplete. But what does (39) reconstruct exactly? To answer this question let us perform the conformal block decomposition in $\hat{d} = 1$,

$$A(z) = g^{(1)}_0(z) + \sum_{\Delta = 2\Delta_\phi + 2\mathbb{N}_{\geq 0}} a_\Delta g^{(1)}_\Delta(z), \quad (164)$$

where $g^{(1)}_\Delta = z^\Delta\, {}_2F_1(\Delta, \Delta; 2\Delta; z)$ are the one-dimensional conformal blocks and $a_\Delta$ are the (squared) OPE coefficients for the exchange of the double twist operators $\phi \partial^{2n} \phi$. These take the following form

$$a_{2\Delta_\phi + 2n} = \frac{2 \left(\Delta_\phi\right)_n \left(2\Delta_\phi\right)_n \left(\Delta_\phi + \frac{1}{2}\right)_n}{(2n)! \left(n + 2\Delta_\phi - \frac{1}{2}\right)_n}. \quad (165)$$

Now it is possible to perform the uplift of the $1\hat{d}$ blocks to $3d$ blocks by applying equation (74). In particular a $\hat{d} = 1$ conformal block is uplifted to a sum of diagonal (i.e. with $z = \bar{z}$) $3d$ blocks as follows,

$$g^{(1)}_{\Delta}(z) = g^{(3)}_{\Delta\,\ell=0}(z,z) + c_{2,0}g^{(3)}_{\Delta+2\,\ell=0}(z,z),\qquad(166)$$

where $c_{2,0} = -\Delta^3/[4(\Delta+1)(2\Delta-1)(2\Delta+1)]$ is the one of (75) restricted to $\ell, \Delta_{12}, \Delta_{34} = 0$ and $\hat{d} = 1$. Since there is no spin in $\hat{d} = 1$, equation (74) defines $d = 3$ blocks which are also scalar. Equation (166) can be checked using the closed form of the diagonal blocks [40]

$$g^{(3)}_{\Delta\,\ell=0}(z,z) = \left(\tfrac{z^2}{1-z}\right)^{\Delta/2} {}_3F_2\left(\tfrac{\Delta-1}{2},\tfrac{\Delta}{2},\tfrac{\Delta}{2};\tfrac{\Delta+1}{2},\tfrac{2\Delta-1}{2};\tfrac{z^2}{4(z-1)}\right).\qquad(167)$$

One possible idea to get the uplifted correlator in $3d$ is to take the decomposition (164), uplift each block according to (166), and finally make the $3d$ blocks depend on both $z$ and $\bar{z}$ as follows

$$A^{\text{uplift?}}(z,\bar{z}) = 1 + \sum_{\Delta=2\Delta_\phi+2\mathbb{N}_{\geq0}} a_\Delta[g^{(d=3)}_{\Delta\,\ell=0}(z,\bar{z}) + c_{2,0}g^{(d=3)}_{\Delta+2\,\ell=0}(z,\bar{z})].\qquad(168)$$

This prescription however clearly misses information. Indeed the conformal block decomposition of the $3d$ theory contains infinitely many spinning operators according to (160), however we only see scalar exchanges, which is a trivial consequence of the fact that we only uplifted scalar blocks. Of course (168) is not the full uplift, but now at least we know exactly which information it reconstructs by comparing (168) with the true uplifted correlator (160). In particular we can see that the coefficients (165) in (168) exactly match the ones of (161) in (160) when restricted to $\ell = 0$ (and so do the superblocks). Let us explain why this is the case.

Mathematically this happens because of an interesting property of the superblocks in the square brackets of (160): one can check that they all vanish at $z = \bar{z}$ for $\ell \geq 2$. A more physical point of view is related to the fact that in the $3d$ PS GFF there are more operators than in $1\hat{d}$ GFF. In particular we can define superprimaries in the graded symmetric and traceless representation of spin $\ell$, which do not have any representative in one dimensions. The prescription for the dimensional reduction indeed would be that a graded traceless and symmetric superprimary $\mathcal{O}^{a_1\dots a_\ell}$ is mapped to a traceless and symmetric primary operator $\hat{O}^{a_1\dots a_\ell}$ that lives in two less dimensions. However the dimensions of spin $\ell$ traceless and symmetric representations of $SO(d)$ is $\frac{(d+2\ell-2)\Gamma(d+\ell-2)}{\Gamma(d-1)\Gamma(\ell+1)}$, therefore in one dimension the only representations that survive are the ones with $\ell = 0, 1$. The spin one representation is better understood as a parity-odd scalar by dualizing it with the epsilon tensor, e.g. $O^\alpha\epsilon_\alpha$, but this will not play a role in our discussion since we only probe even spin and parity-even operators in the decompositions above.[24] Therefore the dimensional reduction of graded traceless and symmetric spin $\ell \geq 2$ superprimaries from a $3d$ PS CFT to $1\hat{d}$ reads

$$\mathcal{O}^{a_1\dots a_\ell} \to 0.\qquad(171)$$

We thus conclude that the spin $\ell \geq 2$ operators in the OPE define the missing information of the $1\hat{d}$ correlator (164) with respect to its uplift (160). Their presence in the uplift is crucial

---

[24]It is instructive to show what happens when spin $\ell = 1$ superblocks are exchanged in $d = 3$. As we explained, these correspond to parity odd scalar exchanges in $\hat{d} = 1$. Indeed we find

$$g^{(1)}_{\Delta}(z) = g^{(3)}_{\Delta\,\ell=1}(z,z) - \frac{(\Delta-1)(\Delta+1)^2}{4(\Delta+2)(2\Delta-1)(2\Delta+1)}g^{(3)}_{\Delta+2\,\ell=1}(z,z),\qquad(169)$$

where the coefficient above is obtained by evaluating $c_{2,0}$ at $\ell = 1 = \hat{d}$, $\Delta_{12}, \Delta_{34} = 0$ and the spin-one diagonal blocks are defined as [40]

$$g^{(3)}_{\Delta\,\ell=1}(z,z) = \frac{(2-z)}{2z}\left(\frac{z^2}{1-z}\right)^{\frac{\Delta+1}{2}} {}_3F_2\left(\frac{\Delta}{2}+\frac{1}{2},\frac{\Delta}{2}+\frac{1}{2},\frac{\Delta}{2};\frac{\Delta}{2}+1,\Delta-\frac{1}{2};\frac{z^2}{4(z-1)}\right).\qquad(170)$$

to get a four-point that depends on two cross ratios. These operators are projected to zero by dimensional reduction which is also related to the fact that the $1\hat{d}$ four-point function only depends on one cross ratio.

Let us be very explicit on the missing operators in (168) with respect to (160). In the uplifted theory we can write double twist operators labelled by two numbers $n$ and $\ell$ as in (105), and their reduction from $3d$ to $1\hat{d}$ is as follows

$$\Phi(\partial^a\partial_a)^n\partial^{a_1}\cdots\partial^{a_\ell}\Phi \to \begin{cases} \hat{\phi}\partial^{2n}\hat{\phi} & (\ell=0), \\ \hat{\phi}\partial^{2n+1}\hat{\phi} & (\ell=1), \\ 0 & (\ell\geq 2), \end{cases} \tag{172}$$

where here we suppressed the normal ordering and the dots of (105) to shorten the notation. So in practice (168) only captures the spin zero double twist operators $\Phi(\partial^a\partial_a)^n\Phi$.

Now that we explained that the complication in the uplift is due to the different operator content of the two theories, we may further ask whether starting from (168) one could reconstruct (160). We did not attempt it here but we think that this might be possible. In particular we notice that (168) contains the correct spectrum and OPE coefficients of the part of the spectrum which is not projected to zero, namely all the scalar operators. Moreover, if on one hand (168) misses completely the information of all spinning superprimaries, we never imposed the constraint of crossing symmetry and we expect that the expression (168) is not crossing invariant (one may try to prove this since the blocks at $\ell=0$ are known exactly in every dimension, but we did not attempt it). It would be interesting to see whether by requiring crossing symmetry and possibly inputting extra information about the uplifted theory one could actually reconstruct the uplifted correlator from the one on the line. We leave this problem for the future.

As a final comment it is interesting to notice that the uplift (160) can be understood as a better analytic continuation of GFF to one dimension. In the sense that by continuing the OPE and spectrum as function of the spacetime dimension we would find (160) and not (164). Also the uplift (160) is expected to have all properties of a $\text{CFT}_3$, which has a richer structure (e.g. the operators should lie in Regge trajectories which are analytic in spin) and can possess a conserved stress tensor (when $\Delta_\phi=1/2$). It may be interesting to investigate better the uplift of one dimensional theories to give them a higher dimensional interpretation. This study could be also fruitful in order to find new ways to export tools which are better understood in one dimension (e.g. the analytic functionals of [41–43]) to higher dimensions.

## 8 Conclusion and outlook

In this paper we started the study of the Parisi-Sourlas uplift of CFTs. We explained that it is possible to uplift correlators of a generic $\text{CFT}_{d-2}$ to define correlators of a PS $\text{CFT}_d$.[25] In particular any given correlation function in $d-2$ dimensions gives rise to a superspace correlator in $d$ dimensions, which can then be expanded in components generating a set of correlators of primary operators in the uplifted theory. We explicitly checked that for two, three and four insertions of scalar operators the kinematic structure of all components is (as expected) compatible with conformal symmetry. All components can be extracted by the action of known differential operators in superspace. In section 3.3 we further show that four-point functions have 43 independent components which can be obtained by applying 43 differential operators $D_{[\vec{S}]}$ in $u$ and $v$ on the lowest component of the four-point function.

---

[25]As reviewed in the introduction, PS CFTs are known to have physical applications for models with random fields, e.g. they describe the RF Ising model in $d=5$ and the RF $\phi^3$ in $2\leq d<8$.

In section 4 we provide a very robust check that for any $\text{CFT}_{d-2}$ its uplifted PS $\text{CFT}_d$ makes sense. Indeed we check that for any scalar four-point function in the $\text{CFT}_{d-2}$, the 43 components of the uplifted correlator have a good $d$-dimensional conformal block decomposition. This is ensured by formula (76), which rewrites the action of $D_{[\vec{s}]}$ on block in $d-2$ dimension as a linear combination of at most five conformal blocks in $d$ dimensions. Formula (76) is also a beautiful result for the theory of conformal blocks, indeed it provides very interesting relations between blocks shifted by two dimensions.[26] Between such relations we find (94) which was already obtained by Dolan and Osborn in [30] to map lower dimensional blocks to higher dimensional ones. Our work gives a clear physical interpretation of such equation.

PS CFTs are non unitary CFTs and by dimensional uplift we expect that their conformal block decomposition often contains exchanges of subtle conformal block which naively seem to diverge. However in section 4.3, by analyzing formula (76), we showed that all these singularities are cancelled. Physically the singularities of the blocks signal the presence of null states in the spectrum of the theory. In unitary theories these are cancelled by modding out the null states. In PS CFT we show that null states can also be part of the spectrum without giving rise to singularities in correlators, because these are subtracted by other conformal blocks contributions with singular OPE coefficients. This structure is ensured kinematically at the conformal block level thanks to PS SUSY.

These results can also be understood as new checks that both PS CFT make sense and that they can arise from the uplift of any $\text{CFT}_{d-2}$. While some checks were also performed in [7], in that paper it was given most attention to the lowest component of the superspace correlation functions, which is the one entering the dimensional reduction. In this paper we wanted to give extra motivation for the uplift, and therefore also the other components were carefully studied.

We then give specific applications of the uplift. In particular we focused on models that must possess a well defined uplift (because of Lagrangian arguments). The first case is GFF, while the second one consists in all the diagonal minimal models.

In section 5 we defined the uplifted GFF explaining how to compute any observable by using Wick contractions in superspace and we show that this description is sometimes very useful as a tool. In particular some observables of the original GFF are constrained by the supersymmetry of the uplift. The constraints arise in GFF because of factorization (therefore our computations can be extended to any theory with this property), namely the fact that any GFF correlator can be written in terms of product of two-point functions. In the uplift we find that some components of some infinite families of correlation functions vanish. This happens because their factorized form always contains a two-point function of different primaries (belonging to the same supermultiplet), which therefore is zero. In practice this reasoning tells us that some classes of GFF correlators are annihilated by some differential operators. In the case of four-point functions these differential operators are a subset of the 43 possible $D_{[\vec{s}]}$ defined in section 3.3. Combining the fact that $D_{[\vec{s}]}$ annihilates the correlator and that it maps a block in lower dimensions to a sum of blocks in higher dimensions according to (76), we find that PS SUSY gives rise to recursion relations on the OPE coefficients of the double twist exchanges. We showed this for any four-point function of the form $\langle \phi \phi^{n_1} \phi^{n_2} \phi^{n_3} \rangle$ for any power $n_i$. For all such four-point functions we find seven recurrence relations, which can be compactly written as in (118). In the case of $n_2 = n_3$ we show that these relations can be easily solved by (123). This is a new result on GFF which was obtained by only assuming the existence of double twist operators and imposing the supersymmetry of the uplift. It is quite remarkable that such minimal assumptions fix the form of infinitely many OPE coefficients of infinitely many correlators.

---

[26]Notice also that relations between conformal blocks can be used to find relations for exchange Witten diagrams [44].

On the philosophical side, the PS SUSY of the uplift defines a new mechanism (besides the usual one of analyticity in spin) which links together different operators of the same double twist family. Indeed the recurrence relations (118) just descend from the fact that a given spin $\ell$ double twist superprimary contains in its supermultiplet a number of primaries of different spin and dimensions. So around GFF, it happens that primaries belonging to different double twist supermultiplets have the same quantum numbers and recombine in such a way to give zero OPE coefficients in the conformal block decomposition, which in turns give rise to vanishing correlation functions. In practice the fact that some components have correlation functions requires a miraculous tuning of all the infinitely many OPE coefficients in the double twist family.

Besides solving for the OPE coefficients, we can also use the fact that $D_{[\vec{s}]}$ annihilates some correlators to solve for the correlators themselves. We showed this in some simple cases. In particular we could fix the correlator of $\langle \phi \phi^{n_1} \phi^{n_2} \phi^{n_3} \rangle$ up to three constants without doing any Wick contraction. Moreover we show that this same logic can have application at perturbative level (around GFF). For example we explain that the D-functions (which also describes contact Witten diagrams in AdS) are also annihilated by some of the operators $D_{[\vec{s}]}$ and that this can be helpful to bootstrap their form. This idea can be surely extended to a large set of diagrammatic computations giving rise to a new tool to bootstrap correlators in perturbation theory. This would be very interesting to explore. On a similar note it would be interesting to check whether the uplifted formulation has benefits for the analytic conformal bootstrap, which is also perturbative in nature.

In section 6 we turned our attention to the uplift of diagonal minimal models. We gave a simple RG argument which explains why the uplift of such models should exist. The idea is that diagonal minimal models can be obtained as the fixed point of the (multicritical) RG flow of a scalar Lagrangian. Since scalar Lagrangians can be uplifted, then these RG flows can be uplifted and so it must be the case for their IR fixed points. We focused on the Ising model and considered the four-point functions of Virasoro primary operators (namely combinations of $\epsilon$ and $\sigma$). We explained that they can be trivially uplifted to four dimensions (as it is the case for all scalar four-point functions) and showed as an example the 43 components of the uplift of $\langle \epsilon \epsilon \epsilon \epsilon \rangle$. We then proceeded to the decomposition of the uplifted four-point functions in $4d$ conformal blocks. This analysis shows explicitly that in the spectrum of the uplifted minimal models there exist infinitely many conserved higher spin currents. These were expected because they also exist in the minimal models themselves. Such operators can be in principle used to define an infinite set of conserved charges which extends the one of the minimal models. It would be very exciting to find the explicit expression for the charge algebra, which certainly deserves further investigations.

Interestingly the integrability properties of minimal models (meaning the existence of infinitely many commuting charges) should uplift to higher dimensions, giving rise to examples of non-trivial higher dimensional integrable models (these would be truly integrable in $d > 2$ without resorting to an auxiliary integrable $2d$ description). Notice that the existence of infinitely many conserved currents in a strongly coupled theory is in contradiction with the no-go theorem of [39], however for PS CFTs this no-go theorem does not apply because it relies on unitarity.

As a curiosity, the uplifted Ising model has special features with respect to other uplifted minimal models. This is due to the Virasoro multiplet of $\epsilon$. By uplifting the operators in this multiplet we find that they map to $4d$ operator with the same quantum numbers of free fields and conserved currents. However we show that, despite the resemblance, the uplifted fields do not satisfy shortening conditions. Namely the $4d$ multiplets contain null states which are not modded out from the spectrum of the theory. In order to prove that the exchange such multiplets is well defined we make use of the observations of section 4.3 which allows us to

conclude that the conformal block decomposition of the uplift of $\langle\sigma\sigma\sigma\sigma\rangle$ does not give rise to singularities even if zero norm states are exchanged.

As we mentioned in the introduction, some of the uplifted minimal models have a microscopic definition in terms of a supersymmetric statistical physics model [17–19]. It would be interesting to see whether this type of construction can be extended to other uplifted minimal models.

All along the paper we focused on observables that are "equivalent" in the $\mathrm{CFT}_{d-2}$ and the PS $\mathrm{CFT}_d$, meaning that they can be dimensionally reduced or uplifted without ever loosing information (or in other words the missing information of the uplift can be fully recovered using the symmetry of the model). From this, it might seem that the uplifted PS $\mathrm{CFT}_d$ is just equivalent to a $\mathrm{CFT}_{d-2}$, but this is actually incorrect. The PS $\mathrm{CFT}_d$ contains the whole information of the $\mathrm{CFT}_{d-2}$ but it also contains more operators and OPE coefficients, so it is a bigger theory. Indeed it is easy to see that some operators of the PS $\mathrm{CFT}_d$ are dimensionally reduced to zero. We exemplify a class of such operators in section 7. Moreover in the same section we also explain that the projection to zero of these operators is responsible for the smaller kinematic space (e.g. the number of cross ratios and tensor structures) in some dimensionally reduced correlators. Indeed it is well known that the kinematic space of correlators depends on the spacetime dimensions and it might seem surprising that the uplift should make sense also when the two kinematic spaces are different. The easiest example is that scalar four-point functions in a $\mathrm{CFT}_1$ depend on a single cross ratio, while their uplifted PS $\mathrm{CFT}_3$ counterparts depend on two. In section 7 we give the example of $1d$ GFF and its $3d$ PS uplift. The PS GFF contains spinning operators which are absent in $1d$ and we show that these are responsible for creating the dependence on the extra cross ratio in four-point functions. We believe that this picture should hold in general, e.g. also in relation to the number of tensor structures in correlators of spinning operators. It would be very interesting to check if this is indeed correct.

The existence of operators that dimensionally reduce to zero gives rise to a technical complication in the uplift of theories, like the minimal models, that are defined only in terms of a CFT (and not with a Lagrangian like GFF). Indeed for such theories we do not have access to a part of the spectrum of the uplifted models and one may ask whether it is possible to reconstruct it only by CFT considerations. While we do not have an explicit answer to this question we think that by imposing the knowledge of the reduced theory along with crossing symmetry it might possible to reconstruct the missing information. It would be very interesting to show if this is the case. Indeed we think that the following conjecture might be true:[27]

*given any $\mathrm{CFT}_{d-2}$, its Parisi-Sourlas uplift exists and is unique.*

Let us comment on this conjecture. As far as the existence, all results of this paper and of [7] point to the fact that the uplift automatically holds just by kinematics, without ever requiring specific properties of the $\mathrm{CFT}_{d-2}$. So this suggests that all CFTs should have an uplift. About the uniqueness, we find it very unlikely to have two different PS $\mathrm{CFT}_d$ realizations that share such enormous part of their spectrum (which indeed defines a full non-trivial $\mathrm{CFT}_{d-2}$) and yet differ on the CFT data of some more exotic spin representations. E.g. in $d \geq 4$ all the four-point functions of scalar superprimaries of the uplifted theory are completely fixed by the $\mathrm{CFT}_{d-2}$ input. We think that the associated CFT data is too large to admit different completions. It would be very interesting to find a proof of such statements, establishing rigorously the existence and uniqueness of the uplift of generic CFTs.

A very interesting extension of this paper is to consider correlators of operators with spin. This has various applications. First one can extend the relations (76) to spinning conformal blocks. Also it would be interesting to see how the spinning setup constrains the CFT data of theories which factorize. It would be nice to see whether one can reconstruct some spinning

---

[27]This conjecture probably requires a lower bound in the dimensions, which might be $d \geq 3$.

correlators of the uplifted minimal models, like the ones involving the stress tensor. This is not a straightforward uplift since the $4d$ spinning correlators contain more tensor structures with respect to their $2d$ counterparts. However we might hope that such correlators will be constrained enough by superconformal Ward identities.

A very promising idea is to use the uplift of spinning correlators to simplify the numerical conformal bootstrap of setups that involve spinning operators like conserved currents and stress tensors (these setups were sometimes considered in the literature [45–47], but they are very heavy both to implement and at the level of the numerics). Let us explain how this new strategy works. The main idea is to use the basic fact that the supermultiplet of spinning operators (e.g. of spin one and two) contains scalar operators. One could thus perform a bootstrap study of such scalar operators in the uplifted theory to learn about the constraints of the spinning operators in the original theory. Notice that, while the uplifted theory is non-unitary, the unknown OPE coefficients are actually the lower dimensional ones, which satisfy the usual positivity properties that allow for efficient implementations of the numerical bootstrap. We do not know how much information would be encoded in this uplifted scalar bootstrap, but it is surely worth exploring this direction.

Finally one can also study deformations of the PS CFTs. Indeed we might hope that in the near future some PS CFTs like the uplifted minimal models will be fully solved. So it is natural to ask whether one can define new theories by deforming the uplifted ones. If the deformations preserve PS SUSY, then the flows can be dimensionally reduced and thus they would again correspond to uplift of known theories. However one could also deform the model by operators that break (part of) the supersymmetry and get completely new higher dimensional models which do not have any lower dimensional counterpart. This could be a new tool to shed light on the space of higher dimensional QFTs.

# Acknowledgments

We would like to thank Apratim Kaviraj, Bruno Le Floch, Hynek Paul, Lucía Córdova, Slava Rychkov and Vasco Gonçalves for valuable discussions and comments on the draft.

**Funding information** This research was partially supported by the ERC Starting Grant 853507. For the purpose of Open Access, a CC-BY public copyright licence has been applied by the author to the present document.

# A  Coefficients of the relations between conformal blocks

In section 4.2 we showed some explicit expressions for the coefficients $c_{i,j}^{[\vec{S}]}$ of formula (76). In this appendix we present all remaining coefficients.

Let us start by considering the case of $\vec{S} = \mathbf{i}, j\bar{k}$ with $i, j, k$ all different. There are two separate cases. If $(j, k) = (1, 3), (1, 4), (2, 3), (2, 4)$ (independently on $i$ as long as it is different from $j$ and $k$) the set $P_{[\vec{S}]}$ contains only four elements. We can for example compute

$$c_{0,-1}^{[12\bar{3}]} = -\ell\left(\beta_{12} - \Delta + \ell\right)\left(d + \Delta - \beta_{12} + \ell - 4\right), \tag{A.1}$$

$$c_{1,0}^{[12\bar{3}]} = \frac{(\Delta-1)(\Delta-\beta_{12}-\ell)\left(\Delta+\Delta_{12}+\ell\right)\left(\Delta+\Delta_{34}+\ell\right)(d-\Delta-\beta_{12}-\ell-2)}{2(\Delta+\ell-1)(\Delta+\ell)},$$

$$c_{1,-2}^{[12\bar{3}]} = \frac{(\Delta-1)(\ell-1)\ell\left(d+\Delta-\beta_{12}+\ell-4\right)(-d+\Delta+\Delta_{12}-\ell+4)(-2d+\Delta+\beta_{12}-\ell+6)\left(-d+\Delta+\Delta_{34}-\ell+4\right)}{2(d+2\ell-6)(d+2\ell-4)(d-\Delta+\ell-4)(d-\Delta+\ell-3)},$$

$$c_{2,-1}^{[12\bar{3}]} = \frac{(\Delta-1)\Delta\ell\left(\Delta+\Delta_{12}+\ell\right)\left(\Delta+\Delta_{34}+\ell\right)(d-\Delta-\Delta_{12}+\ell-4)(\beta_{12}-2d+\Delta-\ell+6)(\beta_{12}-d+\Delta+\ell+2)\left(\Delta_{34}-d+\Delta-\ell+4\right)}{4(d-2(\Delta+1))(d-2(\Delta+2))(\Delta+\ell-1)(\Delta+\ell)(d-\Delta+\ell-4)(d-\Delta+\ell-3)},$$

where we recall that $\beta_{ij} \equiv \Delta_i + \Delta_j$. All other seven coefficients are defined by the following maps,

$$
\begin{aligned}
c_{i,j}^{[21\bar{4}]} &= \pi_{(12)(34)} c_{i,j}^{[12\bar{3}]}, & c_{i,j}^{[3,1\bar{4}]} &= \pi_{(13)(24)} c_{i,j}^{[12\bar{3}]}, \\
c_{i,j}^{[4,2\bar{3}]} &= \pi_{(14)(23)} c_{i,j}^{[12\bar{3}]}, & c_{i,j}^{[12\bar{4}]} &= (-1)^j \pi_{(34)} c_{i,j}^{[12\bar{3}]}, \\
c_{i,j}^{[21\bar{3}]} &= (-1)^j \pi_{(12)} c_{i,j}^{[12\bar{3}]}, & c_{i,j}^{[32\bar{4}]} &= (-1)^j \pi_{(13)(24)(13)} c_{i,j}^{[12\bar{3}]}, \\
c_{i,j}^{[41\bar{3}]} &= (-1)^j \pi_{(12)(23)(14)} c_{i,j}^{[12\bar{3}]}.
\end{aligned}
\tag{A.2}
$$

In the second case of $\vec{S} = \mathbf{i}, j\bar{k}$ with $(j,k) = (1,2),(3,4)$, the set $P_{[\vec{S}]}$ contains five terms. For $[13\bar{4}]$ we obtain

$$
\begin{aligned}
c_{0,0}^{[13\bar{4}]} &= c_{0,0}^{[1]}\left(\Delta - \beta_{34} - \ell\right), \\
c_{0,-2}^{[13\bar{4}]} &= c_{0,-2}^{[1]}\left(d + \Delta - \beta_{34} + \ell - 4\right), \\
c_{1,-1}^{[13\bar{4}]} &= c_{1,-1}^{[1]}\left(d - \beta_{34} - 2\right), \\
c_{2,0}^{[13\bar{4}]} &= c_{2,0}^{[1]}\left(d - \Delta - \beta_{34} - \ell - 2\right), \\
c_{2,-2}^{[13\bar{4}]} &= c_{2,-2}^{[1]}\left(2d - \Delta - \beta_{34} + \ell - 6\right),
\end{aligned}
\tag{A.3}
$$

where here we relate these coefficients to the $c_{i,j}^{[1]}$ of (83) to get more compact expressions. All other three coefficients are obtained as

$$
c_{i,j}^{[23\bar{4}]} = (-1)^j \pi_{(12)} c_{i,j}^{[13\bar{4}]}, \qquad c_{i,j}^{[31\bar{2}]} = \pi_{(24)(13)} c_{i,j}^{[13\bar{4}]}, \qquad c_{i,j}^{[41\bar{2}]} = \pi_{(14)(23)} c_{i,j}^{[13\bar{4}]}.
\tag{A.4}
$$

Let us now consider two level-two superdescendants, namely the cases $[\vec{S}] = [\mathbf{ij}]$. In all these cases there are five terms in $P_{[\vec{S}]}$. However we find it convenient to separate the cases $[\mathbf{12}],[\mathbf{34}]$ from the rest. For $[\mathbf{12}]$, the result is

$$
\begin{aligned}
c_{0,0}^{[12]} &= c_{0,0}^{[1\bar{2}]}(\beta_{12} - \Delta + \ell + 2)(-\beta_{12} + d - \Delta - \ell - 2)(-\beta_{12} + d + \Delta + \ell - 4), \\
c_{0,-2}^{[12]} &= c_{0,-2}^{[1\bar{2}]}(\beta_{12} - \Delta + \ell)(-\beta_{12} + 2d - \Delta + \ell - 6)(-\beta_{12} + d + \Delta + \ell - 6), \\
c_{1,-1}^{[12]} &= c_{1,-1}^{[1\bar{2}]}\frac{(\beta_{12} - \Delta + \ell)(-\beta_{12} + d - \Delta - \ell - 2)(-\beta_{12} + 2d - \Delta + \ell - 6)(-\beta_{12} + d + \Delta + \ell - 4)}{-\beta_{12} + d - 2}, \\
c_{2,0}^{[12]} &= c_{2,0}^{[1\bar{2}]}(\beta_{12} - \Delta + \ell)(-\beta_{12} + d - \Delta - \ell - 4)(-\beta_{12} + 2d - \Delta + \ell - 6), \\
c_{2,-2}^{[12]} &= c_{2,-2}^{[1\bar{2}]}(\beta_{12} - d + \Delta + \ell + 2)(-\beta_{12} + 2d - \Delta + \ell - 8)(-\beta_{12} + d + \Delta + \ell - 4),
\end{aligned}
\tag{A.5}
$$

where we related these coefficients to $c_{i,j}^{[1\bar{2}]}$ defined in (87). The coefficients $[\mathbf{34}]$ are defined by the relation

$$
c_{i,j}^{[34]} = \pi_{(14)(23)} c_{i,j}^{[12]}.
\tag{A.6}
$$

We then define the case $[\mathbf{13}]$ in terms of the coefficients $c_{i,j}^{[1]}$ of (83),

$$
\begin{aligned}
c_{0,0}^{[13]} &= c_{0,0}^{[1]}\left(\Delta + \Delta_{34} + \ell\right)\left(-\beta_{34} + \Delta - \ell\right), \\
c_{0,-2}^{[13]} &= c_{0,-2}^{[1]}\left(d - \Delta - \Delta_{34} + \ell - 4\right)\left(\beta_{34} - d - \Delta - \ell + 4\right), \\
c_{1,-1}^{[13]} &= c_{1,-1}^{[1]}\frac{\left(\beta_{34} - d + 2\right)\left(\Delta + \Delta_{34} + \ell\right)\left(d - \Delta - \Delta_{34} + \ell - 4\right)}{\Delta_{34}}, \\
c_{2,0}^{[13]} &= c_{2,0}^{[1]}\frac{\left(\Delta + \Delta_{34} + \ell + 2\right)\left(d - \Delta - \Delta_{34} + \ell - 4\right)\left(-\beta_{34} + d - \Delta - \ell - 2\right)}{\Delta - \Delta_{34} + \ell}, \\
c_{2,-2}^{[13]} &= c_{2,-2}^{[1]}\frac{\left(\Delta + \Delta_{34} + \ell\right)\left(d - \Delta - \Delta_{34} + \ell - 6\right)\left(\beta_{34} - 2d + \Delta - \ell + 6\right)}{d - \Delta + \Delta_{34} + \ell - 4}.
\end{aligned}
\tag{A.7}
$$

The other three coefficients are related to these by

$$c_{i,j}^{[14]} = (-1)^j \pi_{(34)} c_{i,j}^{[13]}, \qquad c_{i,j}^{[23]} = (-1)^j \pi_{(12)} c_{i,j}^{[13]}, \qquad c_{i,j}^{[24]} = \pi_{(14)(23)} c_{i,j}^{[13]}. \qquad (A.8)$$

We now focus on the cases $\vec{S} = \mathbf{ij}m\bar{n}$. Again there are two different cases. First we consider the case when $(m,n) = (1,3),(1,4),(2,3),(2,4)$. We start by defining $\vec{S} = \mathbf{132\bar{4}}$,

$$\begin{aligned}
c_{0,-1}^{[132\bar{4}]} &= c_{0,-1}^{[12\bar{3}]} \left(-\Delta + \beta_{34} + \ell\right) \left(d + \Delta - \beta_{34} + \ell - 4\right), \\
c_{1,0}^{[132\bar{4}]} &= c_{1,0}^{[12\bar{3}]} \left(\Delta - \beta_{34} - \ell\right) \left(-d + \Delta + \beta_{34} + \ell + 2\right), \\
c_{1,-2}^{[132\bar{4}]} &= c_{1,-2}^{[12\bar{3}]} \left(-2d + \Delta + \beta_{34} - \ell + 6\right) \left(d + \Delta - \beta_{34} + \ell - 4\right), \\
c_{2,-1}^{[132\bar{4}]} &= c_{2,-1}^{[12\bar{3}]} \left(-d + \Delta + \beta_{34} + \ell + 2\right) \left(2d - \Delta - \beta_{34} + \ell - 6\right),
\end{aligned} \qquad (A.9)$$

where we defined these coefficients in terms of $c_{i,j}^{[12\bar{3}]}$ of (A.1). The other three cases are defined by

$$c_{i,j}^{[142\bar{3}]} = (-1)^j \pi_{(34)} c_{i,j}^{[132\bar{4}]}, \qquad c_{i,j}^{[241\bar{3}]} = \pi_{(12)(34)} c_{i,j}^{[132\bar{4}]}, \qquad c_{i,j}^{[231\bar{4}]} = (-1)^j \pi_{(12)} c_{i,j}^{[132\bar{4}]}. \qquad (A.10)$$

Now we turn to $\vec{S} = \mathbf{ij}m\bar{n}$ with $(m,n) = (1,2),(3,4)$ and show the coefficients for $\vec{S} = \mathbf{341\bar{2}}$,

$$\begin{aligned}
c_{0,0}^{[341\bar{2}]} &= c_{0,0}^{[1\bar{2}]} \left(\beta_{34} - \Delta + \ell\right) \left(\beta_{34} - \Delta + \ell + 2\right) \left(\beta_{34} - d + \Delta + \ell + 2\right) \left(d - \beta_{34} + \Delta + \ell - 4\right), \\
c_{0,-2}^{[341\bar{2}]} &= c_{0,-2}^{[1\bar{2}]} \left(\beta_{34} - \Delta + \ell\right) \left(2d - \beta_{34} - \Delta + \ell - 6\right) \left(d - \beta_{34} + \Delta + \ell - 6\right) \left(d - \beta_{34} + \Delta + \ell - 4\right), \\
c_{1,-1}^{[341\bar{2}]} &= c_{1,-1}^{[1\bar{2}]} \left(\beta_{34} - \Delta + \ell\right) \left(d - \beta_{34} - \Delta - \ell - 2\right) \left(2d - \beta_{34} - \Delta + \ell - 6\right) \left(d - \beta_{34} + \Delta + \ell - 4\right), \\
c_{2,0}^{[341\bar{2}]} &= c_{2,0}^{[1\bar{2}]} \left(\beta_{34} - \Delta + \ell\right) \left(d - \beta_{34} - \Delta - \ell - 4\right) \left(d - \beta_{34} - \Delta - \ell - 2\right) \left(2d - \beta_{34} - \Delta + \ell - 6\right), \\
c_{2,-2}^{[341\bar{2}]} &= c_{2,-2}^{[1\bar{2}]} \left(\beta_{34} - d + \Delta + \ell + 2\right) \left(2d - \beta_{34} - \Delta + \ell - 8\right) \left(2d - \beta_{34} - \Delta + \ell - 6\right) \left(d - \beta_{34} + \Delta + \ell - 4\right),
\end{aligned} \qquad (A.11)$$

where $c_{i,j}^{[1\bar{2}]}$ is defined in (87). The coefficients labelled by $\mathbf{123\bar{4}}$ are then obtained through the map

$$c_{i,j}^{[123\bar{4}]} = \pi_{(14)(23)} c_{i,j}^{[341\bar{2}]}. \qquad (A.12)$$

We proceed by defining the coefficients with $\vec{S} = \mathbf{ijk}$. For all four cases there are always five coefficients. Let us define $c_{i,j}^{[123]}$,

$$\begin{aligned}
c_{0,0}^{[123]} &= c_{0,0}^{[13\bar{4}]} \left(\beta_{34} - \Delta + \ell + 2\right) \left(\beta_{34} - d - \Delta - \ell + 4\right) \left(\beta_{34} - d + \Delta + \ell + 2\right), \\
c_{0,-2}^{[123]} &= c_{0,-2}^{[13\bar{4}]} \left(\beta_{34} - \Delta + \ell\right) \left(-\beta_{34} + 2d - \Delta + \ell - 6\right) \left(-\beta_{34} + d + \Delta + \ell - 6\right), \\
c_{1,-1}^{[123]} &= c_{1,-1}^{[13\bar{4}]} \frac{\left(\beta_{34} - \Delta + \ell\right) \left(-\beta_{34} + d - \Delta - \ell - 2\right) \left(-\beta_{34} + 2d - \Delta + \ell - 6\right) \left(-\beta_{34} + d + \Delta + \ell - 4\right)}{-\beta_{34} + d - 2}, \\
c_{2,0}^{[123]} &= c_{2,0}^{[13\bar{4}]} \left(\beta_{34} - \Delta + \ell\right) \left(-\beta_{34} + d - \Delta - \ell - 4\right) \left(-\beta_{34} + 2d - \Delta + \ell - 6\right), \\
c_{2,-2}^{[123]} &= c_{2,-2}^{[13\bar{4}]} \left(\beta_{34} - d + \Delta + \ell + 2\right) \left(-\beta_{34} + 2d - \Delta + \ell - 8\right) \left(-\beta_{34} + d + \Delta + \ell - 4\right),
\end{aligned} \qquad (A.13)$$

where $c_{i,j}^{[13\bar{4}]}$ is defined in (A.3). The other three coefficients are then obtained as

$$c_{i,j}^{[124]} = \pi_{(23)(14)} c_{i,j}^{[134]}, \qquad c_{i,j}^{[123]} = \pi_{(24)(13)} c_{i,j}^{[134]}, \qquad c_{i,j}^{[234]} = (-1)^j \pi_{(12)} c_{i,j}^{[134]}. \qquad (A.14)$$

Finally we define the last coefficients related to the top primary components of all four operators. This takes the form

$$\begin{aligned}
c_{0,0}^{[1234]} &= c_{0,0}^{[341\bar{2}]} \left(\beta_{12} - \Delta + \ell + 2\right) \left(d - \beta_{12} - \Delta - \ell - 2\right) \left(d - \beta_{12} + \Delta + \ell - 4\right), \\
c_{0,-2}^{[1234]} &= c_{0,-2}^{[341\bar{2}]} \left(\beta_{12} - \Delta + \ell\right) \left(2d - \beta_{12} - \Delta + \ell - 6\right) \left(d - \beta_{12} + \Delta + \ell - 6\right), \\
c_{1,-1}^{[1234]} &= c_{1,-1}^{[341\bar{2}]} \frac{\left(\beta_{12} - \Delta + \ell\right) \left(d - \beta_{12} - \Delta - \ell - 2\right) \left(2d - \beta_{12} - \Delta + \ell - 6\right) \left(d - \beta_{12} + \Delta + \ell - 4\right)}{d - \beta_{12} - 2}, \\
c_{2,0}^{[1234]} &= c_{2,0}^{[341\bar{2}]} \left(\beta_{12} - \Delta + \ell\right) \left(d - \beta_{12} - \Delta - \ell - 4\right) \left(2d - \beta_{12} - \Delta + \ell - 6\right), \\
c_{2,-2}^{[1234]} &= c_{2,-2}^{[341\bar{2}]} \left(\beta_{12} - d + \Delta + \ell + 2\right) \left(2d - \beta_{12} - \Delta + \ell - 8\right) \left(d - \beta_{12} + \Delta + \ell - 4\right),
\end{aligned} \qquad (A.15)$$

where the coefficients $c_{i,j}^{[341\bar{2}]}$ are defined in (A.11).

# B Resolving the singularities of the conformal blocks

As we explained in section 4.3, there are various conformal block exchanges which naively diverge in the PS uplift of a unitary CFT. These correspond to scalars with $\Delta = \frac{d}{2} - 2, \frac{d}{2} - 1$ and operators with $\Delta = d + \ell - 3, d + \ell - 2, d + \ell - 1$ and spin $\ell$. In this appendix we show that all the these exchanges are well defined, and non-singular.

$\boxed{d/2 - 1}$

First we pursue the study of the exchange of an operator with the same labels of a free scalar in a PS CFT. In section 4.3 we explained what happens to the lowest component of the four-point function but we need to investigate also all other 42 components. The result is that for all 26 components that belong to the set $P^{(0)}$ (like the lowest component) the same argument of section 4.3 holds. In particular we checked that all components $[\vec{S}] \in P^{(0)}$ of the scalar superblocks collapse to the sum of two terms

$$\tilde{g}_{d/2-1\,\ell=0}^{[\vec{S}]} \equiv \lim_{\Delta \to d/2-1} c_{0,0}^{[\vec{S}]} \Sigma_{[\vec{s}]} g_{\Delta\,\ell=0} + c_{2,0}^{[\vec{S}]} \Sigma_{[\vec{s}]} g_{\Delta+2\,\ell=0}\,, \tag{B.1}$$

where we checked that the coefficients satisfy

$$c_{2,0}^{[\vec{S}]}/c_{0,0}^{[\vec{S}]} \sim -\frac{R_{\mathrm{III},1}}{\Delta - (d/2 - 1)}\,, \qquad \text{for} \qquad \Delta \to d/2 - 1\,. \tag{B.2}$$

So, as above, for all these cases, supersymmetry implies that the block $\tilde{g}_{d/2-1\,\ell=0}^{[\vec{S}]}$ is finite. For the 16 cases belonging to $[\vec{S}] \in P^{(1)}$ the situation is easier because the superblocks reduces to a single scalar block with dimension $\Delta = d/2$, which has no poles. For the remaining case in $P^{(2)}$ the scalar superblock collapses to zero, so again no singularity is present.

$\boxed{\Delta = d + \ell - 2}$

We now consider the case of the exchange of an PS superblock with spin $\ell$ and dimension $\Delta = d + \ell - 2$. Again we can think of this as arising from the uplift of a $\hat{d}$-dimensional block with the same labels. As before in $\hat{d}$ dimensions there are no poles for this exchange so the same should happen also in $d$, even if this is not trivial since a single conformal block has a pole at $\Delta = d + \ell - 2$. In this case the superblocks of all 26 components $P^{(0)}$ generically contain five contributions of $g_{\Delta\,\ell}, g_{\Delta\,\ell-2}, g_{\Delta+1\,\ell-1}, g_{\Delta+2\,\ell}, g_{\Delta+2\,\ell-2}$, where $g_{\Delta\,\ell}$ is the only one that has pole at $\Delta = d + \ell - 2$. The combination

$$\tilde{g}_{d+\ell-2\,\ell}^{[\vec{S}]} \equiv \lim_{\Delta \to d+\ell-2} c_{0,0}^{[\vec{S}]} \Sigma_{[\vec{s}]} g_{\Delta\,\ell} + c_{1,-1}^{[\vec{S}]} \Sigma_{[\vec{s}]} g_{\Delta+1\,\ell-1}\,, \tag{B.3}$$

which is contained in the superblock is perfectly tuned to cancel the pole of formula (98), indeed

$$c_{1,-1}^{[\vec{S}]}/c_{0,0}^{[\vec{S}]} \sim -\frac{R_{\mathrm{II},1}}{\Delta - (d + \ell - 2)}\,, \qquad \text{for} \qquad \Delta \to d + \ell - 2\,, \tag{B.4}$$

for all $[\vec{S}] \in P^{(0)}$. The rest of the $c_{i,j}^{[\vec{S}]}$ and respective conformal blocks inside the superblock are finite. It is also easy to see that all the blocks exchanged in the cases $P^{(1)}$ and $P^{(2)}$ are not singular.

$$\boxed{\Delta = d + \ell - 3}$$

We now turn to the exchange of a superblock with spin $\ell$ and dimension $\Delta = d + \ell - 3$, which again arises from the uplift of a non-singular block in $\hat{d}$ dimensions. Let us consider the components in $P^{(0)}$ of the superblock. As for the previous case they contain generically five contributions, where only $g_{\Delta \ell}$ has a pole at $\Delta = d + \ell - 3$ with residue equal to $R_{\mathrm{II},2} g_{\Delta+2\ell-2}$. In this case the combination

$$\tilde{g}^{[\vec{S}]}_{d+\ell-3\,\ell} \equiv \lim_{\Delta \to d+\ell-3} c^{[\vec{S}]}_{0,0} \Sigma_{[\vec{s}]} g_{\Delta \ell} + c^{[\vec{S}]}_{2,-2} \Sigma_{[\vec{s}]} g_{\Delta+2\ell-2}, \tag{B.5}$$

given by the superblock cancels the pole because of

$$c^{[\vec{S}]}_{1,-1}/c^{[\vec{S}]}_{0,0} \sim -\frac{R_{\mathrm{II},2}}{\Delta - (d+\ell-3)}, \qquad \text{for} \qquad \Delta \to d+\ell-3, \tag{B.6}$$

for all $[\vec{S}] \in P^{(0)}$. In this case also for the components in $P^{(1)}$ there are poles to consider which are due to $g_{\Delta+1\,\ell}$ and $g_{\Delta\,\ell-1}$, however in both cases the pole is cancelled because these blocks appear in the combination

$$\tilde{g}^{[\vec{S}]}_{d+\ell-2\,\ell} \equiv \lim_{\Delta \to d+\ell-3} c^{[\vec{S}]}_{1,0} \Sigma_{[\vec{s}]} g_{\Delta+1\,\ell} + c^{[\vec{S}]}_{2,-1} \Sigma_{[\vec{s}]} g_{\Delta+2\,\ell-1}, \tag{B.7}$$

$$\tilde{\tilde{g}}^{[\vec{S}]}_{d+\ell-3\,\ell-1} \equiv \lim_{\Delta \to d+\ell-3} c^{[\vec{S}]}_{0,-1} \Sigma_{[\vec{s}]} g_{\Delta\,\ell-1} + c^{[\vec{S}]}_{1,-2} \Sigma_{[\vec{s}]} g_{\Delta+1\,\ell-2}, \tag{B.8}$$

where the coefficients satisfy

$$c^{[\vec{S}]}_{2,-1}/c^{[\vec{S}]}_{1,0} \sim -\frac{R_{\mathrm{II},1}}{\Delta - (d+\ell-3)}, \qquad c^{[\vec{S}]}_{1,-2}/c^{[\vec{S}]}_{0,-1} \sim -\frac{R_{\mathrm{II},1}|_{\ell \to \ell-1}}{\Delta - (d+\ell-3)}, \quad \text{for} \quad \Delta \to d+\ell-3. \tag{B.9}$$

Finally the block in $P^{(2)}$ is not singular.

Now we turn to the uplift of blocks that are at the unitarity bound in $\hat{d}$ dimensions. Because of this, already in $\hat{d}$ dimensions, they must satisfy some constraints in order to be exchanged (as shown in (97) and (99)).

$$\boxed{\Delta = d/2 - 2}$$

Let us start by a the exchange of a scalar block with $\Delta = \frac{d}{2} - 2$, which corresponds to the exchange of a free field in $\hat{d}$ dimensions. The $\hat{d}$-dimensional block must satisfy the condition (97) with $d \to \hat{d}$. For simplicity we focus on the case

$$\Delta_{12}, \Delta_{34} = \pm\left(\frac{d}{2} - 2\right). \tag{B.10}$$

Let us consider the components $[\vec{S}] \in P^{(0)}$ of the superblock. Because of $\ell = 0$, they are written as a liner combination of only two conformal blocks, namely $g_{\Delta\,\ell=0}$ and $g_{\Delta+2\,\ell=0}$ (see e.g. (100)) where now in principle both blocks would diverge. However when we impose (B.10) we find that the coefficients $c^{[\vec{S}]}_{2,0}$ always vanish, so the superblock collapses to a single contribution

$$\lim_{\Delta \to \frac{d}{2}-2} c^{[\vec{S}]}_{0,0} \Sigma_{[\vec{s}]} g_{\Delta\,\ell=0}, \tag{B.11}$$

where the remaining single block diverges as $g_{\Delta\,\ell=0} \sim \frac{R_{\mathrm{III},2}}{\Delta - (\frac{d}{2}-2)} g_{\Delta+4\,\ell=0}$ for $\Delta \to \frac{d}{2} - 2$. Fortunately we could show that this singularity is always avoided. Indeed after imposing (B.10) one of the following two conditions happens, depending on the choice of $[\vec{S}]$:

- either $\Sigma_{[\vec{s}]} R_{\mathrm{III},2} = 0$ (e.g. for the lowest component) thus the superblock is non-singular,

- or $c_{0,0}^{[\vec{S}]} = 0$, thus the superblock vanishes.

In both cases the resulting superblock is non-singular. This resolves all singularities for the components $P^{(0)}$. The situation for $P^{(1)}$ is similar: the superblock collapses to

$$\lim_{\Delta \to \frac{d}{2} - 2} c_{1,0}^{[\vec{S}]} \Sigma_{[\vec{S}]} g_{\Delta + 1 \, \ell = 0} \,, \tag{B.12}$$

where $g_{\Delta + 1, \ell = 0}$ in principle has a pole at $\Delta = \frac{d}{2} - 2$. However this singularity is avoided either because the respective residue $\Sigma_{[\vec{S}]} R_{\mathrm{III},1}$ vanishes, or because the coefficient in front of the block is zero $c_{1,0}^{[\vec{S}]} = 0$ depending on the case $[\vec{S}]$. To conclude, for the component in $P^{(2)}$ the scalar superblocks vanish.

$$\boxed{\Delta = d + \ell - 4}$$

Finally we turn to the exchange of a conserved tensor in $\hat{d}$ dimensions, namely an operator of spin $\ell$ and dimension $\Delta = d + \ell - 4$, where we have to impose (99) (with $d \to \hat{d}$). In particular, for conserved currents, we should require a stronger version of (99), namely

$$\Delta_{12}, \Delta_{34} = 0 \,. \tag{B.13}$$

Let us start by analyzing the components $[\vec{S}] \in P^{(0)}$ of the superblock and check that the five conformal block contributions are non-singular. First we find that two blocks never contribute because, when (B.13) holds, $c_{1,-1}^{[\vec{S}]}, c_{2,-2}^{[\vec{S}]} = 0$. Therefore the superblock collapses to

$$\lim_{\Delta \to d + \ell - 4} c_{0,0}^{[\vec{S}]} \Sigma_{[\vec{S}]} g_{\Delta \ell} + c_{0,-2}^{[\vec{S}]} \Sigma_{[\vec{S}]} g_{\Delta \ell - 2} + c_{2,0}^{[\vec{S}]} \Sigma_{[\vec{S}]} g_{\Delta + 2 \ell} \,. \tag{B.14}$$

The three blocks naively have a pole at $\Delta = d + \ell - 4$, but we shall argue that these singularities are always avoided. First it is useful to notice that for all $[\vec{S}] \in P^{(0)}$ the action $\Sigma_{[\vec{S}]}$ on (B.13) is such that $\Sigma_{[\vec{S}]} \Delta_{12}, \Sigma_{[\vec{S}]} \Delta_{34} = 0, +2, -2$. Keeping this in mind it is easy to see that the block $\Sigma_{[\vec{S}]} g_{\Delta \ell}$ has no pole at $\Delta = d + \ell - 4$ because the residue $\Sigma_{[\vec{S}]} R_{\mathrm{II}, n=3}$ vanishes. The fate of the remaining two contributions depends on whether $\Sigma_{[\vec{S}]}$ acts trivially or not. When $\Sigma_{[\vec{S}]} = 1$ the blocks $g_{\Delta + 2 \ell}$ and $g_{\Delta \ell - 2}$ have no singularity at $\Delta = d + \ell - 4$ because the residue $R_{\mathrm{II}, n=1}$ at this pole vanishes thanks to (B.13). Conversely for all $[\vec{S}]$ such that $\Sigma_{[\vec{S}]}$ is non-trivial, we checked that the coefficients $c_{0,-2}^{[\vec{S}]}, c_{2,0}^{[\vec{S}]}$ are always zero which implies that the associated blocks are not exchanged (this is important since the residue $\Sigma_{[\vec{S}]} R_{\mathrm{II},1}$ at their pole $\Delta = d + \ell - 4$ can now be non-vanishing because of the shift).

Let us now consider the components in $P^{(1)}$ (which all satisfy $\Sigma_{[\vec{S}]} \Delta_{12}, \Sigma_{[\vec{S}]} \Delta_{34} = \pm 1$). First we checked that all $c_{1,-2}^{[\vec{S}]}, c_{2,-1}^{[\vec{S}]}$ vanish because of (B.13). This leaves us with a superblock

$$\lim_{\Delta \to d + \ell - 4} c_{0,-1}^{[\vec{S}]} \Sigma_{[\vec{S}]} g_{\Delta \ell - 1} + c_{1,0}^{[\vec{S}]} \Sigma_{[\vec{S}]} g_{\Delta + 1 \ell} \,, \tag{B.15}$$

which is finite. Indeed, while in principle the two blocks have a pole at $\Delta = d + \ell - 4$, the associated residue $\Sigma_{[\vec{S}]} R_{\mathrm{II}, n=2}$ vanishes when $\Sigma_{[\vec{S}]} \Delta_{12}, \Sigma_{[\vec{S}]} \Delta_{34} = \pm 1$. Finally in the case of $P^{(2)}$ the block $g_{\Delta + 1 \ell - 1}$ has a pole at $\Delta = d + \ell - 4$ but the residue $R_{\mathrm{II}, n=1}$ vanishes because of (B.13).

With this we finished the study of possible singularities of the Parisi-Sourlas uplift of all $\hat{d}$-dimensional blocks with $\Delta$ above/at the unitarity bounds. We conclude that, because of supersymmetry, the uplifted blocks are such that the singularities are always cancelled.

Table 10: The first global primaries currents in the Virasoro multiplet of the identity.

| $n$ | Holomorphic global primaries currents at level $n$ |
|---|---|
| 1 | 0 |
| 2 | $c_1 L_{-2}\mathbb{I}$ |
| 3 | 0 |
| 4 | $c_1(L_{-4}-\frac{5}{3}L_{-2}^2)\mathbb{I}$ |
| 5 | 0 |
| 6 | $(c_1 L_{-6}+c_2 L_{-4}L_{-2}+\frac{21c_1-10c_2}{4}L_{-3}^2+\frac{5c_2-14c_1}{3}L_{-2}^2 L_{-2})\mathbb{I}$ |
| 7 | 0 |
| 8 | $\left(c_1 L_{-8}+c_2 L_{-6}L_{-2}+c_3 L_{-5}L_{-3}+\frac{-15c_1+7c_2+9c_3}{30}L_{-2}^4+\frac{9(5c_1-7c_2-9c_3)}{50}L_{-4}^2+\frac{60c_1-63c_2-56c_3}{20}L_{-3}^2 L_{-2}+\frac{-15c_1+21c_2+17c_3}{5}L_{-4}L_{-2}^2\right)\mathbb{I}$ |
| 9 | $c_1(8L_{-9}-6L_{-7}L_{-2}+12L_{-6}L_{-3}-8L_{-5}L_{-2}^2-5L_{-3}^2 L_{-3}+12L_{-4}L_{-3}L_{-2})\mathbb{I}$ |

## C  Global multiplets inside Virasoro multiplets

In this appendix we study the global primaries inside Virasoro multiplets. Given a Virasoro primary $O$ we can build the possible states in the module as $\bar{L}_{-k}^{\bar{n}_k}\cdots\bar{L}_{-1}^{\bar{n}_1}L_{-k}^{n_k}\cdots L_{-1}^{n_1}O$. Global primaries are annihilated by $L_1$ and $\bar{L}_1$, while descendants are created by applying $L_{-1},\bar{L}_{-1}$ on global primaries. In order to count the holomorphic global primaries we can take the holomorphic Virasoro character $\chi_h(q)=q^h\prod_{k=1}^{\infty}\frac{1}{1-q^k}$ and multiply it by $(1-q)$, indeed $\frac{1}{1-q}$ counts the descendants in a global multiplet. So the number of global primaries in a non-singular Virasoro multiplet is encoded by $\chi_h(q)\chi_{\bar{h}}(\bar{q})(1-q)(1-\bar{q})$.

We are interested in the multiplets which appear in minimal models. These are by definition singular, and thus satisfy BPZ shortening conditions. Let us first focus on the identity multiple which satisfies the condition $L_n\mathbb{I}$ for all $n\geq-1$ (and similarly for the antiholomorphic part). The possible states in the module are defined by $\bar{L}_{-k}^{\bar{n}_k}\cdots\bar{L}_{-2}^{\bar{n}_2}L_{-k}^{n_k}\cdots L_{-2}^{n_2}\mathbb{I}$, which can be global primary or descendants. E.g. $L_{-2}\mathbb{I}$ is a global primary, while $L_{-3}\mathbb{I}=L_{-1}L_{-2}\mathbb{I}$ is a descendant. Let us focus on just the holomorphic part of the multiplet, which is very important since it only contains global conserved currents of spin $\ell=\sum_i i n_i=\Delta$, where $\ell$ also counts the Virasoro level (see [48] for a related discussion). In table 10 we present the first few holomorphic global primaries in the identity multiplet. The constants $c_i$ in table 10 parametrize the dimensionality of the space of such operators at each level. E.g. at level eight there are three constants $c_1,c_2,c_3$ and thus three possible independent primaries. It is easy to generate a table of all such primaries at given level (e.g. with a simple code we could generate the 192 primaries at level 30), but this explicit computation quickly becomes too expensive since their number grows very fast. In order to count the number $p_n^{(\mathbb{I})}$ of such holomorphic primaries at level $n$ one can use the following character,

$$\chi^{\mathbb{I}\,\text{global}}(q)=(1-q)\left(\prod_{k=2}^{\infty}\frac{1}{(1-q^k)}-1\right)+1=\sum_{n=0}^{\infty}p_n^{(\mathbb{I})}q^n.\qquad(\text{C.1})$$

This is built as the generating function of all states in the multiplet $\prod_{k=2}^{\infty}\frac{1}{(1-q^k)}$ where the multiplication by $(1-q)$ removes all the global descendants. We subtract one and add one in order to eliminate from the counting all the descendants of the identity since they vanish. It is easy to see that $p_n^{(\mathbb{I})}=p_{n-2}-2p_{n-1}+p_n+\delta_{n,1}$, where $p_n$ counts the number of partitions of the integer $n$. One can easily table these numbers, as shown in table 11. Using the known large $n$ behaviour of the partition of integers, it is straightforward to find that asymptotically

$p_n^{(\mathbb{I})}$ grows as

$$p_n^{(\mathbb{I})} \sim \frac{\pi^2 e^{\pi\sqrt{\frac{2}{3}n}}}{24\sqrt{3}n^2} \qquad (n \to \infty). \tag{C.2}$$

This means that in the Virasoro multiplet of the identity there is an exponentially growing number of higher spin currents of increasingly large spin (recall that $\ell = n$). Using the global conserved current it is possible to construct charges, which will satisfy an algebra inherited from the Virasoro one. It is worth mentioning that there exists an infinite dimensional abelian subalgebra which can be constructed by considering a special linear combination of these currents for every even spin (see e.g. [48]). Finally in order to get all the global primaries inside the Virasoro multiplet, one only needs to combine the holomorphic and antiholomorphic modules (e.g. $\bar{L}_{-2}L_{-2}\mathbb{I}$ is a scalar global primary with $\Delta = 4$). The resulting operators however are typically not holomorphic (antiholomorphic) and they do not define conserved currents.

A similar construction works for the multiplets of $\epsilon$ and $\sigma$. For these cases the shortening conditions are $L_{-2}\epsilon - \frac{3}{4}L_{-1}^2\epsilon = 0$ and $L_{-2}\sigma - \frac{4}{3}L_{-1}^2\sigma = 0$ (and similarly for the antiholomorphic part). One can generate again the global primaries in the multiplet by requiring this shortening condition and that they are annihilated by $L_1$ and $\bar{L}_1$. In table 12 we present the explicit form of the first few operators. We further want to count their number using a character. In this case the holomorphic character is

$$\chi^{O\,\text{global}}(q) = q^{h_O}(1-q)\prod_{k=1}^{\infty}\frac{1-q^2}{1-q^k} = q^{h_O}\sum_{n=0}^{\infty}p_n^{(O)}q^n \qquad (O = \sigma, \epsilon). \tag{C.3}$$

Since the shortening condition has the same structure the two characters are the same for $\epsilon$ and $\sigma$, thus $p_n^{(\sigma)} = p_n^{(\epsilon)}$. Using this formula one can also obtain $p_n^{(\sigma)} = p_n^{(\epsilon)} = p_n - p_{n-1} - p_{n-2} + p_{n-3}$ in terms of the partitions of integers $p_n$. For concreteness in table 13 we show the number of primaries up to degree $n = 30$. Finally we can study the asymptotic behaviour of $p_n^{(\sigma)} = p_n^{(\epsilon)}$ which reads,

$$p_n^{(\sigma)} = p_n^{(\epsilon)} \sim \frac{\pi^2 e^{\pi\sqrt{\frac{2}{3}n}}}{12\sqrt{3}n^2} \qquad (n \to \infty). \tag{C.4}$$

As before one could combine holomorphic and antiholomorphic multiplets to generate all possible global primaries.

## C.1 Global conformal block decomposition of minimal model correlators

Let us perform the conformal block decomposition of the four-point functions of Virasoro primaries (141) in the Ising model. We present in table 14 the decomposition of $\langle\epsilon\epsilon\epsilon\epsilon\rangle$, in table 15 the one of of $\langle\sigma\sigma\epsilon\epsilon\rangle$, in 16 the one of $\langle\sigma\epsilon\epsilon\sigma\rangle$, and finally in 17 the one of $\langle\sigma\sigma\sigma\sigma\rangle$. The notation is as usual that $f_{O_1O_2O_3O_4} = \sum_{\Delta\ell}a_{\Delta\ell}g_{\Delta\ell}$ (where we suppress the dependence of the external operators in $a_{\Delta\ell}$). It is important to notice that all the decompositions are compatible with tables 11 and 13. In particular the latter predict when a coefficient $a_{\Delta,\ell}$ is missing. On the other hand the opposite is not true, meaning that there are more operators in tables 11 and 13 then the ones that are really exchanged in the OPE. This might be puzzling at first, but it is ultimately due to the infinitely many symmetries that minimal models possess. Such symmetries impose new selection rules which at first sight are not expected, e.g. they must

Table 11: Number of global primaries $p_n^{(\mathbb{I})}$ at level $n$ in the module of the identity.

| $n$ | 1 | 2 | 3 | 4 | 5 | 6 | 7 | 8 | 9 | 10 | 11 | 12 | 13 | 14 | 15 | 16 | 17 | 18 | 19 | 20 | 21 | 22 | 23 | 24 | 25 | 26 | 27 | 28 | 29 | 30 |
|---|---|---|---|---|---|---|---|---|---|---|---|---|---|---|---|---|---|---|---|---|---|---|---|---|---|---|---|---|---|---|
| $p_n^{(\mathbb{I})}$ | 0 | 1 | 0 | 1 | 0 | 2 | 0 | 3 | 1 | 4 | 2 | 7 | 3 | 10 | 7 | 14 | 11 | 22 | 17 | 32 | 28 | 45 | 43 | 67 | 63 | 95 | 96 | 134 | 139 | 192 |

Table 12: First holomorphic global primaries in the multiplets of $O = \sigma, \epsilon$ where respectively $h = \frac{1}{16}, \frac{1}{2}$.

| $n$ | Holomorphic global primaries in the multiplets of $O = \sigma, \epsilon$ at level $n$ |
|---|---|
| 1 | 0 |
| 2 | 0 |
| 3 | $c_1\left(L_{-3} - \frac{2L_{-2}L_{-1}}{3h+1}\right)O$ |
| 4 | $c_1\left(L_{-3}L_{-1} - \frac{2}{5}(h-1)L_{-4} - \frac{2}{3}L_{-2}^2\right)O$ |
| 5 | $c_1\left(h^2 L_{-5} - 3h L_{-4}L_{-1} + (5h+2)L_{-3}L_{-2} - 2L_{-2}^2 L_{-1}\right)O$ |
| 6 | $\left[\frac{L_{-3}^2(40c_2 - 3(3h-1)(2c_2 h + 7c_1))}{4(9h+5)} + \frac{(c_2(2h+5)+7c_1)\left(9L_{-3}L_{-2}L_{-1} - 4L_{-2}^2 L_{-2}\right)}{3(9h+5)} + c_1 L_{-6} + c_2\left(L_{-5}L_{-1} - 2L_{-4}L_{-2}\right)\right]O$ |

Table 13: Number of global primaries $p_n^{(\sigma)} = p_n^{(\epsilon)}$ at level $n$ in the multiplets of $\sigma$ and $\epsilon$.

| $n$ | 1 | 2 | 3 | 4 | 5 | 6 | 7 | 8 | 9 | 10 | 11 | 12 | 13 | 14 | 15 | 16 | 17 | 18 | 19 | 20 | 21 | 22 | 23 | 24 | 25 | 26 | 27 | 28 | 29 | 30 |
|---|---|---|---|---|---|---|---|---|---|---|---|---|---|---|---|---|---|---|---|---|---|---|---|---|---|---|---|---|---|---|
| $p_n^{(\sigma)}$ | 0 | 0 | 1 | 1 | 1 | 2 | 2 | 3 | 4 | 5 | 6 | 9 | 10 | 13 | 17 | 21 | 25 | 33 | 39 | 49 | 60 | 73 | 88 | 110 | 130 | 158 | 191 | 230 | 273 | 331 |

Table 14: Decomposition of $\epsilon\epsilon\epsilon$: coefficients (in the identity multiplet) with $\Delta \leq 10$.

| $(\Delta, \ell)$ | (0,0) | (2,2) | (4,0) | (4,4) | (6,2) | (6,6) | (8,0) | (8,4) | (8,8) | (10,2) | (10,6) | (10,10) |
|---|---|---|---|---|---|---|---|---|---|---|---|---|
| $a_{\Delta\ell}$ | $\frac{1}{2}$ | 4 | $\frac{1}{2}$ | $\frac{8}{5}$ | $\frac{2}{5}$ | $\frac{32}{63}$ | $\frac{1}{200}$ | $\frac{8}{63}$ | $\frac{64}{429}$ | $\frac{1}{315}$ | $\frac{16}{429}$ | $\frac{512}{12155}$ |

Table 15: Decomposition of $\sigma\sigma\epsilon$: coefficients (in the identity multiplet) with $\Delta \leq 10$.

| $(\Delta, \ell)$ | (0,0) | (2,2) | (4,0) | (4,4) | (6,2) | (6,6) | (8,0) | (8,4) | (8,8) | (10,2) | (10,6) | (10,10) |
|---|---|---|---|---|---|---|---|---|---|---|---|---|
| $a_{\Delta\ell}$ | $\frac{1}{2}$ | $\frac{1}{2}$ | $\frac{1}{128}$ | $\frac{3}{40}$ | $\frac{3}{1280}$ | $\frac{5}{336}$ | $\frac{9}{819200}$ | $\frac{5}{10752}$ | $\frac{175}{54912}$ | $\frac{1}{229376}$ | $\frac{175}{1757184}$ | $\frac{441}{622336}$ |

Table 16: Decomposition of $\sigma\epsilon\sigma$: All coefficient with $\Delta \leq 10$.

| $(\Delta, \ell)$ | (1/8,0) | (25/8,3) | (41/8,5) | (49/8,0) | (49/8,6) | (57/8,7) | (65/8,2) | (65/8,8) | (73/8,3) | (73/8,9) |
|---|---|---|---|---|---|---|---|---|---|---|
| $a_{\Delta\ell}$ | $\frac{1}{8}$ | $\frac{-4}{51}$ | $\frac{-16}{1045}$ | $\frac{1}{5202}$ | $\frac{32}{215865}$ | $\frac{-14400}{4411463}$ | $\frac{4}{53295}$ | $\frac{256}{4373439}$ | $\frac{-8}{11009115}$ | $\frac{-265984}{366285997}$ |

ensure that the coefficient $a_{\Delta,\ell}$ with $(\Delta, \ell) = (33/8, 4)$ in table 16 vanishes. We checked that this is indeed true because the OPE coefficients is zero. This can be shown by using the precise form of the level four global primary defined in table 12 and applying it to the three-point function with $\sigma$ and $\epsilon$. Similarly there is an extra selection rule in the OPE of equal operators which forbids the exchanges of global primaries built with an odd holomorphic and/or antiholomorphic level. So for example $L_{-3}\bar{L}_{-3}\epsilon$ is not exchanged in 17.

Table 17: Decomposition of $\sigma\sigma\sigma\sigma$. The table on the top contains even dimensions exchanges which belong to the identity multiplet with $\Delta \leq 10$. The table on the bottom contains odd dimensions exchanges which belong to the $\epsilon$ multiplet with $\Delta \leq 11$.

| $(\Delta,\ell)$ | $(0,0)$ | $(2,2)$ | $(4,0)$ | $(4,4)$ | $(6,2)$ | $(6,6)$ |
|---|---|---|---|---|---|---|
| $a_{\Delta\ell}$ | $\frac{1}{2}$ | $\frac{1}{16}$ | $\frac{1}{8192}$ | $\frac{9}{2560}$ | $\frac{9}{655360}$ | $\frac{25}{57344}$ |
| $(\Delta,\ell)$ | $(8,0)$ | $(8,4)$ | $(8,8)$ | $(10,2)$ | $(10,6)$ | $(10,10)$ |
| $a_{\Delta\ell}$ | $\frac{81}{3355443200}$ | $\frac{25}{14680064}$ | $\frac{15527}{224919552}$ | $\frac{45}{7516192768}$ | $\frac{15527}{57579405312}$ | $\frac{251145}{20392706048}$ |

| $(\Delta,\ell)$ | $(1,0)$ | $(5,4)$ | $(7,6)$ | $(9,0)$ | $(9,8)$ | $(11,2)$ | $(11,10)$ |
|---|---|---|---|---|---|---|---|
| $a_{\Delta\ell}$ | $\frac{1}{8}$ | $\frac{1}{4096}$ | $\frac{1}{20480}$ | $\frac{1}{2147483648}$ | $\frac{1125}{117440512}$ | $\frac{1}{5368709120}$ | $\frac{227}{117440512}$ |

Table 18: Decomposition of the component $[\mathbf{14}]$ of $\epsilon\epsilon\epsilon\epsilon$: Coefficients for exchanges with $\Delta \leq 10$.

| $(\Delta,\ell)$ | $(2,2)$ | $(4,0)$ | $(4,4)$ | $(6,0)$ | $(6,2)$ |
|---|---|---|---|---|---|
| $a_{\Delta\ell}$ | $256$ | $64$ | $\frac{2048}{5}$ | $-96$ | $\frac{512}{5}$ |
| $(\Delta,\ell)$ | $(6,6)$ | $(8,0)$ | $(8,2)$ | $(8,4)$ | $(8,8)$ |
| $a_{\Delta\ell}$ | $\frac{2048}{7}$ | $\frac{192}{5}$ | $\frac{-640}{7}$ | $\frac{512}{7}$ | $\frac{65536}{429}$ |
| $(\Delta,\ell)$ | $(10,0)$ | $(10,2)$ | $(10,4)$ | $(10,6)$ | $(10,10)$ |
| $a_{\Delta\ell}$ | $\frac{-64}{7}$ | $\frac{192}{7}$ | $\frac{-1792}{33}$ | $\frac{16384}{429}$ | $\frac{163840}{2431}$ |

# D  More conformal block decompositions in the uplifted Ising model

As we explained, for each four-point function in $\hat{d} = 2$ there are 43 correlators in $d = 4$ which we can in principle decompose in conformal blocks. Let us consider e.g. the component $[\mathbf{14}]$ of the correlation function of four $\mathcal{E}$, which is defined in table 3. Its conformal block decomposition reads

$$D_{[\mathbf{14}]}f_{\epsilon\epsilon\epsilon\epsilon} = \sum_{\Delta=2\mathbb{N}_{\geq 0}} \sum_{\ell=0,2,\ldots,\Delta} a_{\Delta\ell}g_{\Delta\ell}^{(d=4)}, \tag{D.1}$$

where we computed all $a_{\Delta\ell}$ for $\Delta \leq 10$, which are shown in table 18. We notice that the decomposition of the correlator $\langle \tilde{\mathcal{E}}_{\theta\bar{\theta}}\mathcal{E}_0\mathcal{E}_0\tilde{\mathcal{E}}_{\theta\bar{\theta}} \rangle$ is controlled by the OPE of $\tilde{\mathcal{E}}_{\theta\bar{\theta}}$ with $\mathcal{E}_0$ which are different primaries, but still the only exchanged operators have even spin $\ell$ because of the specific supersymmetric relation between these two fields (for this it is also crucial that the superdimension of $\mathcal{E}$ is equal to one). The decomposition (D.1) shares the same features of the one of (142), in particular it contains the exchanges of the operators in the multiplet of the supercurrents $\mathcal{J}$ of (144). However it is interesting to point out that the operators $\mathcal{J}_{\theta\bar{\theta}}^{\mu_1\ldots\mu_\ell}$ and $\mathcal{J}_0^{\mu_1\ldots\mu_{\ell-2}\theta\bar{\theta}}$ (or better their primary counterparts) which have $\Delta = \ell + 2$ are not exchanged, as we mentioned in the discussion below equation (B.14). This must happen because such exchanges would give rise to a singularity in the conformal block which is due to the fact that the variables $\Delta_{12}, \Delta_{34}$ for the component $[\mathbf{14}]$ are shifted by two units and thus the condition (99) is not satisfied.

Table 19: Decomposition of the component [**1**] of $\epsilon\epsilon\epsilon$: coefficients for exchanged operators with $\Delta \leq 10$.

| $(\Delta,\ell)$ | (2,2) | (4,0) | (4,4) | (6,0) | (6,2) |
|---|---|---|---|---|---|
| $a_{\Delta\ell}$ | $-32$ | $8$ | $\frac{-128}{5}$ | $-4$ | $\frac{32}{5}$ |

| $(\Delta,\ell)$ | (6,6) | (8,0) | (8,2) | (8,4) | (8,8) |
|---|---|---|---|---|---|
| $a_{\Delta\ell}$ | $\frac{-256}{21}$ | $\frac{4}{5}$ | $\frac{-16}{7}$ | $\frac{64}{21}$ | $\frac{-2048}{429}$ |

| $(\Delta,\ell)$ | (10,0) | (10,2) | (10,4) | (10,6) | (10,10) |
|---|---|---|---|---|---|
| $a_{\Delta\ell}$ | $\frac{-4}{35}$ | $\frac{8}{21}$ | $\frac{-32}{33}$ | $\frac{512}{429}$ | $\frac{-4096}{2431}$ |

Table 20: Decomposition of the component [**13$\bar{4}$**] of $\epsilon\epsilon\epsilon$: All exchanged operators with $\Delta \leq 10$.

| $(\Delta,\ell)$ | (2,2) | (4,0) | (4,4) | (6,0) | (6,2) |
|---|---|---|---|---|---|
| $a_{\Delta\ell}$ | $64$ | $16$ | $\frac{256}{5}$ | $-8$ | $\frac{64}{5}$ |

| $(\Delta,\ell)$ | (6,6) | (8,0) | (8,2) | (8,4) | (8,8) |
|---|---|---|---|---|---|
| $a_{\Delta\ell}$ | $\frac{512}{21}$ | $\frac{8}{5}$ | $\frac{-32}{7}$ | $\frac{128}{21}$ | $\frac{4096}{429}$ |

| $(\Delta,\ell)$ | (10,0) | (10,2) | (10,4) | (10,6) | (10,10) |
|---|---|---|---|---|---|
| $a_{\Delta\ell}$ | $\frac{-8}{35}$ | $\frac{16}{21}$ | $\frac{-64}{33}$ | $\frac{1024}{429}$ | $\frac{8192}{2431}$ |

Table 21: Decomposition of the component [**1234**] of $\epsilon\epsilon\epsilon$: All exchanged operators with $\Delta \leq 10$.

| $(\Delta,\ell)$ | (2,2) | (4,4) | (6,0) | (6,6) | (8,0) | (8,2) | (8,8) | (10,0) | (10,2) | (10,4) | (10,10) |
|---|---|---|---|---|---|---|---|---|---|---|---|
| $a_{\Delta\ell}$ | $16384$ | $\frac{1179648}{5}$ | $-6144$ | $\frac{3276800}{7}$ | $\frac{110592}{5}$ | $\frac{-163840}{7}$ | $\frac{205520896}{429}$ | $\frac{-147456}{7}$ | $\frac{307200}{7}$ | $\frac{-344064}{11}$ | $\frac{849346560}{2431}$ |

In table 19, 20 and 21 we present respectively the decompositions of the components [**1**], [**13$\bar{4}$**] and [**1234**] of $\epsilon\epsilon\epsilon$. We see that these share most of the features of the component [**14**]. In the case of [**1234**] there are fewer coefficients because of the specific form of (A.15).

So far we only considered components in $P^{(0)}$, but it is worth showing a couple of examples of components in $P^{(1)}$, where fermionic operators are being exchanged in the OPE. In table 22 and 23 we show respectively the decomposition of the component [**1$\bar{4}$**] and [**12$\bar{3}$**] of $\epsilon\epsilon\epsilon$. A first feature which we observe is that there are odd spin exchanges. This is generically expected since we are taking the OPE of two different operators (even with different bosonic/fermionic statistics). A more detailed property of these decompositions is the presence of new operator exchanges inside the supermultiplet of the supercurrents $\mathcal{J}^{a_1 \dots a_\ell}$. In particular we see superblock exchanges of the type (B.15) which are made of two finite contributions (because $\Delta_{12} = 0 = \Delta_{34}$). The two finite exchanges correspond to the operators $\mathcal{J}_0^{\mu_1 \dots \mu_{\ell-1}\theta}$ and $\mathcal{J}_\theta^{\mu_1 \dots \mu_\ell}$ (or better their primary counterpart which we call $\tilde{\mathcal{J}}_0^{\mu_1 \dots \mu_{\ell-1}\theta}$ and $\tilde{\mathcal{J}}_\theta^{\mu_1 \dots \mu_\ell}$) which have dimensions $\Delta = \ell + 1$ spin respectively equal to $\ell - 1$ and $\ell$. These operators satisfy a generalized conservation equation of the type $\partial_{\mu_1}\partial_{\mu_2}\tilde{\mathcal{J}}_0^{\mu_1 \dots \mu_{\ell-1}\theta} = 0$ and $\partial_{\mu_1}\partial_{\mu_2}\tilde{\mathcal{J}}_\theta^{\mu_1 \dots \mu_\ell} = 0$. From these operators it is therefore possible to build conserved charges. In table 24 we provide the final example of the decomposition of the component [**1$\bar{3}$2$\bar{4}$**] of $\epsilon\epsilon\epsilon$. This belongs to $P^{(2)}$ and thus all the exchanges are charge-two under $Sp(2)$. First we notice that there are exchanges

Table 22: Decomposition of the component $[1\bar{4}]$ of $\epsilon\epsilon\epsilon$: All exchanged operators with $\Delta \leq 10$.

| $(\Delta,\ell)$ | (2,1) | (3,2) | (4,3) | (5,0) | (5,4) | (6,1) | (6,5) |
|---|---|---|---|---|---|---|---|
| $a_{\Delta\ell}$ | 8 | $\frac{-8}{3}$ | $\frac{32}{5}$ | $-2$ | $\frac{-96}{35}$ | $\frac{4}{5}$ | $\frac{64}{21}$ |

| $(\Delta,\ell)$ | (7,0) | (7,2) | (7,6) | (8,1) | (8,3) | (8,7) | (9,0) |
|---|---|---|---|---|---|---|---|
| $a_{\Delta\ell}$ | $\frac{1}{3}$ | $\frac{-8}{7}$ | $\frac{-320}{231}$ | $\frac{-8}{105}$ | $\frac{32}{63}$ | $\frac{512}{429}$ | $\frac{-1}{25}$ |

| $(\Delta,\ell)$ | (9,2) | (9,4) | (9,8) | (10,1) | (10,3) | (10,5) | (10,9) |
|---|---|---|---|---|---|---|---|
| $a_{\Delta\ell}$ | $\frac{4}{27}$ | $\frac{-16}{33}$ | $\frac{-3584}{6435}$ | $\frac{2}{315}$ | $\frac{-32}{693}$ | $\frac{32}{143}$ | $\frac{1024}{2431}$ |

Table 23: Decomposition of the component $[12\bar{3}]$ of $\epsilon\epsilon\epsilon$: All exchanged operators with $\Delta \leq 10$.

| $(\Delta,\ell)$ | (2,1) | (3,0) | (3,2) | (4,1) | (4,3) | (5,0) | (5,2) | (5,4) | (6,1) | (6,3) |
|---|---|---|---|---|---|---|---|---|---|---|
| $a_{\Delta\ell}$ | $-32$ | 16 | $\frac{128}{3}$ | $\frac{-64}{3}$ | $\frac{-512}{5}$ | $-16$ | $\frac{256}{5}$ | $\frac{4224}{35}$ | $\frac{112}{5}$ | $\frac{-2112}{35}$ |

| $(\Delta,\ell)$ | (6,5) | (7,0) | (7,2) | (7,4) | (7,6) | (8,1) | (8,3) | (8,5) | (8,7) | (9,0) |
|---|---|---|---|---|---|---|---|---|---|---|
| $a_{\Delta\ell}$ | $\frac{-2816}{21}$ | $\frac{16}{3}$ | $-32$ | $\frac{1408}{21}$ | $\frac{2560}{21}$ | $\frac{-464}{105}$ | $\frac{2048}{63}$ | $\frac{-1280}{21}$ | $\frac{-4096}{39}$ | $\frac{-28}{25}$ |

| $(\Delta,\ell)$ | (9,2) | (9,4) | (9,6) | (9,8) | (10,1) | (10,3) | (10,5) | (10,7) | (10,9) | |
|---|---|---|---|---|---|---|---|---|---|---|
| $a_{\Delta\ell}$ | $\frac{160}{27}$ | $\frac{-1024}{33}$ | $\frac{2048}{39}$ | $\frac{530432}{6435}$ | $\frac{40}{63}$ | $\frac{-3392}{693}$ | $\frac{3712}{143}$ | $\frac{-265216}{6435}$ | $\frac{-151552}{2431}$ | |

Table 24: Decomposition of the component $[1\bar{3}2\bar{4}]$ of $\epsilon\epsilon\epsilon$: All exchanged operators with $\Delta \leq 10$.

| $(\Delta,\ell)$ | (3,1) | (5,3) | (7,1) | (7,5) | (9,3) | (9,7) |
|---|---|---|---|---|---|---|
| $a_{\Delta,\ell}$ | 16 | $\frac{192}{5}$ | 8 | $\frac{640}{21}$ | $\frac{64}{9}$ | $\frac{7168}{429}$ |

of novel components of the supercurrents, namely $\mathcal{J}_{\theta}^{\mu_1\cdots\mu_{\ell-1}\theta}$, which satisfy $\Delta = \ell+2$ and thus have the quantum numbers of usual conserved currents in four dimensions. We also notice that because of the form of (94), to each block in lower dimension we have at most one block in higher dimension. Moreover (94) is such that all scalar exchanges are automatically projected to zero (and similarly $\Delta = 1$ exchanges). Because of these reasons the decomposition is extremely sparse and contains even less terms than the original one in table 14. It is quite interesting that the action of the differential operator (69) on a correlator provides a new crossing covariant correlator where all scalar contributions are subtracted. It would be nice to find a bootstrap application of this observation.

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
