# Peer review of "The Parisi-Sourlas Uplift and Infinitely Many Solvable 4d CFTs"

_SciPost Physics, doi:SciPost Phys. 18, 056 (2025)_

## Round 1 · Referee Report · Anonymous (Referee 1) · 2024-12-6

Report

The paper explores a class of supersymmetric (SUSY) CFTs called Parisi-Sourlas (PS) CFTs and a duality relation that such a theory enjoys with a non-supersymmetric CFT in two dimensions less. Usually $d$ dimensional PS CFTs are associated with critical points of random field disorder models. A PS CFT has a dimensional reduction property that is: the SUSY kinematics and OPE data, when restricted to a $\hat{d}=d-2$ submanifold, are identical to that of a $d-2$ dimensional CFT. The present paper addresses if this can happen in reverse: given a CFT in $\hat{d}$ dimensions can one identify a $d$-dimensional PS CFT? This is called dimensional uplift. Although a definitive answer to this question is not found, the author has presented interesting points strongly indicating that uplift always exists. The main finding of this paper is the uplift of scalar correlation functions and also how they lead to interesting results in various examples.

The paper is divided into 8 sections: In section 2 PS dimensional reduction is reviewed, and the idea of dimensional uplift for 4-point scalar correlation functions from $d-2$ to $d$ dimensions is defined. Section 3 describes features of uplifted SUSY correlators of a superprimary multiplet. This multiplet packs together different primary operators whose correlation functions are all independent, but SUSY links them to one another through special differential operators. In section 4, the conformal block decomposition in d dimensions is discussed. It is shown that SUSY imposes 43 different relations between $d$ dimensional conformal blocks and $\hat{d}$ dimensional blocks. A subtle fact regarding how poles in the blocks cancel by SUSY in a PS CFT is also clarified. Section 5 is on uplift of generalized free theory (GFF) CFT. SUSY imposes special restrictions on its correlations and recursion relations in the OPE coefficients. For a class of GFF correlators these recursion relations can be solved to obtain an exact analytic expression for the OPE coefficients. This treatment can also be generalized to constrain contact diagram expressions in AdS. In section 6 the uplift of $\hat{d}=2$ minimal models is addressed. It is argued that this uplift straightforwardly follows from the Landau-Ginzburg scalar field theory description. The specific case of Ising model has been addressed in detail. It is pointed out that uplifted minimal models should have multiple families of infinite number of conserved currents. Section 7 addresses the crucial fact some PS SUSY operator representations do not exist in d-2 dimensions and are projected to zero in dimensional reduction. In an uplifted theory this information cannot be reconstructed. E.g. to go from 1d to 3d one only gets correlators restricted to a line and cannot reconstruct the spinning operators in the spectrum. It is shown that spinning superconformal blocks go to zero anyway when restricted to a line.

I have a few minor questions/suggestions:

  1. The GFF theory in section 5 and its uplift are defined by nonlocal quadratic scalar Lagrangians. However the same is also be described by local massive scalar free theory in AdS space without dynamic gravity. In the uplifted AdS theory (that should reflect PS SUSY), can one allow perturbations that do not alter the $\hat{d}$ scalar correlators? 

(Note that certain perturbations to GFF affect its correlators selectively. E.g. a sextic ($\sim \phi^6$) interaction in AdS will not affect $<\phi\phi\phi\phi>$ correlator at leading order but will affect $<\phi\phi\phi^2\phi^2>$. See for instance 2109.13261.)

  1. The eq. 5.26 is termed as a contact diagram in AdS. Shouldn’t there be an integral for the extra AdS coordinate? 

  2. Towards the end of section 8 it is suggested that numerical bootstrap could be a promising way to find an uplifted theory. Even though the theory is non-unitary, the author comments about the positivity of OPE coefficients which should follow from the unitarity of the $\hat{d}$ theory. 
However there are also the operators which project to zero in $\hat{d}$ dimensions. It is not clear to me why should their OPE coefficients should also be positive. 

  3. A main point of interest in PS CFTs is the connection to random field models. Can uplifted correlators be identified as observables in a disorder critical point? It would be nice if some comments are made in this direction in the conclusion section. 

The topic of PS SUSY and dimensional reduction is an old one, but the present paper adds an interesting new direction to it. The paper is well-written and the author has done an exhaustive analysis, with many useful details and simple, interesting examples for clarification of arguments. I request the author to address the above questions and make some appropriate changes. Apart from those I recommend the paper for publication.

Recommendation

Ask for minor revision

  • validity: -
  • significance: -
  • originality: -
  • clarity: -
  • formatting: -
  • grammar: -

Author:  Emilio Trevisani  on 2024-12-27  [id 5068]

(in reply to Report 1 on 2024-12-06)

I thank very much the referee for reading the paper and for the useful comments/questions which I address below:

1) It is for sure possible to add perturbations that selectively affect the correlators, as the example of $\phi^6$ for the correlator $\langle \phi \phi \phi \phi \rangle$ at one loop given by the referee. In the uplifted theory it would be the same story where we replace usual fields by superfields.

2) Formula (5.26) is an integral in flat space, which is well known to be computed as a D-function. The same D-function appears from an AdS computation. A simple proof that the two integrals give the same result can be obtained using the Schwinger representation e.g. (5.4) of [23]. I will add a reference which might help clarifying this point.

3) This is a great point. Indeed it is true that we do not know the sign of the OPE coefficients of the operators projected to zero. However these sometimes are not actually exchanged, depending on the observable and on the space-time dimensions. For example for scalar 4pt functions there are no such operator exchanges in all $\hat d \geq 2$. Similarly there are no such operators for all 4pt functions of traceless and symmetric operators in $\hat d \geq 6$. So this means that there always exist applications of this strategy which would work without extra assumptions (e.g. it could be used to prove no go theorems of CFTs in large dimensions). When these exchanges are present, an interesting option is to try to require positivity of the extra OPE coefficients and check what is the result (in some cases the negativity of some OPE coefficients cannot quite be seen in the numerical bootstrap, e.g. for Wilson Fisher fixed point in fractional dimensions). I avoided these comments in the conclusion, because I feel they are too detailed. Also I would prefer to keep them for future works, where these strategies will be actually implemented.

4) Yes, it is correct, the uplifted theories sometimes define some random field models and their correlators are indeed observable in those cases. Comments on this point are given in the first two pages of the introduction, but indeed it was missing a comment in the conclusion which I will add in the resubmission.

---

## Round 1 · Referee Report · Anonymous (Referee 2) · 2024-12-17

Report

This paper is a far-reaching outgrowth the program of studying Parisi-Sourlas supersymmetric CFTs which was started some years ago by the author in collaboration with Kaviraj and Rychkov. That program was motivated by explaining the lack of dimensional reduction and Parisi-Sourlas of some random field stat-phys models. This paper instead focuses on Parisi-Sourlas (PS) SUSY CFTs per se, and studies the question of dimensional uplift - can any CFT in d-2 dimensions be uplifted to a PS SUSY CFT in d dimensions. The author conjectures in the conclusions that yes. The bulk of the paper is a beautiful compendium of a miriad results linked by mutual consistency which lead the author to this conjecture. I read all of them. I was particularly impressed by the uplift of minimal models, and also by how uplift can help solve questions about ordinary CFTs and their perturbations having nothing to do with PS SUSY. This is a paper of highest caliber which should lead to much followup work.

Recommendation

Publish (surpasses expectations and criteria for this Journal; among top 10%)

---

## Editorial Decision

published